# WHICH INVARIANCE SHOULD WE TRANSFER?
# A CAUSAL MINIMAX LEARNING APPROACH

## ABSTRACT

A major barrier to deploy current machine learning models lies in their sensitivity to dataset shifts. To resolve this problem, most existing studies attempted to transfer stable information to unseen environments. Among these, graph-based methods causally decomposed the data generating process into stable and mutable mechanisms. By removing the effect of mutable generation, they identified a set of stable predictors. However, a key question regarding robustness remains: *which subset of the whole stable information should the model transfer, in order to achieve optimal generalization ability?* To answer this question, we provide a comprehensive minimax analysis that fully characterizes conditions for a subset to be optimal. Particularly in general cases, we propose to maximize over mutable mechanisms (*i.e.*, the source of dataset shifts), which is provable to identify the worst-case risk over all environments. This ensures us to select the optimal subset with the minimal worst-case risk. To reduce computational costs, we propose to search over only equivalent classes in terms of worst-case risk, instead of over all subsets. In cases when the searching space is still large, we turn this subset selection problem into a sparse min-max optimization scheme, which enjoys the simplicity and efficiency of implementation. The utility of our methods is demonstrated on the diagnosis of Alzheimer's Disease and gene function prediction.

## 1 INTRODUCTION

Current machine learning systems, which are commonly deployed based on their in-distribution performance, often encounter *dataset shifts* Subbaswamy et al. (2019) such as covariate shift, label shift, *etc.*, due to changes in the data generating process. When such a shift exists in deployment environments, the model may give unreliable prediction results, which can cause severe consequences in safe-critical tasks such as healthcare (Hendrycks et al., 2021). At the heart of this unreliability issue are *stability* and *robustness* aspects, which refer to the insensitivity of prediction behavior and generalization errors over shifts, respectively.

For example, consider the system deployed to predict the Functional Activities Questionnaire (FAQ) score, which is commonly adopted Mayo (2016) to measure the severity of Alzheimer's Disease (AD). During prediction, the system can only access biomarkers or volumes of brain regions with anonymous demographic information for privacy consideration. However, the changes in demographics can cause shifts in covariates. To achieve reliability for the deployed model, it is desired for its prediction to be stable against demographic changes, and meanwhile to be constantly accurate over all different populations. To incorporate both aspects, this paper targets at finding the most robust (*i.e.*, min-max optimal Müller et al. (2020)) predictor, among the set of stable predictors over all distributions.

To achieve this goal, many studies have proposed to learn invariance to transfer to unseen data. Examples include ICP Peters et al. (2016) and (Arjovsky et al., 2019; Liu et al., 2021; Ahuja et al., 2021) that assumed the prediction mechanism given causal features or representations to be invariant; Anchor Regression Rothenhäusler et al. (2021) that explicitly attributed the variation to exogenous variables. Particularly, the Subbaswamy & Saria (2020); Subbaswamy et al. (2019) causally decomposed the joint distribution into mutable $M$ and stable $S$ sets, with respectively changed and unchanged causal mechanisms. They then proposed to intervene on $M$ to obtain a set of stable predictors. Still, a question regarding robustness remains: *which subset of stable information should the model transfer, in order to be most robust against dataset shifts?* The answer given by

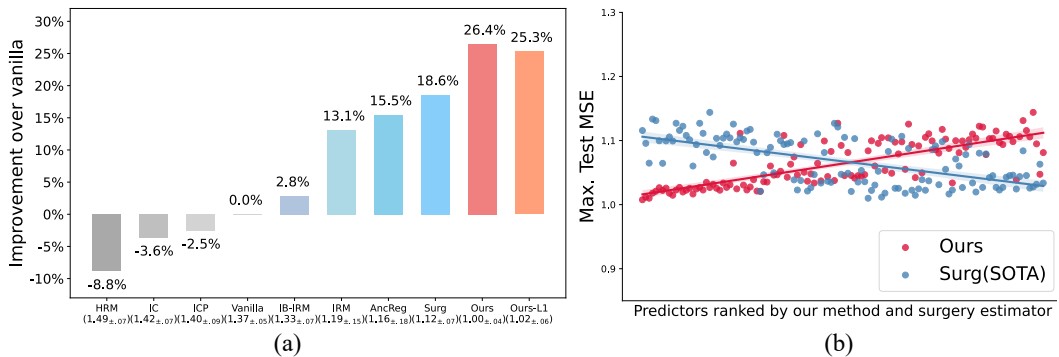

Figure 1: FAQ prediction in Alzheimer's Disease. (a) Maximal mean square error (MSE) over test environments; (b) Maximal MSE of predictors that are ranked in ascending order from left to right, respectively according to the estimated worst-case risk of our method and the validation's loss of the graph surgery estimator Subbaswamy et al. (2019). As shown, our method is more reflective of the maximal MSE than the graph surgery method.

Subbaswamy et al. (2019) was to simply search over all subsets in $S$ and took the one with the minimal validation loss. However, it lacks theoretical and practical guarantees for the validation's loss to reflect the worst-case risk, as shown by Fig. 1 (b).

To give a comprehensive answer, we first provide a graphical condition that is sufficient for the whole stable set to be optimal. This condition can be easily tested via causal discovery. When this condition fails, we prove that the worst-case risk can be identified by maximizing over the generating mechanism of $M$, *i.e.*, the only source of shift. This conclusion ensures us to select the optimal subset in a more accurate way. Consider again the example of FAQ prediction in AD diagnosis, Fig. 1 (b) shows that our method is more reflective of the maximal mean squared error (MSE) than Subbaswamy et al. (2019), which explains our advantage in predicting FAQ across patient groups shown in Fig. 1 (a). Besides, to reduce the searching cost, we propose to search over only equivalent classes in terms of worst-case risk. We however find that in some cases such a search can still be expensive. To improve efficiency in these cases, we turn this subset selection task into a sparse min-max optimization scheme, which alternates between a gradient ascent step on the $M$'s generating function and a sparse optimization with Lasso-type penalty to detect the optimal subset. We demonstrate the utility of our methods on a synthetic dataset and two real-world applications: Alzheimer's Disease diagnosis and gene function prediction.

**Contributions.** We summarize our contributions as follows:

1. We propose to identify the optimal subset of invariance to transfer, guided by a comprehensive min-max analysis. To the best of our knowledge, we are the *first* to comprehensively study the problem of *which part among all sources of invariance should the model transfer*, in the literature of robust learning.

2. We introduce the concept of "equivalent relation" in terms of worst-case risk, in order to analyze the computational complexity, and propose a sparse min-max optimization method as a surrogate scheme to improve efficiency.

3. Our method can significantly outperform others in terms of subset selection and generalization robustness, on Alzheimer's Disease diagnosis and gene function prediction.

## 2 PRELIMINARIES AND BACKGROUND

**Problem Setup & Notations.** We consider the supervised regression scenario, where the system includes a target variable $Y \in \mathcal{Y}$, a multivariate predictive variable $\mathbf{X} := [X_1, ..., X_d] \in \mathbb{R}^d$, and data collected from heterogeneous environments. In practice, different "environments" can refer to different groups of subjects or different experimental settings. We use $\{D_e | e \in \mathcal{E}_{\mathrm{Tr}}\}$ to denote our training data, with $D_e := \{(\boldsymbol{x}_k^e, y_k^e)\}_{k=1}^{n_e} \sim_{i.i.d} p^e(x, y)$ being the data from environment $e$ with

sample size $n_e$. The total number of training samples is $n := \sum_e n_e$. We say a predictor $f : \mathbb{R}^d \to \mathcal{Y}$ is stable if it can be learned from the training environments $\mathcal{E}_{\text{Tr}}$ and transferred to a broader family of environments $\mathcal{E}$ without any adjustment. We denote the stable predictor set as $\mathcal{F}^S$ and the distribution set as $\mathcal{P} := \{P^e(\mathbf{X}, Y)\}_{e \in \mathcal{E}}$, with $P^e(\mathbf{X}, Y)$ the distribution over $\mathbb{R}^d \times \mathcal{Y}$ in environment $e$. For a causal directed acyclic graph (DAG) $G$, we denote the parents, children, and descendants of the node set $\mathbf{V}$ as $\text{Pa}(\mathbf{V})$, $\text{Ch}(\mathbf{V})$ and $\text{De}(\mathbf{V})$, respectively. For a subset $\mathbf{V}' \subset \mathbf{V}$, $G_{\overline{\mathbf{V}'}}$ denotes the sub-graph obtained by deleting edges pointing to any member of $\mathbf{V}'$. We denote conditional independence and d-separation by $\perp$ and $\perp_G$, respectively.

Our goal is to find the most robust predictor $f^*$ among the stable predictor set $\mathcal{F}^S$ using data from $\mathcal{E}_{\text{Tr}}$. A commonly used way Peters et al. (2016); Ahuja et al. (2021) to measure this robustness is to investigate the predictor's worst-case risk, which provides a safeguard for deployment in unseen environments. That is, we want $f^*$ to have the following min-max property:

$$f^*(x) = \operatorname*{argmin}_{f \in \mathcal{F}^S} \max_{P^e \in \mathcal{P}} \mathbb{E}_{P^e}[(Y, f(\boldsymbol{x}))^2]. \tag{1}$$

Next, we introduce the *causal model*, *Markovian and faithfulness* assumptions that our methods are based on. These assumptions are commonly made in the literature of causal inference and learning Pearl (2009); Spirtes et al. (2000); Arjovsky et al. (2019).

**Assumption 2.1** (Causal Model). We assume that $P^e(\mathbf{X}, Y)$ is entailed by an *unknown* DAG $G := (\mathbf{V}, \mathbf{E})$ for all $e \in \mathcal{E}$, where $\mathbf{V} := \mathbf{X} \cup Y$ denotes the node set and $\mathbf{E}$ denotes the edge set. Each variable $V_i \in \mathbf{V}$ is associated with a structural equation $g_i^e$: $V_i \leftarrow g_i^e(\text{Pa}(V_i), U_i)$, where $U_i$ denotes the exogenous variable. Each edge in $\mathbf{E}$ represents a direct causal relationship (Pearl, 1995).

**Assumption 2.2** (Markovian and Faithfulness). The Markovian means $\{U_i\}$ are mutually independent. Together with faithfulness, it means $\forall$ disjoint sets $\mathbf{V}_i, \mathbf{V}_j, \mathbf{V}_k$: $\mathbf{V}_i \perp \mathbf{V}_j | \mathbf{V}_k \iff \mathbf{V}_i \perp_G \mathbf{V}_j | \mathbf{V}_k$.

**Graph Surgery Estimator.** Under assumptions 2.1, 2.2, the graph surgery estimator Subbaswamy et al. (2019) causally decomposed the joint distribution $p^e(x, y)$ into disentangled generating factors:

$$p^e(\boldsymbol{x}, y) = p(y|pa(y)) \prod_{i \in S} p(x_i|pa(x_i)) \prod_{i \in M} p^e(x_i|pa(x_i)), \ d_S := |S|, \ d_M := |M|, \tag{2}$$

where $S, M$ respectively denote stable and mutable sets such that $\mathbf{X}_S := \{X_i | \forall e \in \mathcal{E}, p^e(x_i|pa(x_i)) \equiv p(x_i|pa(x_i))\}$ contains variables with stable mechanisms and $\mathbf{X}_M := \{X_i | \exists \ e_1 \neq e_2 \in \mathcal{E}, \ p^{e_1}(x_i|pa(x_i)) \neq p^{e_2}(x_i|pa(x_i))\}$ contains those with unstable mechanisms. They then removed the unstable mechanisms by intervening on $\mathbf{X}_M$ and obtained a set of stable predictors $\mathcal{F}^S$: $\{f_{S_-} := \mathbb{E}_P[Y|\boldsymbol{x}_{S_-}, do(\boldsymbol{x}_M)]|S_- \subset S\}$ that are independent of $e$. In this regard, identifying $f^*$ in Eq. 1 is equivalent to selecting the optimal subset $S^* \subset S$ such that $f^* = f_{S^*}$.

To identify $S^*$, they shown that the whole set $S$ is optimal under the degeneration condition: $p(y|\boldsymbol{x}_S, do(\boldsymbol{x}_M)) = p(y|\boldsymbol{x}')$ for some $\mathbf{X}' \subset \mathbf{X}$. In the general cases, they searched over all $S_- \subset S$ and selected the best one with minimal held-out validation loss. However, this analysis is theoretically incomplete and practically defective: **i)** it does not provide a procedure to test the degeneration condition, making it inapplicable; **ii)** the selected subset may not be min-max optimal, as the validation loss does not necessarily reflect the worst-case risk (Fig. 1 (b)); **iii)** the searching cost is exponentially expensive w.r.t. $d_S$, making it hard to be applied to large-scale scenarios. In the next section, we will provide a comprehensive min-max analysis to remedy these issues.

## 3 METHODOLOGY

In this section, we introduce our method to identify $S^*$. Specifically, in Sec. 3.1, we first introduce a comprehensive min-max analysis to identify the $S^*$, followed by the learning method in Sec. 3.2. We then analyze the computational complexity in Sec. 3.3 via the lens of $g$-equivalence, and show that its searching cost can still be exponentially expensive in some cases. To improve efficiency in these cases, we in Sec. 3.4 introduce a sparse min-max optimization algorithm, which turns the subset selection problem into a sparse optimization scheme that enjoys model selection consistency.

### 3.1 IDENTIFICATION WITH MIN-MAX ANALYSIS

In this section, we introduce our method to identify the min-max optimal subset $S^*$ with theoretical guarantees. Our analysis is composed of two main results: Thm. 3.1 and Thm. 3.3. First, Thm.

3.1 provides a testable graphical condition that is sufficient for $S^* = S$. When this condition fails, we show that the whole stable set $S$ is not necessarily optimal via a counter-example. We then in Thm. 3.3 provide a sufficient and necessary condition for a subset to be optimal in the general cases.

In the following, we first introduce the graphical condition for $S^* = S$, which is equivalent to the degeneration condition in Subbaswamy et al. (2019).

**Theorem 3.1** (Graphical Condition for $f^* = f_S$). *Suppose assumptions 2.1, 2.2 hold. Denote* $\mathbf{X}_M^0 := \mathbf{X}_M \cap \mathrm{Ch}(Y)$ *as mutable variables in $Y$'s children, and* $\mathbf{K} := \mathrm{De}(\mathbf{X}_M^0) \backslash \mathbf{X}_M^0$ *as descendants of* $\mathbf{X}_M^0$. *Then, $p(y|\boldsymbol{x}_S, do(\boldsymbol{x}_M))$ can degenerate to conditional distribution if and only if $Y$ does not point to any member of* $\mathbf{K}$. *Further, under either of the two conditions, we have $S^* = S$.*

**Example 1.** To illustrate, consider the example shown in Fig. 2, where the graphical condition holds when the dashed arrow from $Y \to \mathbf{K}$ does not exist. To see its equivalence to the degeneration condition, we can set $\mathbf{K}$ as stable variables. When $Y \not\to \mathbf{K}$, the path from $Y$ to $\mathbf{K}$ can be blocked by $\mathbf{X}_M^0$. We then have $Y \perp_{G_{\overline{X_M^0}}} \mathbf{K}|\mathbf{X}_M^0$ and thus $p(y|\boldsymbol{k}, do(\boldsymbol{x}_M^0)) = p(y)$ according to the inference rules in Pearl (2009). While when such a dashed arrow of $Y \to \mathbf{K}$ holds, the $\mathbf{K}$ becomes a collider on the path between $Y$ and $\mathbf{X}_M^0$, making it incapable to remove "do" in $p(y|\boldsymbol{k}, do(\boldsymbol{x}_M^0))$.

Compared to the degeneration condition, our graphical condition is more intuitive and can be easily tested via causal discovery, as guaranteed by the following proposition.

**Proposition 3.2.** *Under assumptions 2.1, 2.2, we have **i)** $\mathbf{K}$ is identifiable, and **ii)** $Y \to \mathbf{K}$ is testable from joint distribution over training environments.*

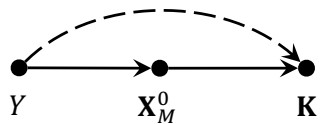

Figure 2: Illustration of Thm. 3.1.

Thm. 3.1 also reminds us that the sufficient condition for $S^* = S$ may not always hold. Indeed, we provide a counter-example in Sect. B.3 in the appendix showing that the whole stable set has a larger worst-case risk than some subsets.

To identify $S^*$ when the graphical condition fails, we turn to estimate the expected worst-case risk of each subset $S_- \subset S$ from $\{D_e\}_{e \in \mathcal{E}_{\mathrm{Tr}}}$. By noticing that the variation of unstable mechanisms in $\mathbf{X}_M$ is the only source of shifts in $P^e(\mathbf{X}, Y)$, we propose to parameterize these mechanisms and let them vary arbitrarily, in order to explore the behavior of the worst-case environment. Specifically, we consider a distribution family $\{P_J\}_J$ for any $J : \mathrm{Pa}(\mathbf{X}_M) \to \mathbf{X}_M$ and $P_J := P(Y, \mathbf{X}_S|do(\mathbf{X}_M = J(Pa(\mathbf{X}_M))))$. By maximizing the population risk of over $J$ for each subset, we can obtain the worst-case risk of this subset, as shown in Thm. 3.3.

**Theorem 3.3** (Min-max Property). *Denote $h^*(S_-) := \max_J \mathbb{E}_{P_J}[(Y - f_{S_-}(\boldsymbol{x}))^2]$ as the maximal expected risk in $P_J$ for subset $S_-$. Then, we have $h^*(S_-) = \max_{P^e \in \mathcal{P}} \mathbb{E}_{P^e}[(Y - f_{S_-}(\boldsymbol{x}))^2]$. In this regard, the optimal subset $S^*$ can be attained via $S^* := \operatorname{argmin}_{S_- \subset S} h^*(S_-)$.*

This theorem informs us to optimize $\mathbb{E}_{P_J}[(Y - f_{S_-}(\boldsymbol{x}))^2]$ over $J$ to obtain $h^*$ for $S_-$, as it equals the worst-case risk of using subset $S_-$. With this theorem, it is sufficient to compare $h^*$ of each subset to identify the optimal one. The following proposition ensures that the optimization is tractable, as each component, *i.e.*, $\mathrm{Pa}(\mathbf{X}_M), P_J, f_{S_-}$ used in optimization is identifiable.

**Proposition 3.4.** *Under assumptions 2.1, 2.2, the $\mathrm{Pa}(\mathbf{X}_M)$, $P_J$, and $f_{S_-}$ are identifiable.*

### 3.2 LEARNING METHOD

According to the last section, we have $S^* = S$ if $Y \not\to \mathbf{K}$. Otherwise, we can simply search over all subsets of $S$ and compare their $h^*$ to identify the optimal one, as similarly adopted in Subbaswamy et al. (2019). However, this exhaustive search can be redundant, as some subsets are equivalent in terms of prediction. Formally speaking, we introduce g-equivalence, *i.e.*, $\sim_G$, as follows:

**Definition 3.5** (g-equivalence). For two subsets $S_i, S_j$, we say $S_i \sim_G S_j$ if $\exists S_{ij} \subseteq S_i \cap S_j$ such that $Y \perp_{G_{\overline{\mathbf{X}_M}}} (\mathbf{X}_{S_i} \cup \mathbf{X}_{S_j} \backslash \mathbf{X}_{S_{ij}})|\mathbf{X}_{S_{ij}}, \mathbf{X}_M$. We call the elements of the quotient space $\mathrm{Pow}(S)/\sim_G$ as g-equivalent classes, and denote $N_G := |\mathrm{Pow}(S)/\sim_G|$ as the number of equivalent classes.

Under assumption 2.2, it is easy to see that if $S_i \sim_G S_j$, then we have $P(Y|\mathbf{X}_{S_i}, do(\mathbf{X}_M)) = P(Y|\mathbf{X}_{S_j}, do(\mathbf{X}_M))$ and thus $S_i, S_j$ have the same efficacy of robustness. In this regard, it suffices

to search over $\mathrm{Pow}(S)/\sim_G$ to identify the optimal subset, rather than exhaustively searching the $\mathrm{Pow}(S)$. To enable this searching, we provide a recovering algorithm that is provable to recover all $g$-equivalent classes. For the reason of coherence and space-saving, we leave this algorithm and its analysis to Sec. C.1 in the appendix. Equipped with $\mathrm{Pow}(S)/\sim_G$, we are now ready to introduce our algorithm to identify $S^*$.

---

**Algorithm 1** Identification of the min-max optimal subset $S^*$ and predictor $f^*$.

---

**INPUT:** The training data $\{D_e | e \in \mathcal{E}_{\mathrm{Tr}}\}$.

1: Causal discovery to obtain the partially directed acylic graph (PDAG).
2: Detect $\mathbf{K}$ and whether $Y \to \mathbf{K}$.
3: **if** $Y \not\to \mathbf{K}$ **then**
4:     Set $S^* = S$ and estimate $f^* = f_S$.           ◁ according to Thm. 3.1
5: **else**
6:     Recover $\mathrm{Pow}(S)/\sim_G$.           ◁ with Alg. 6 in Sec. C.1.
7:     Set $h_{\min} = \infty, S^* = \varnothing$.
8:     **for** $S_G \in \mathrm{Pow}(S)/\sim_G$ **do**
9:         Randomly pick a $S_- \in S_G$, estimate $f_{S_-}$ and $h^*(S_-)$.
10:         **if** $h^*(S_-) < h_{\min}$ **then**
11:             Set $h_{\min} = h^*(S_-), S^* = S_-, f^* = f_{S_-}$.           ◁ according to Thm. 3.3
12:         **end if**
13:     **end for**
14: **end if**
15: **return** $S^*$ and $f^*$.

---

As the causal graph is unknown, Alg. 1 involves **i)** causal discovery to detect $\mathbf{K}$ and whether $Y \to \mathbf{K}$; **ii)** estimation of $f_{S_-}$; and **iii)** estimation of $h^*(S_-)$. In the following, we roughly introduce the main ideas of our method and left the details to Sec. C in the appendix.

**Causal discovery to detect K and examine whether** $Y \to \mathbf{K}$. We first detect a partial directed acyclic graph (PDAG) via the PC algorithm (Spirtes et al., 2000), followed by our method to detect $\mathbf{X}_M$ under the assistance of domain index variable $E$. Specifically, we have $X_i \in \mathbf{X}_M$ iff $E \to X_i$ in the detected PDAG, according to Huang et al. (2020). In a similar way, we can identify $\mathrm{Pa}(X_i)$, $\mathrm{Ch}(X_i)$ for $i \in M$, which is sufficient to detect $\mathbf{X}_M^0 := \mathbf{X}_M \cap \mathrm{Ch}(Y)$. Applying this method iteratively, we can detect $\mathrm{De}(\mathbf{X}_M)$ and $\mathrm{Pa}(X_i)$ for $X_i \in \mathrm{De}(\mathbf{X}_M)$, which is sufficient to identify $\mathbf{K} := \mathrm{De}(\mathbf{X}_M^0) \setminus \mathbf{X}_M^0$ and have $Y \to \mathbf{K}$ iff $\exists X_i \in \mathbf{K}$ such that $Y \in \mathrm{Pa}(X_i)$.

**Estimate** $f_{S_-}$. We adopt soft-intervention Eberhardt & Scheines (2007) to replace $P^e(\mathbf{X}_M | \mathrm{PA}(\mathbf{X}_M))$ with $P(\mathbf{X}_M)$ and define $\bar{P}(\mathbf{X}, Y) := P(Y | \mathrm{Pa}(Y)) \Pi_{i \in S} P(X_i | Pa(X_i)) P(\mathbf{X}_M)$. Then we have $f_{S_-}(\boldsymbol{x}) = \mathbb{E}_{\bar{P}}[Y | \boldsymbol{x}_{S_-}, \boldsymbol{x}_M]$. Here we set $p(\boldsymbol{x}_M) := \sum_{e \in \mathcal{E}_{\mathrm{Tr}}} \left(\frac{p_e}{\sum_{e \in \mathcal{E}_{\mathrm{Tr}}} p_e}\right) p^e(\boldsymbol{x}_M)$, with $p_e \approx n_e/n$. To generate data from $\bar{P}$, we first permute $\mathbf{X}_M$ in a sample-wise manner to generate data from $P(\mathbf{X}_M)$. Then, we recursively regenerate data for each variable in $\mathrm{De}(\mathbf{X}_M)$ from its parents in $G_{\overline{\mathbf{X}_M}}$, by estimating structural equations. This is tractable since $\mathrm{De}(\mathbf{X}_M)$ and the parent nodes for each variable in $\mathrm{De}(\mathbf{X}_M)$ are identifiable, as mentioned earlier.

**Estimate** $h^*(S_-)$. We first learn $h(S_-, J) := \mathbb{E}_{P_J}[(Y - f_{S_-}(\boldsymbol{x}))^2]$. As $f_{S_-}$ can be estimated, we only need to obtain data from $P_J$. As $\mathrm{Pa}(\mathbf{X}_M)$ is identifiable, we iteratively regenerate data for $\mathbf{X}_M$ from $J(\mathrm{Pa}(\mathbf{X}_M))$ and also for $X_i \in \mathrm{Pa}(\mathbf{X}_M)$ from its parents, in order to obtain samples from $p_J$. Then we maximize $h(S_-, J)$ over $J$ to obtain $h^*(S_-)$.

### 3.3 COMPLEXITY ANALYSIS

In this section, we discuss the complexity of Alg. 1. Benefit from the testable condition in Thm. 3.1, our algorithm enjoys a constant cost when $Y$ does not point to any member of $\mathbf{K}$. This situation happens when the target variable of interest represents the effect of the predictive covariates, *e.g.*, the number of bike riding is decided by temperature, weather, *etc*.

When $Y$ does point to $\mathbf{K}$, *e.g.*, $Y$ represents the disease and $\mathbf{K}$ represents its symptoms or biomarkers, Alg. 1 needs to search among $g$-equivalent classes and the complexity is $O(N_G)$. We will give some

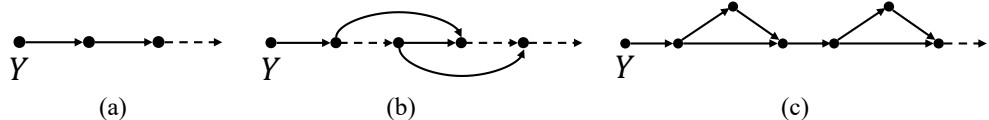

Figure 3: Examples of $G_{\overline{\mathbf{X}}_M}$: (a) chain (b) skip-chain (c) knot.

examples to show that $N_G$ can be polynomial w.r.t. $d_S$ in some cases while can also exponentially increase w.r.t. $d_S$, depending on the number of edges in the graph. Before this, we first show (Lemma. D.1 in appendix) that $N_G$ will not decrease (increase) if we add (delete) edges to (from) $G$.

**Claim 3.6.** For chain, we have $N_G = O(d_S)$; for the skip-chain, we have $N_G = O(d_S^{2k})$, where $k$ is the number of added edges; for the knot graph, we have $N_G = O(c^{d_S})$ for some $1 < c < 2$.

We first consider the chain graph, *i.e.*, $Y \to X_{S,(1)} \to ... \to X_{S,(d_S)}$ (where $(1),...,(d_S)$ is a permutation of $1,...,d_S$ according to the generating order in the chain) in Fig. 3 (a), in which we find $N_G$ is polynomial as shown in Claim 3.6. This is simply because blocking $X_{S,(i)}$ will $d$-separate $Y$ and $X_{S,(j)}$ for any $j > i$, making $\{X_{S,(i)}\}$ g-equivalent to $\{\{X_{S,(i)}, X_{S,(j_1)}, ..., X_{S,(j_k)}\} : i < j_1 \leq ... \leq j_k$ for any $k\}$. Next, we consider two cases of adding $k$ edges to the chain: **i)** the skip-chain graph (Fig. 3 (b)) where $k$ does not increasing with $d_S$. In this case, $N_G$ is still polynomial since the number of paths between $Y$ and any $X_{S,(i)}$ can be bounded; **ii)** the knot graph (Fig. 3 (c)) where $k$ increases with $d_S$, in this case, $N_G$ is shown to exponentially increase w.r.t. $d_S$, because the number of paths between $Y$ and $X_{S,(i)}$ can be exponentially large. Proof of Claim 3.6 and more examples are left to Sec. C.1 in the appendix.

### 3.4 SPARSE MIN-MAX OPTIMIZATION

According to the previous discussion, the overall searching complexity can still be exponentially large. To improve efficiency, we provide an alternative method that turns the subset selection problem of Eq. 1 into the following sparse min-max optimization scheme:

$$\min_{\alpha,\beta} \max_{\theta} \mathbb{E}_{\bar{p}(\boldsymbol{x},y|\boldsymbol{x}_M=J_\theta(pa(\boldsymbol{x}_M)))} \left[ \left(y - f_\alpha(\boldsymbol{x}_S\beta, \boldsymbol{x}_M)\right)^2 \right] + \lambda\|\beta\|_1, \qquad (3)$$

where we introduce the coefficient vector $\beta$ and implement a lasso-type penalty on $\beta$ with hyperparameter $\lambda > 0$. This penalty regularizes $\beta$ to be sparse and its support set, *i.e.*, $\text{supp}(\beta) := \{i : \beta(i) \neq 0\}$ is used to select the optimal subset. To optimize, we alternatively take a gradient ascent with respect to $\theta$, followed by the minimization over $(\alpha, \beta)$. Note that under irrepresentable and restricted convexity conditions Zhao & Yu (2006), we have model selection consistency, *i.e.*, the true support set of $\beta$ can be recovered and $\ell_2$-consistency properties when $d_S$ is fixed, according to Rejchel (2016). When $d_S$ increases with $n$, under restricted convexity conditions, we showed that this lasso-type estimator that belongs to a broader family of $M$-estimators Negahban et al. (2012), are $\ell_2$-consistent. To further reduce the complexity in the minimization step, we propose to implement *Linearized Bregman Iteration* (LBI) via differential inclusion, which enjoys the efficiency of generating a whole regularization solution path. In each iteration, the original minimization step can be replaced by a gradient descent step followed by a soft thresholding step. Details are left to Sec. E in the appendix.

## 4 EXPERIMENT

In this section, we evaluate our method on synthetic data and two real-world applications: diagnosis of Alzheimer's Disease which is one of the most common types of dementia among elder people, and gene function prediction that can potentially help understand the human-disease progress Muñoz-Fuentes et al. (2018).

**Compared Baselines.** We compare our methods with the following baselines: **i)** ICP (Peters et al., 2016; Bühlmann, 2020) that assumed invariance of $P(Y|\text{Pa}(Y))$; **ii)** IC (Rojas-Carulla et al., 2018) that extended the above assumption to features beyond $\text{Pa}(Y)$; **iii)** Anchor regression (Rothenhäusler et al., 2021) that interpolated between ordinary least square (LS) and causal minimax LS, and constrained the residue in the anchor subspace to be small; **iv)** IRM (Arjovsky et al., 2019) that

learned an invariant representation to transfer; **v)** HRM (Liu et al., 2021) that extended IRM to the case with unknown domain labels, by exploring the heterogeneity in data via clustering; **vi)** IB-IRM (Ahuja et al., 2021) that leveraged the information bottleneck regularization to supplement the invariance principle in IRM; and **vii)** Graph Surgery estimator Subbaswamy et al. (2019) that used validation's loss to identify the optimal subset.

**Implementation Details.** We leave implementation details to Sec. F in the appendix.

## 4.1 SIMULATION STUDY

**Data Generation.** We follow Fig. 4 to generate $\mathbf{X}_S := \{X_1, X_2, X_3\}$ and $\mathbf{X}_M := \{X_4\}$ via the following structural equations: $x_3 \leftarrow u_3$ with $u_3 \sim \mathrm{N}(-2,1)$; $x_2 \leftarrow g_2(x_3) + u_2$ with $u_2 \sim \mathrm{N}(0,1)$; $y \leftarrow g_y(x_2) + u_y$ with $u_y \sim \mathrm{N}(0,1)$; $x_4 \leftarrow \beta_e g_4(y) + u_4$ with $\beta_e = e - 5$ varied in different domains, $u_4 \sim \mathrm{N}(0,1)$; $x_1 \leftarrow g_1(x_4, y) + u_1$ with $u_1 \sim \mathrm{N}(0, 0.2)$. We consider three settings: **i)** $g_2(x_3) = 0.5x_3$, $g_y(x_2) = -1.5x_2$, $g_4(y) = y$, and $g_1(x_4, y) = x_4$; **ii)** $g_1(\cdot)$ is changed to $g_1(x_4, y) = x_4 + 2.5y$; **iii)** $g_2(x_3) = 10\mathrm{sinc}(x_3)$, $g_y(x_2) = 2\tanh(x_2)$, $g_4(y) = -0.25y^3 + y$, and $g_1(x_4, y) = \mathrm{Sigmoid}(x_4 + y)$. For each setting, we generate 10 environments with $e = 1, ..., 10$, where $n_e = 200$ for each environment. To remove the effect of randomness, we repeat 10 times, and each time we randomly select five domains for training and the rest for testing.

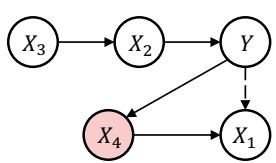

Figure 4: DAG with $\mathbf{X}_M := \{X_4\}$. Dotted arrow exists in setting-2,3.

**Causal Discovery and Complexity Analysis.** We use $F_1$ score, precision, and recall in terms of directed edges, to assess our causal discovery algorithm. We repeat 10 times and the average results are $F_1 = 0.99$, precision $= 1.00$, and recall $= 0.98$, which suggests the validity of our algorithm. Validations on larger graphs are left in the appendix. As for complexity, in setting-1, the condition in Thm. 3.1 holds and we expect $\{X_1, X_2, X_3\}$ to be the optimal set. In setting-2,3, the condition is violated and we need to compare $h^*$ of each equivalent class to find the optimal subset. There are seven equivalent classes, as the only equivalent relation is $\{X_2\} \sim_G \{X_2, X_3\}$.

Table 1: Mean Squared Error (MSE) on simulation data.

| Predictor | Setting-1 | | Setting-2 | | Setting-3 | |
|---|---|---|---|---|---|---|
| | $h^*(S_-)$ | max MSE | $h^*(S_-)$ | max MSE | $h^*(S_-)$ | max MSE |
| Vanilla[1] | - | $1.90_{\pm 0.58}$ | - | $.07_{\pm .00}$ | - | $1.72_{\pm 0.72}$ |
| $\varnothing$ | $5.24_{\pm 2.83}$ | $4.61_{\pm .25}$ | $5.97_{\pm 2.73}$ | $4.12_{\pm .25}$ | $4.27_{\pm .51}$ | $3.75_{\pm .18}$ |
| $\{X_1\}$ | $5.20_{\pm 2.76}$ | $4.61_{\pm .25}$ | $\mathbf{.0003}_{\pm .00}$ | $\mathbf{.0075}_{\pm .00}$ | $6.84_{\pm .52}$ | $4.42_{\pm 1.15}$ |
| $\{X_2\}$ | $1.08_{\pm .03}$ | $1.21_{\pm .06}$ | $1.37_{\pm .57}$ | $1.13_{\pm .07}$ | $\mathbf{1.16}_{\pm .18}$ | $\mathbf{1.10}_{\pm .05}$ |
| $\{X_3\}$ | $3.56_{\pm .66}$ | $4.01_{\pm .13}$ | $5.26_{\pm 2.51}$ | $3.51_{\pm .28}$ | $3.10_{\pm .30}$ | $3.11_{\pm .11}$ |
| $\{X_{1,2}\}$ | $1.08_{\pm .03}$ | $1.21_{\pm .06}$ | $.06_{\pm .00}$ | $.07_{\pm .00}$ | $1.51_{\pm 1.02}$ | $1.20_{\pm .13}$ |
| $\{X_{1,3}\}$ | $3.55_{\pm .65}$ | $4.00_{\pm .13}$ | $.01_{\pm .00}$ | $.02_{\pm .00}$ | $6.86_{\pm .55}$ | $4.39_{\pm 1.15}$ |
| $\{X_{2,3}\}$ | $1.06_{\pm .03}$ | $1.19_{\pm .06}$ | $1.23_{\pm .26}$ | $1.14_{\pm .08}$ | $1.17_{\pm .20}$ | $1.10_{\pm .05}$ |
| $\{X_{1,2,3}\}$ | $\mathbf{1.06}_{\pm .03}$ | $\mathbf{1.18}_{\pm .06}$ | $.06_{\pm .00}$ | $.07_{\pm .00}$ | $1.17_{\pm .01}$ | $1.21_{\pm .13}$ |

[1] Results of other baselines are left in the appendix.

**Results Analysis.** In Tab. 1, we report the estimated $h^*$ and the maximal mean squared error (MSE) over test sets of each subset $S_-$ and the vanilla regression method. As shown, in setting-1, the whole stable set enjoys the minimal max MSE, which agrees with Thm. 3.1; while in setting-2,3, the subset ($\{X_1\}$ in setting-2, $\{X_2\}$ in setting-3) identified by minimal $h^*$ has minimal max MSE. This suggests the effectiveness of our method in finding the optimal subset. Besides, we observe that $h^*$ can estimate the maximal MSE well in most cases, *e.g.*, $h^*(\{X_1, X_2\}) = 0.06$ vs 0.07 of max MSE in setting-2, $h^*(\{X_3\}) = 3.10$ vs 3.11 of max MSE in setting-3. In addition, equivalent subsets have similar performance, *e.g.*, max MSE of $\{X_2\}$ and $\{X_2, X_3\}$ in setting-3 are both 1.10, which verifies the searching can be conducted only over equivalent space.

## 4.2 REAL-WORLD APPLICATIONS

**Datasets.** We consider the Alzheimer's Disease Neuroimaging Initiative (ADNI) Petersen et al. (2010) dataset for Alzheimer's Disease diagnosis, and the International Mouse Phenotyping Consortium (IMPC) CRM workshop (2016) dataset for gene function prediction.

- *ADNI*. The dataset includes $n = 757$ patients enrolled in ADNI-GO/1/2 periods. We apply the Automatic Anatomical Labeling atlas (Tzourio-Mazoyer et al., 2002) and region index Young et al. (2018) to partition the whole brain into 9 brain regions: frontal lobe (FL), medial temporal lobe (MTL), parietal lobe (PL), occipital lobe (OL), cingulum (CIN), insula (INS), amygdala (AMY), hippocampus (HP), and pallidum (PL). In addition to brain region volumes, we also include gender (GED) and genetic information (number of ApoE-4 alleles (ApE)). With these covariates, we predict the Functional Activities Questionnaire (FAQ) score $Y$ for each patient. We split the dataset into seven environments according to age (age <60, 60-65, 65-70, 70-75, 75-80, 80-85, >85), which respectively contains $n_e = 27, 59, 90, 240, 182, 117, 42$ samples. We repeat 15 times, with each time four domains are randomly selected for training and the rest for testing.
- *IMPC*. The dataset contains the hematology phenotype of both wild-type and mutant mice with 13 kinds of single-gene knockout. To predict the function of the target gene, we knock it out and assess the cell counts of monocyte (MON), with cell counts of neutrophil (NEU), lymphocyte (LYM), eosinophil (EO), basophil (BA), and large unstained cell (LUC) as covariates. Each environment corresponds to a kind of gene-knockout. We repeat 45 times: each time five randomly picked gene knockouts and the wild-type are selected as training sets and the rest eight kinds are left for testing.

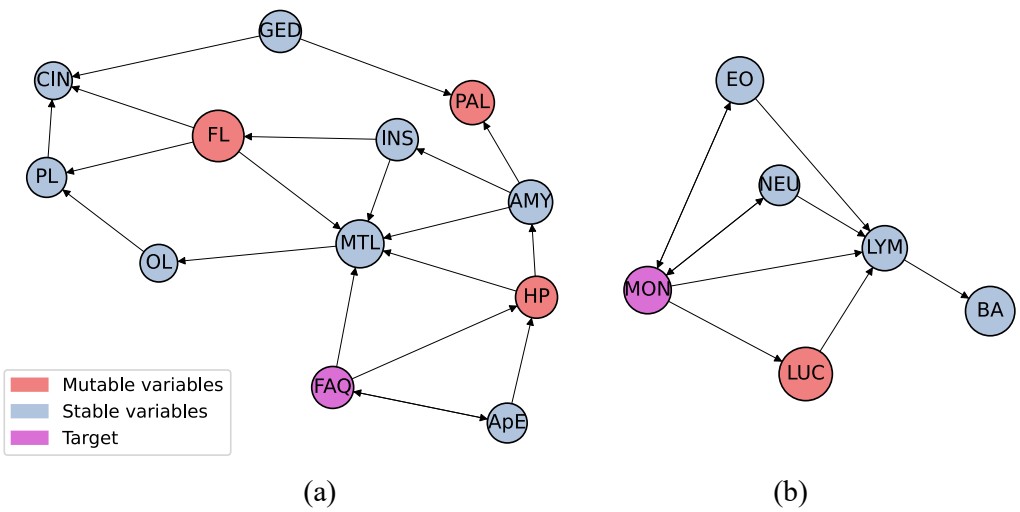

(a) (b)

Figure 5: Learned causal graphs on (a) ADNI and (b) IMPC. $\leftrightarrow$ to denote undirected edges.

**Causal Discovery.** We implement our causal discovery algorithm in Sec. 3.2 to learn PDAG respectively shown in Fig. 5 (a,b). For ADNI, Fig. 5 (a) shows that the affection of AD, measured by the FAQ score, firstly shows in the medial temporal lobe (MTL) and the hippocampus (HP), then propagates to other brain regions, which echos existing studies that MTL and HP are early degenerated regions (Barnes et al., 2009; Duara et al., 2008). Besides, we observe that the frontal lobe (FL), pallidum (PAL), and hippocampus (HP) are mutable regions, which agrees with heterogeneity across different age groups found in existing studies Cavedo et al. (2014); Fiford et al. (2018). For IMPC, Fig. 5 (b) shows that the monocyte (MON) affects the number of lymphocytes (LYM) and large unstained cells (LUC), which reflects the activation mechanisms of LYM (Carr et al., 1994) and LUC (Lee et al., 2021). It also plays a role in the activation of basophil (BA) through the causal chain MON $\rightarrow$ LYM $\rightarrow$ BA, as also found in existing study (Goetzl et al., 1984).

**Complexity Analysis.** On both ADNI and IMPC, the condition in Thm. 3.1 is violated, as $Y$ (FAQ on ADNI, MON on IMPC) points to $\mathbf{K}$ (MTL on ADNI, LYM on IMPC). So, we need to search over $g$-equivalent classes and compare their $h^*$, as suggested by Thm. 3.3. The numbers of $g$-equivalent classes are 98 (out of the $2^8 = 256$ subsets) on ADNI and 12 (out of the $2^4 = 16$ subsets) on IMPC.

**Results Analysis.** Fig. 1 (a) and Fig. 6 (a) report maximal MSE of our method and baselines. As we can see, our methods significantly outperform the others (7.8% on ADNI, 9.7% on IMPC). Besides, our sparse optimization is comparable to the searching method in Alg. 1. These results demonstrate the utility of Thm. 3.3 in identifying the optimal subset, as well as the effectiveness of

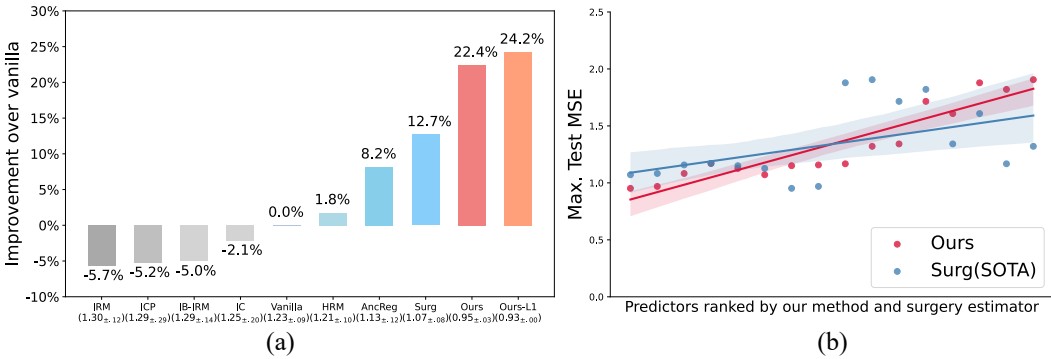

Figure 6: Results on IMPC. (a) Maximal MSE over test environments. (b) Maximal MSE of predictors that are ranked in ascending order from left to right, respectively according to $h^*$ of our method and the validation's loss of the graph surgery estimator.

sparse optimization when the searching cost is expensive. As for advantages over the graph surgery estimator, Fig. 1 (b) and Fig. 6 (b) show that our $h^*$ well reflects the worst-case risk, as it increases with the worst-case risk; while there is no such property for the validation's loss in the surgery estimator. Particularly, the top subsets ranked by our $h^*$ also have the minimal max MSE; while the one selected fails to identify $S^*$. The improvements over ICP, IC, IRM, and their extensions are due to our utilization of stable information beyond causal features/representations. The advantage over Anchor regression may be due to the relaxation of the linearity assumption.

In addition, Fig. 7 shows that subsets in the same equivalent class have similar maximal MSE, which further shows the validity of searching over only $g$-equivalent classes in Alg. 1.

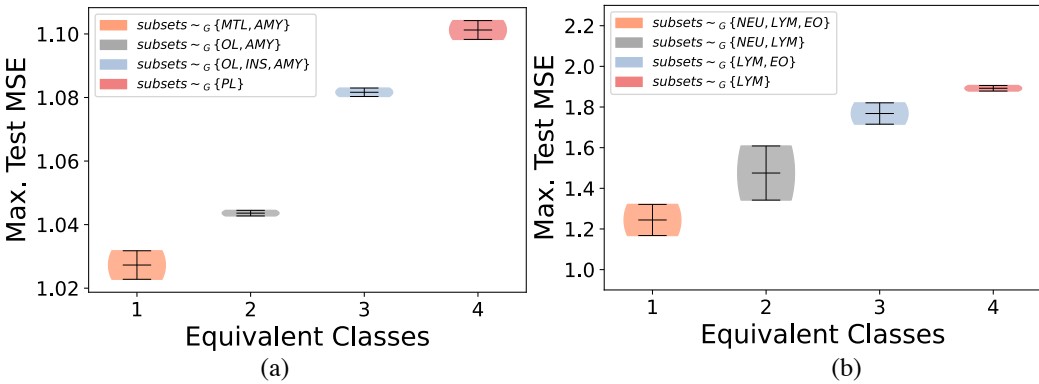

Figure 7: Max MSE of predictors in the same equivalent class on (a) ADNI and (b) IMPC datasets.

## 5  CONCLUSION

In this paper, we propose a minimax learning approach to identify the optimal subset of invariance to transfer, in order to achieve robustness against dataset shift. Among all subsets of stable information, we provide a sufficient and necessary condition for a subset to be min-max optimal. We analyze the searching complexity by introducing the notion of graphical equivalence and propose a sparse min-max optimization algorithm when the searching cost is expensive. The subset identified by our method outperforms others in terms of robustness, on Alzheimer's Disease diagnosis and gene function prediction tasks. In the future, we are interested to study the scenarios where the DAG is also allowed to vary across domains, which may happen when the number of environments is large.

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

## REPRODUCIBILITY STATEMENT

Data, code, and instructions to reproduce the main experimental results are provided. Specifically, the ADNI data set is available at `http://adni.loni.ucla.edu`), the IMPC data set is avaiable at `http://www.crm.umontreal.ca/2016/Genetics16/competition_e.php`, the codes are provided in the supplementary materials, the implementation instructions are provided in Sect. F in the appendix.

## APPENDIX

## A  RELATED WORKS

**Causal learning for domain generalization.** There have been emerging works that consider the domain generalization problem from a causal perspective. One line of work Arjovsky et al. (2019); Xie et al. (2020); Müller et al. (2020) promoted invariance as a key surrogate feature of causation where the causal graph is more of a motivation. Another line of work Ilse et al. (2020); Lu et al. (2021); Mahajan et al. (2020); Mitrovic et al. (2021) considered domain generalization for unstructured data using specifically designed causal graphs to incorporate priors of the distribution shift, in which the causal features are modeled as latent variables to be inferred for robust prediction. The works most relevant to us pursued robust optimization by making invariance assumptions regarding causal mechanisms Subbaswamy et al. (2019); Bühlmann (2020); Peters et al. (2016); Subbaswamy & Saria (2020). Specifically, the Peters et al. (2016) assumed the generation of $Y$ from its parents is invariant; hence they only utilize $Y$'s parents for transfer. The Subbaswamy et al. (2019); Subbaswamy & Saria (2020) considered a selection diagram framework, in which mutable variables are children of the selection variable. They then remove the unstable mechanism by intervening of $\mathbf{X}_M$ and obtain a set of stable covariates $S$. To obtain the optimal subset, they simply search over all subsets in $S$ and took the one with the minimal validation loss. However, their method is theoretically incomplete and practically defective, as the selected subset may not be min-max optimal and the searching cost is expensive in large-scale graphs. **In contrast**, we provide a comprehensive min-max analysis to guarantee the identification of the optimal subset. For practical employment, we analyze the searching complexity via the lens of $g$-equivalence. For those graph with expensive searching cost, we provide a sparse min-max optimization scheme that can larger improve the efficiency.

**Optimization-based domain generalization.** There are emerging works that view the domain generalization problem as an optimization problem. These methods directly formulate the objective of out-of-distribution generalization and conduct optimization for min-max optimum. For example, Distributional Robust Optimization (DRO) Duchi & Namkoong (2021) constrained the distance between test and training distribution with f-divergence or Wasserstein distance and optimized over the min-max objective. One of its popular extensions, GroupDRO Sagawa et al. (2019), provided extra regularization (*e.g.*, weight-decay or early stop) and allowed DRO models to achieve better performances in large neural networks.

However, these methods heavily rely on data-driven optimization and lack analysis of the source of distributional shifts. For this reason, they have to constrain the distributional shifts to a limited extent, so as to achieve optimization convenience. Such a limitation affects their ability to generalize to a broader distribution family, thus limiting their applications in the real-world. **In contrast**, we consider domain generalization from a causal perspective. Benefiting from the causal framework for distributional shifts, our method can identify the reasons behind distributional shifts and achieve min-max optimum even when the distribution can vary arbitrarily.

**Causal discovery in heterogeneous data.** Our work also benefits from the recent progress in causal discovery Mooij et al. (2020); Huang et al. (2020); Huang & Zhang (2019); Ghassami et al. (2018), a field that focuses on identifying the causal relations from data. Our work shares a similar framework with Huang et al. (2020) in formulating the distribution shift. However, they focused on recovering the full causal graph to study relations among variables, we provide a local discovery procedure, which aids the analysis of min-max properties and identification of robust predictors.

## B  APPENDIX FOR SEC. 3.1: IDENTIFICATION WITH MIN-MAX PROPERTY

### B.1  PROOF FOR THM. 3.1: GRAPHICAL CRITERION FOR $f^* = f_S$

**Theorem 3.1** (Graphical Criterion for $f^* = f_S$). *Suppose assumptions 2.1,2.2 hold. Denote* $\mathbf{X}_M^0 := \mathbf{X}_M \cap \mathrm{Ch}(Y)$ *as mutable variables in $Y$'s children, and* $\mathbf{K} := \mathrm{De}(\mathbf{X}_M^0) \backslash \mathbf{X}_M^0$ *as descendants of* $\mathbf{X}_M^0$. *Then,* $p(y|\boldsymbol{x}_S, do(\boldsymbol{x}_M))$ *can degenerate to conditional distribution if and only if $Y$ does not point to any member of* $\mathbf{K}$. *Further, under either of the two conditions, we have $f^* = f_S$.*

*Proof.* Denote the causal DAG as $G$, the intervened graph that removes all arrowheads into $\mathbf{V}$ as $G_{\overline{\mathbf{V}}}$. Define $\mathbf{X}_M^1 := \mathbf{X}_M \backslash \mathbf{X}_M^0$, $\mathbf{K}_2 := (\mathbf{X} \backslash \mathbf{X}_M^0) \backslash \mathrm{De}(\mathbf{X}_M^0)$.

**We firstly prove the equivalence of the following conditions (1), (2), and (3):**

1. $Y \perp_{G_{\overline{\mathbf{X}_M^0}}} \mathbf{K}|\mathbf{K}_2$;

2. $Y$ and $\mathbf{K}$ are not adjacent in $G$ [1];

3. $p(y|\boldsymbol{x}_S, do(\boldsymbol{x}_M))$ can degenerate to conditional distribution.

**(1)→(2)**   If $Y$ and $\mathbf{K}$ are adjacent, they are also adjacent in $G_{\overline{\mathbf{X}_M^0}}$ because $\mathbf{K} \cap \mathbf{X}_M^0 = \varnothing$, so $Y$ and $\mathbf{K}$ can not be d-separated by any variable in $G_{\overline{\mathbf{X}_M^0}}$, which contradicts with (1).

**(2)→(3)**   Define

$$I := p(y|\boldsymbol{k}, \boldsymbol{k_2}, do(\boldsymbol{x_M^0})) = \frac{p(y|pa(y)) \prod_{X_j \in \mathbf{K}} p(x_j|pa(x_j)) \prod_{X_i \in \mathbf{K}_2} p(x_i|pa(x_i))}{\int p(y|pa(y)) \prod_{X_j \in \mathbf{K}} p(x_j|pa(x_j)) \prod_{X_i \in \mathbf{K}_2} p(x_i|pa(x_i))dy}.$$

Since $\mathrm{PA}(Y) \cap \{\mathbf{X}_M^0, \mathbf{K}\} = \varnothing$ and $\forall X_i \in \mathbf{K}_2, \mathrm{PA}(X_i) \cap \{\mathbf{X}_M^0, \mathbf{K}\} = \varnothing$, we have:

$$I = \frac{p(y, \boldsymbol{k}_2) \prod_{X_j \in \mathbf{K}} p(x_j|pa(x_j))}{\int p(y, \boldsymbol{k}_2) \prod_{X_j \in \mathbf{K}} p(x_j|pa(x_j))dy}.$$

If $Y$ and $\mathbf{K}$ are not adjacent, then $\forall X_j \in \mathbf{K}, Y \notin \mathrm{PA}(X_j)$. Therefore, $I = \frac{p(y, \boldsymbol{k}_2)}{\int p(y, \boldsymbol{k}_2)dy} = p(y|\boldsymbol{k}_2)$.

**(3)→(1)**   We will prove by contradiction. Specifically, we will show that if $Y \not\perp_{G_{\overline{\mathbf{x}_M^0}}} \mathbf{K}|\mathbf{K}_2$, *i.e.*, (1) does not hold, then $p^e(y|\boldsymbol{x}_S, do(\boldsymbol{x}_M))$ can not degenerate to any conditional distribution, *i.e.*, (3) does not hold.

We firstly show $Y \not\perp_{G_{\overline{\mathbf{x}_M^0}}} \mathbf{K}|\mathbf{K}_2 \Rightarrow p^e(y|\boldsymbol{x}_S, do(\boldsymbol{x}_M)) \neq p^e(y|\boldsymbol{k}_2, do(\boldsymbol{x}_M^0))$, then show $p^e(y|\boldsymbol{x}_S, do(\boldsymbol{x}_M)) \neq p^e(y|\boldsymbol{k}_2, do(\boldsymbol{x}_M^0)) \Rightarrow p^e(y|\boldsymbol{x}_S, do(\boldsymbol{x}_M))$ can not degenerate to any conditional distribution.

Since $Y \notin \mathrm{PA}(\mathbf{X}_M^1)$, we have:

$$
\begin{aligned}
p^e(y|\boldsymbol{x}_S, \boldsymbol{x}_M^1, do(\boldsymbol{x}_M^0)) &= \frac{p^e(y, \boldsymbol{x}_S, \boldsymbol{x}_M^1 | do(\boldsymbol{x}_M^0))}{\int p^e(y, \boldsymbol{x}_S, \boldsymbol{x}_M^1 | do(\boldsymbol{x}_M^0))dy} \\
&= \frac{p^e(y|pa(y)) \prod_{i \in S} p^e(x_i|pa(x_i)) \prod_{X_i \in \mathbf{X}_M^1} p^e(x_i|pa(x_i))}{\int p^e(y|pa(y)) \prod_{i \in S} p^e(x_i|pa(x_i)) \prod_{X_i \in \mathbf{X}_M^1} p^e(x_i|pa(x_i))dy} \\
&= \frac{p^e(y|pa(y)) \prod_{i \in S} p^e(x_i|pa(x_i))}{\int p^e(y|pa(y)) \prod_{i \in S} p^e(x_i|pa(x_i))dy} \\
&= p^e(y|\boldsymbol{x}_S, do(\boldsymbol{x}_M))
\end{aligned}
$$

Since $\mathbf{K} \cup \mathbf{K}_2 = \mathbf{X}_S \cup \mathbf{X}_M^1$, we have $p^e(y|\boldsymbol{x}_S, do(\boldsymbol{x}_M)) = p^e(y|\boldsymbol{k}, \boldsymbol{k}_2, do(\boldsymbol{x}_M^0))$. Thus, we can prove: $Y \not\perp_{G_{\overline{\mathbf{x}_M^0}}} \mathbf{K}|\mathbf{K}_2 \Rightarrow p^e(y|\boldsymbol{x}_S, do(\boldsymbol{x}_M)) = p^e(y|\boldsymbol{k}, \boldsymbol{k}_2, do(\boldsymbol{x}_M^0)) \neq p^e(y|\boldsymbol{k}_2, do(\boldsymbol{x}_M^0))$.

Next, we prove $p^e(y|\boldsymbol{x}_S, do(\boldsymbol{x}_M)) \neq p^e(y|\boldsymbol{k}_2, do(\boldsymbol{x}_M^0)) \Rightarrow p^e(y|\boldsymbol{x}_S, do(\boldsymbol{x}_M))$ can not degenerate to any conditional distribution.

Suppose $p^e(y|\boldsymbol{x}_S, do(\boldsymbol{x}_M)) = p^e(y|\boldsymbol{k}', \boldsymbol{k}_2, do(\boldsymbol{x}_M^0))$. We will show if $\boldsymbol{k}' \neq \varnothing$, then the *do*-operator can not be removed with either Rule 2 (action to observation) or Rule 3 (deletion of action). To express $do(\boldsymbol{x}_M^0)$ explicitly, denote $\mathbf{X}_M^0 = \{X_{M,i}^0\}_{i=1}^r$ and $p^e(y|\boldsymbol{k}', \boldsymbol{k}_2, do(\boldsymbol{x}_M^0)) = p^e(y|\boldsymbol{k}', \boldsymbol{k}_2, do(x_{M,1}^0), \ldots, do(x_{M,r}^0))$.

- Rule 2 can not remove the *do*-operator of any $X_{M,i}^0 \in \mathbf{X}_M^0$.

---

[1] Note the edge between $Y$ and $X \in \mathbf{K}$ can only be $Y \to X$.

Recall Rule 2 states that "$p(\boldsymbol{y}|do(\boldsymbol{x}), do(\boldsymbol{z}), \boldsymbol{w}) = p(\boldsymbol{y}|do(\boldsymbol{x}), \boldsymbol{z}, \boldsymbol{w})$ if $\mathbf{Y} \perp_{G_{\overline{\mathbf{X}}\underline{\mathbf{Z}}}} \mathbf{Z}|\mathbf{X}, \mathbf{W}$ for any disjoint subsets of variables $\mathbf{X}, \mathbf{Y}, \mathbf{Z}$, and $\mathbf{W}$ ".

If Rule 2 can remove the *do*-operator of $X^0_{M,i} \in \mathbf{X}^0_M$, then

$$Y \perp_{G_{\overline{\mathbf{X}^0_M \setminus \{X^0_{M,i}\}}\underline{X^0_{M,i}}}} X^0_{M,i}|\mathbf{K}', \mathbf{K}_2, \mathbf{X}^0_M \setminus \{X^0_{M,i}\}. \tag{4}$$

As we have $\mathbf{Z} = \{X^0_{M,i}\}, \mathbf{X} = \mathbf{X}^0_M \setminus X^0_{M,i}, \mathbf{W} = \mathbf{K}' \cup \mathbf{K}_2$ in the notations of Rule 2.

In the following, we explain why Eq. 4 can not be true. Note $X^0_{M,i} \in \mathrm{Ch}(Y)$ and the direct edge $Y \to X^0_{M,i}$ is reserved in the intervend graph $G_{\overline{\mathbf{X}^0_M \setminus \{X^0_{M,i}\}}\underline{X^0_{M,i}}}$, which means that $Y$ and $X^0_{M,i}$ can not be d-separated by any set of variables in the intervened graph. Thus, Eq. 4 can not be true.

- Rule 3 can not remove the *do*-operator of all $X^0_{M,i} \in \mathbf{X}^0_M$.

Recall Rule 3 states that "$p(\boldsymbol{y}|do(\boldsymbol{x}), do(\boldsymbol{z}), \boldsymbol{w}) = p(\boldsymbol{y}|do(\boldsymbol{x}), \boldsymbol{w})$ if $\mathbf{Y} \perp_{G_{\overline{\mathbf{X}, \overline{\mathbf{Z}(\mathbf{W})}}}} \mathbf{Z}|\mathbf{X}, \mathbf{W}$[2] for any disjoint subsets of variables $\mathbf{X}, \mathbf{Y}, \mathbf{Z}$, and $\mathbf{W}$ ". If Rule 3 can remove the *do*-operator of $\mathbf{X}^0_M$, then:

$$Y \perp_{G_{\overline{\mathbf{X}^0_M(\mathbf{K}' \cup \mathbf{K}_2)}}} \mathbf{X}^0_M|\mathbf{K}' \cup \mathbf{K}_2 \tag{5}$$

because the notations in Rule 3 mean $\mathbf{X} = \varnothing, \mathbf{Z} = \mathbf{X}^0_M, \mathbf{W} = \mathbf{K}' \cup \mathbf{K}_2$. In the following, we will show that when $\mathbf{K}' \neq \varnothing$, Eq. 5 can not hold. When $\mathbf{K}' \neq \varnothing$, note by definition $\mathbf{K}' \subset \mathrm{De}(\mathbf{X}^0_M)$, so $\mathrm{An}(\mathbf{K}') \cap \mathbf{X}^0_M \neq \varnothing$. Therefore, $\mathbf{X}^0_M(\mathbf{K}' \cup \mathbf{K}_2) = \mathbf{X}^0_M \setminus \{\mathrm{An}(\mathbf{K}') \cup \mathbf{K}_2\} \neq \mathbf{X}^0_M$. That is $\mathbf{X}^0_M \setminus \mathbf{X}^0_M(\mathbf{K}' \cup \mathbf{K}_2) \neq \varnothing$. Suppose $X^0_{M,i} \in \mathbf{X}^0_M \setminus \mathbf{X}^0_M(\mathbf{K}' \cup \mathbf{K}_2)$, then the edge $Y \to X^0_{M,i}$ is in the intervened graph $G_{\overline{\mathbf{X}^0_M(\mathbf{K}' \cup \mathbf{K}_2)}}$, so $Y$ and $X^0_{M,i}$ can not be d-separated by any variable set. So Eq. 5 does not hold.

In summary, we have proved that when $\mathbf{K}' \neq \varnothing$, the *do*-operator on $\mathbf{X}^0_M$ can not be removed entirely by Rule 2 and 3 .

Besides, according to Corollary 3.4.2 in Pearl (2009), the inference rules are complete in the sense that if the intervention probability (with *do* ) can be reduced to a probability expression (without *do* ), the "reduction" can be realized by a sequence of transformations, each conforming to one of the Inference Rules 1-3. Note that only Rule 2 and 3 are related to the disappearance of *do*-operator, so it is sufficient to prove that Rule 2 and 3 can not remove the *do*-operator on $\mathbf{X}^0_M$.

**We then prove under either of conditions (1), (2), or (3), $f^* = f_S$.**

Given any one of the three conditions (1), (2), or (3), $f^*(\boldsymbol{x}) = \mathrm{E}_{P^e}[Y|\boldsymbol{x}_S, do(\boldsymbol{x}_M)]$ satisfies the following min-max property:

$$f^*(\boldsymbol{x}) = \mathrm{argmin}_{f:\mathcal{X} \mapsto \mathcal{Y}} \max_{P \in \mathcal{P}} \mathrm{E}_P[Y - f(\boldsymbol{x})]^2.$$

Under any one of the conditions (1)-(3), we have $p^e(y|\boldsymbol{x}_S, do(\boldsymbol{x}_M)) = p^e(y|\boldsymbol{k}_2)$ for $P^e \in \mathcal{P}$.

For $P^e \in \mathcal{P}$, let $p^e(\boldsymbol{x}^0_M) = \sum_{X_i \in V \setminus \mathbf{X}^0_M} p^e(\boldsymbol{v})$ be the marginal distribution of $\mathbf{X}^0_M$. Define $\widetilde{P^e}$ as:

$$\widetilde{p}^e(\boldsymbol{v}) = p(y|pa(y)) \prod_{X_i \in \mathbf{K}} p^e(x_i|pa(x_i)) \prod_{X_i \in \mathbf{K}_2} p^e(x_i|pa(x_i)) p^e(\boldsymbol{x}^0_M),$$

by replacing the term $\prod_{X_i \in \mathbf{X}^0_M} p(x_i|pa(x_i))$ in $p^e(\boldsymbol{v})$ with $p^e(\boldsymbol{x}^0_M)$.

(i) By the definition of $\mathcal{P}, \widetilde{P^e} \in \mathcal{P}$ and $\widetilde{p}^e(y|\boldsymbol{x}) = \widetilde{p}^e(y|\boldsymbol{x}_S, \boldsymbol{x}^1_M, \boldsymbol{x}^0_M) = \widetilde{p}^e(y|\boldsymbol{x}_S, \boldsymbol{x}^1_M, do(\boldsymbol{x}^0_M)) = \widetilde{p}^e(y|\boldsymbol{k}_2)$

(ii) In the following, we will show $\widetilde{p}^e(y, \boldsymbol{k}_2) = p^e(y, \boldsymbol{k}_2)$.

---

[2]$\mathbf{Z}(\mathbf{W})$ is the set of $Z$-nodes that are not ancestors of any $W$-node in $G_{\overline{\mathbf{X}}}$.

First, note that $\mathbf{K} \subset \mathrm{De}\left(\mathbf{X}_M^0\right)$ and $\mathbf{X}_M^0 \subset \mathrm{Ch}(Y)$, we have $\mathbf{K} \cup \mathbf{X}_M^0 \subset \mathrm{De}(Y)$. Thus, $\mathrm{PA}(Y) \cap \left\{\mathbf{K} \cup \mathbf{X}_M^0\right\} = \varnothing$ because otherwise there would be a cycle.

Second, $\mathrm{PA}(\mathbf{K}_2) \cap \left\{\mathbf{K} \cup \mathbf{X}_M^0\right\} = \varnothing$ because if there exist $X_i \in \mathbf{K} \cup \mathbf{X}_M^0$ and also $X_i \in \mathrm{PA}(\mathbf{K}_2)$, then $\mathbf{K}_2 \cap \mathrm{De}\left(\mathbf{X}_M^0\right) \neq \varnothing$, which contradicts with the definition that $\mathbf{K}_2 := \left(\mathbf{X} \backslash \mathbf{X}_M^0\right) \backslash \mathrm{De}\left(\mathbf{X}_M^0\right)$.

In summary, we have $\mathrm{PA}(\mathbf{K}_2 \cup Y) \cap \left\{\mathbf{K} \cup \mathbf{X}_M^0\right\} = \varnothing$, which leads to

$$p^e\left(\boldsymbol{k}_2, y\right) = \int p(y|pa(y)) \Pi_{X_i \in \mathbf{K}_2} p^e\left(x_i|pa(x_i)\right) \Pi_{X_i \in \mathbf{K}} p^e\left(x_i|pa(x_i)\right) \Pi_{X_i \in \mathbf{X}_M^0} p^e\left(x_i|pa(x_i)\right) d\boldsymbol{x}_M^0 d\boldsymbol{k}$$

$$= p(y|pa(y)) \Pi_{X_i \in \mathbf{K}_2} p^e\left(x_i|pa(x_i)\right) \int \Pi_{X_i \in \mathbf{K}} p^e\left(x_i|pa(x_i)\right) \Pi_{X_i \in \mathbf{X}_M^0} p^e\left(x_i|pa(x_i)\right) d\boldsymbol{x}_M^0 d\boldsymbol{k}$$

$$= p(y|pa(y)) \Pi_{X_i \in \mathbf{K}_2} p^e\left(x_i|pa(x_i)\right)$$

and

$$\tilde{p}^e\left(\boldsymbol{k}_2, y\right) = \int p(y|pa(y)) \Pi_{X_i \in \mathbf{K}_2} p^e\left(x_i|pa(x_i)\right) \Pi_{X_i \in \mathbf{K}} p^e\left(x_i|pa(x_i)\right) p^e\left(\boldsymbol{x}_M^0\right) d\boldsymbol{x}_M^0 d\boldsymbol{k}$$

$$= p(y|pa(y)) \Pi_{X_i \in \mathbf{K}_2} p^e\left(x_i|pa(x_i)\right) \int \Pi_{X_i \in \mathbf{K}} p^e\left(x_i|pa(x_i)\right) p^e\left(\boldsymbol{x}_M^0\right) d\boldsymbol{x}_M^0 d\boldsymbol{k}$$

$$= p(y|pa(y)) \Pi_{X_i \in \mathbf{K}_2} p^e\left(x_i|pa(x_i)\right)$$

Therefore, we have $\widetilde{p}^e(\boldsymbol{k}_2, y) = p^e(\boldsymbol{k}_2, y)$.

Note that $\mathbf{K}_2 \subset \mathbf{X}$, we have

$$\mathrm{Var}_{P^e}\left(Y|\mathbf{K}_2\right) = \mathrm{E}_{P^e}\left[\mathrm{Var}_{P^e}(Y|\mathbf{X})|\mathbf{K}_2\right] + \mathrm{Var}_{P^e}\left[\mathrm{E}_{P^e}(Y|\mathbf{X})|\mathbf{K}_2\right],$$

therefore

$$\mathrm{E}_{P^e}\left[\mathrm{Var}_{P^e}\left(Y|\mathbf{K}_2\right)\right] = \mathrm{E}_{P^e}\left[\mathrm{Var}_{P^e}(Y|\mathbf{X})\right] + \mathrm{E}_{P^e}\left[\mathrm{Var}_{P^e}\left[\mathrm{E}(Y|\mathbf{X})|\mathbf{K}_2\right]\right],$$

and hence $\mathrm{E}_{P^e}\left[\mathrm{Var}_{P^e}\left(Y|\mathbf{K}_2\right)\right] \geq \mathrm{E}_{P^e}\left[\mathrm{Var}_{P^e}(Y|\mathbf{X})\right]$.

(iii) Because $\tilde{p}^e\left(\boldsymbol{k}_2, y\right) = p^e\left(\boldsymbol{k}_2, y\right), \mathrm{E}_{\widetilde{P}^e}\left[\mathrm{Var}_{\widetilde{P}_e}\left(Y|\mathbf{K}_2\right)\right] = \mathrm{E}_{P^e}\left[\mathrm{Var}_{P^e}\left(Y|\mathbf{K}_2\right)\right]$, we have $\mathrm{E}_{\widetilde{P}^e}\left[\mathrm{Var}_{\widetilde{P}^e}\left(Y|\mathbf{K}_2\right)\right] \geq \mathrm{E}_{P^e}\left[\mathrm{Var}_{P^e}(Y|\mathbf{X})\right]$ Besides, since $\tilde{p}^e(y|\boldsymbol{x}) = \tilde{p}^e\left(y|\boldsymbol{k}_2\right)$, $\mathrm{E}_{\widetilde{P}^e}\left[\mathrm{Var}_{\widetilde{P}^e}(Y|\mathbf{X})\right] = \mathrm{E}_{\widetilde{P}^e}\left[\mathrm{Var}_{\widetilde{P}^e}\left(Y|\mathbf{K}_2\right)\right] \geq \mathrm{E}_{P^e}\left[\mathrm{Var}_{P^e}(Y|\mathbf{X})\right]$.

(iv) In summary, for each $P^e \in \mathcal{P}$, we may construct $\widetilde{P}^e$ such that

$$\mathrm{E}_{\widetilde{P}^e}\left[\mathrm{Var}\,\widetilde{P}^e(Y|\mathbf{X})\right] \geq \mathrm{E}_{P^e}\left[\mathrm{Var}_{P^e}(Y|\mathbf{X})\right].$$

Denote $\widetilde{\mathcal{P}} := \left\{\widetilde{P^e}|P^e \in \mathcal{P}\right\}$ and $P^* := \mathrm{argmax}_{P \in \mathcal{P}} \mathrm{E}_P\left[\mathrm{Var}_P(Y|\mathbf{X})\right]$, then $P^* \in \widetilde{\mathcal{P}}$. Besides, note that for any $\widetilde{P}^e \in \widetilde{\mathcal{P}}, \mathrm{E}_{\widetilde{P}^e}[Y|\boldsymbol{x}] = \mathrm{E}_{\widetilde{P}^e}\left[Y|\boldsymbol{x}_S, \boldsymbol{x}_M^1, do\left(\boldsymbol{x}_M^0\right)\right] = \mathrm{E}_{\widetilde{P}^e}[Y|\boldsymbol{x}_S, do(\boldsymbol{x}_M)]$, so $f^*(\boldsymbol{x}) = \mathrm{E}_{P^*}[Y|\boldsymbol{x}] = \mathrm{E}_{P^*}[Y|\boldsymbol{x}_S, do(\boldsymbol{x}_M)]$. As $\mathrm{E}_{P^e}[Y|\boldsymbol{x}_S, do(\boldsymbol{x}_M)]$ is invariant for all $P^e \in \mathcal{P}$. Therefore, we have $f^*(\boldsymbol{x}) = \mathrm{E}_{P^e}[Y|\boldsymbol{x}_S, do(\boldsymbol{x}_M)]$.

$\square$

### B.2 Proof for Prop. 3.2: Testability of Thm. 3.1

**Proposition 3.2.** *Denote* $\mathbf{X}_M^0 := \mathbf{X}_M \cap Ch(Y)$ *as mutable variables in* $Y$*'s children, and* $\mathbf{K} := De(\mathbf{X}_M^0) \backslash \mathbf{X}_M^0$ *as descendants of* $\mathbf{X}_M^0$*. Under assumptions 2.1, 2.2, the* $\mathbf{K}$ *is identifiable; besides, we can determine whether* $Y \to \mathbf{K}$ *from the joint distribution over training domains.*

*Proof.* We firstly show $\mathbf{K}$ is identifiable. Since all variables in $\mathbf{K}$ are descendants of $Y$, we have $Y \to X_i, X_i \in \mathbf{K}$ iff $X_i$ is adjacent to $Y$ in the skeleton of DAG (which is identifiable under assumption 2.2). Thus, we can determine whether $Y \to \mathbf{K}$.

---

**Algorithm 2** Detection of $\mathbf{X}_M$ and construct the causal skeleton of $G$

---

1. Start with $\mathbf{X}_M = \varnothing$. For $V_i \in \mathbf{V}$, test if $V_i \perp E$ or if there exist a subset $\mathbf{C}_{v_i,e} \subseteq \mathbf{V}$ such that $V_i \perp E | \mathbf{C}_{v_i,e}$. If $V_i \not\perp E$ and there exists no such $\mathbf{C}_{v_i,e}$, then include $V_i$, $\mathbf{X}_M = \mathbf{X}_M \cup V_i$.

2. Start with an undirected graph $G$ including edges for any two variables in $\mathbf{V}$ and the arrows $E \rightarrow V_i$ for $V_i \in \mathbf{X}_M$. For each pair of $\{V_i, V_j\}$. If $V_i \perp V_j$ or there exists a subset $\mathbf{C}_{v_i,v_j} \subset \mathbf{V}$ such that $V_i \perp V_j | \mathbf{C}_{v_i,v_j}$, we delete the edge $V_i - V_j$ from $G$.

---

Note that $\mathbf{K} = (\mathbf{X} \setminus \mathbf{X}_M^0) \cap \mathrm{De}(\mathbf{X}_M^0) = (\mathbf{X} \setminus \mathbf{X}_M^0) \cap \{\mathrm{De}(\mathbf{X}_M^0) \cup \mathbf{X}_M^0\}$. So it suffices to prove the identifiability of $\mathbf{X}_M^0 \cup \mathrm{De}(\mathbf{X}_M^0)$, where $\mathbf{X}_M^0 := \mathbf{X}_M \cap \mathrm{Ch}(Y)$. This can be accomplished by three steps: (i) identification of $\mathbf{X}_M$, (ii) identification of $\mathbf{X}_M^0$, and (iii) identification of $\mathbf{X}_M^0 \cup \mathrm{De}(\mathbf{X}_M^0)$.

The following algorithm shows step (i), which is the same as Huang et al. (2020).

The following Alg. 3 shows the steps (ii) and (iii), which basically relies on the faithful assumption (conditional independence in probability $\Rightarrow$ d-separation in graph).

---

**Algorithm 3** Detection of $\mathbf{X}_M^0$ and $\mathbf{X}_M^0 \cup \mathrm{De}(\mathbf{X}_M^0)$

---

2.1 Detect $\mathbf{X}_M^0 := \mathbf{X}_M \cap \mathrm{Ch}(Y)$:

1: **for** $X_i \in \mathbf{X}_M$ and adjacent to $Y$ **do**
2:     If $Y \not\perp E | \mathbf{C}_{y,e} \cup \{X_i\}$
3:     then $X_i \in \mathbf{X}_M \cap \mathrm{Ch}(Y)$
4: **end for**

2.2 Detect $\{\mathrm{Ch}(Y) \cap \mathbf{X}_M\} \cup \mathrm{De}(\mathrm{Ch}(Y) \cap \mathbf{X}_M)$

1: Start with $\mathbf{A} = \mathbf{B} = \mathrm{Ch}(Y) \cap \mathbf{X}_M$ and $\mathrm{visited}(X_i) = \mathrm{FALSE}$
2: **while** $\mathbf{B} \neq \varnothing$ **do**
3:    **for** $X_j \in \mathbf{B}$ **do**
4:      **for** $X_i \in \mathrm{Adj}(X_j)$ **do**
5:       **if** $X_i \notin \mathbf{X}_M$ and $X_i \perp E | \mathbf{C}_{e,x_i} \cup \{X_j\} \setminus \mathbf{D}_{x_i,e}$ **then**
6:         $\mathbf{A} = \mathbf{A} \cup \{X_i\}$
7:         **if** $\mathrm{visited}(X_i) = \mathrm{FALSE}$ **then**
8:           $\mathbf{B} = \mathbf{B} \cup \{X_i\}$
9:         **end if**
10:       **end if**
11:       **if** $X_i \in \mathbf{X}_M$ and $X_i \notin \mathrm{Adj}(Y)$ and $X_i \perp Y | \mathbf{C}_{x_i,y} \cup \{X_j\} \setminus \mathbf{D}_{y,x_j}$ **then**
12:         $\mathbf{A} = \mathbf{A} \cup \{X_i\}$
13:         **if** $\mathrm{visited}(X_i) = \mathrm{FALSE}$ **then**
14:           $\mathbf{B} = \mathbf{B} \cup \{X_i\}$
15:         **end if**
16:       **end if**
17:      **end for**
18:      Let $\mathbf{B} = \mathbf{B} \setminus \{X_j\}$
19:    **end for**
20: **end while**

---

Explanations for 2.2 :

- Line 1 in 2.2 : The set $\mathbf{A}$ is the final output. The set $\mathbf{B}$ only plays a part as an instrumental set that starts with $\mathbf{X}_M \cap \mathrm{Ch}(Y)$ and ends with $\varnothing$. During the process, $\mathbf{B}$ stores the nodes in $\mathbf{X}_M \cap \mathrm{Ch}(Y)$ that has not been searched for the children. Once $X_j \in \mathbf{B}$ is searched, it is excluded from the set $\mathbf{B}$ (Line 18 ) and the children of $X_j$ are added to $\mathbf{B}$ if it has not been visited (Line 8 and 14), which is essentially a breadth-first-search algorithm.

- Line 5 to 10 (the case when $X_i \notin \mathbf{X}_M$): The fact $X_i \notin \mathbf{X}_M$ means $E$ and $X_i$ are not adjacent. Besides, note that $X_j \in \mathbf{X}_M \cap \mathrm{Ch}(Y)$, there is a structure in the form $E \to \cdots \to X_j - X_i$ where $X_i$ and $E$ are not adjacent. In the notation $X_i \perp E | \mathbf{C}_{x_i,e} \cup \{X_j\} \setminus \mathbf{D}_{e,x_j}$, $\mathbf{C}_{x_i,e}$ denotes a separating set such that $X_i \perp E | \mathbf{C}_{x_i,e}$ and $\mathbf{D}_{e,x_j}$ denotes the set of variables along the directed path between $E \to \cdots \to X_j$. The existence of $\mathbf{C}_{x_i,e}$ is guaranteed since $X_i$ and $E$ are not adjacent, so a separating set has been found when constructing the skeleton. The set $\mathbf{D}_{e,x_j}$ is also clear as it is determined in the breadth-first-search process.

- Line 11 to 19 (the case when $X_i \in \mathbf{X}_M$): Firstly, we explain why it is unnecessary to consider the case when $X_i \in \mathbf{X}_M$ and $X_i \in \mathrm{Adj}(Y)$. If $X_i \in \mathrm{PA}(Y)$, $X_i$ can not be in $\mathrm{De}(\mathrm{Ch}(Y) \cap \mathbf{X}_M) \cup \{\mathrm{Ch}(Y) \cap \mathbf{X}_M\}$ as it would induce a cycle in this way. If $X_i \in \mathrm{Ch}(Y)$, it means $X_i \in \mathrm{Ch}(Y) \cap \mathbf{X}_M$ and has been identified in 2.1 and included in set $\mathbf{A}$ in the beginning.
  So the remaining case is when $X_i \in \mathbf{X}_M$ and $X_i \notin \mathrm{Adj}(Y)$. Note in this case $X_j \in \mathrm{Ch}(Y)$ or $X_j \in \mathrm{De}(Y)$, there exists a structure $Y \to \cdots \to X_j - X_i$, which is the same as $E \to \cdots \to X_j - X_i$ in Line 5 to 10.

$\square$

## B.3 Counter Example of $f^* \neq f_S$

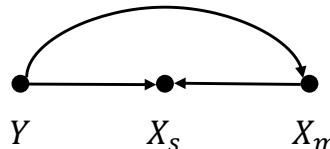

Figure 8: DAG of the counter example.

Consider the DAG in Fig. 8, in which we denote $Y, X_s, X_m$ are binary variables. We will show that in this scenario, there exists $P(Y), P(X_s|X_m, Y)$ such that $f_S := \mathbb{E}[Y|x_s, do(x_m)]$ is not min-max optimal. We show this by proving that:

$$\mathbb{E}\left[Y - \mathbb{E}[Y|x_s, do(x_m)]\right]^2 > \mathbb{E}\left[Y - \mathbb{E}[Y|do(x_m)]\right]^2. \tag{6}$$

Since we have that

$$\mathbb{E}\left[Y - \mathbb{E}[Y|x_s, do(x_m)]\right]^2 = \mathbb{E}[Y^2] + \mathbb{E}\left[\mathbb{E}^2[Y|x_s, do(x_m)]\right] - 2\mathbb{E}[Y \cdot \mathbb{E}[Y|x_s, do(x_m)]],$$

and that $\mathbb{E}\left[Y - \mathbb{E}(Y|do(x_m))\right]^2 = E[Y^2] - E[Y]^2$ due to that $p(y|do(x_m)) = p(y)$, the Eq. equation 6 is equivalent to that

$$\mathbb{E}\left[\mathbb{E}^2[Y|x_s, do(x_m)]\right] > 2\mathbb{E}[Y \cdot \mathbb{E}[Y|x_s, do(x_m)]] - E^2[Y]. \tag{7}$$

Besides, we have

$$\mathbb{E}\left[\mathbb{E}^2[Y|x_s, do(x_m)]\right] = \sum_{x_s, x_m} \left[\left[\sum_y p(x_s|x_m, y)p(x_m|y)p(y)\right] \cdot \mathbb{E}^2[Y|x_s, do(x_m)]\right], \tag{8}$$

$$\mathbb{E}[Y \cdot \mathbb{E}[Y|x_s, do(x_m)]] = \sum_{x_s, x_m} \left[\left[\sum_y p(x_s|x_m, y)p(x_m|y)p(y) \cdot y\right] \cdot \mathbb{E}[Y|x_s, do(x_m)]\right]. \tag{9}$$

Since we have $p(y|x_s, do(x_m)) = \frac{p(y)p(x_s|x_m, y)}{\sum_y p(y)p(x_s|x_m, y)}$, we have

$$\mathbb{E}[Y|x_s, do(x_m)] = \frac{p(y=1)p(x_s|x_m, y=1)}{\sum_y p(y)p(x_s|x_m, y)}. \tag{10}$$

Substituting Eq. equation 10 into Eq. equation 8, equation 9, we have

$$\mathbb{E}\left[\mathbb{E}^2[Y|X_s, do(X_m)]\right] = \sum_{x_s, x_m} \left[\left[\sum_y p(x_s|x_m, y)p(x_m|y)p(y)\right] \cdot \left[\frac{p(y=1)p(x_s|x_m, y=1)}{\sum_y p(y)p(x_s|x_m, y)}\right]^2\right],$$

$$\mathbb{E}\left[Y \cdot \mathbb{E}[Y|X_s, do(X_m)]\right] = \sum_{x_s, x_m} \left[\left[\sum_y p(x_s|x_m, y)p(x_m|y)p(y) \cdot y\right] \cdot \left[\frac{p(y=1)p(x_s|x_m, y=1)}{\sum_y p(y)p(x_s|x_m, y)}\right]\right]$$

$$= \sum_{x_s, x_m} \left[\left[\sum_y p(x_s|x_m, y=1)p(x_m|y=1)p(y=1)\right] \cdot \left[\frac{p(y=1)p(x_s|x_m, y=1)}{\sum_y p(y)p(x_s|x_m, y)}\right]\right].$$

Denote $a_y := p(y = 1)$, $p(x_m = 1|y) := a_{my}$, $p(x_s = 1|x_m, y) = a_{smy}$, then the left hand side in Eq. equation 7 has

$$\mathbb{E}\left[\mathbb{E}^2[Y|x_s, do(x_m)]\right] = \mathbb{1}(x_s=1, x_m=1)\left(a_{s11}a_{m1}a_y + a_{s10}a_{m0}(1-a_y)\right)\left[\frac{a_y a_{s11}}{a_y a_{s11} + (1-a_y)a_{s10}}\right]^2 +$$

$$\mathbb{1}(x_s=1, x_m=0)\left[a_{s11}(1-a_{m1})a_y + a_{s10}(1-a_{m0})(1-a_y)\right]\left[\frac{a_y a_{s01}}{a_y a_{s01} + (1-a_y)a_{s00}}\right]^2 +$$

$$\mathbb{1}(x_s=0, x_m=1)\left[(1-a_{s11})a_{m1}a_y + (1-a_{s10})a_{m0}(1-a_y)\right]\left[\frac{a_y(1-a_{s11})}{a_y(1-a_{s11}) + (1-a_y)(1-a_{s10})}\right]^2 +$$

$$\mathbb{1}(x_s=0, x_m=0)\left[(1-a_{s01})(1-a_{m1})a_y + (1-a_{s00})(1-a_{m0})(1-a_y)\right]\left[\frac{a_y(1-a_{s01})}{a_y(1-a_{s01}) + (1-a_y)(1-a_{s00})}\right]^2.$$

The right-hand side has

$$2\mathbb{E}\left[Y\mathbb{E}[Y|x_s, do(x_m)]\right] - \mathbb{E}[Y^2] = 2\Big[\mathbb{1}(x_s = 1, x_m = 1)\frac{a_y^2 a_{s11}^2 a_{m1}}{a_y a_{s11} + (1-a_y)a_{s10}} +$$

$$\mathbb{1}(x_s = 1, x_m = 0)\frac{a_y^2 a_{s01}^2 (1-a_{m1})}{a_y a_{s01} + (1-a_y)a_{s00}} +$$

$$\mathbb{1}(x_s = 0, x_m = 1)\frac{a_y^2 (1-a_{s11})^2 a_{m1}}{a_y(1-a_{s11}) + (1-a_y)a_{s10}} +$$

$$\mathbb{1}(x_s = 0, x_m = 0)\frac{a_y^2 (1-a_{s01})(1-a_{m1})}{a_y(1-a_{s01}) + (1-a_y)(1-a_{s00})}\Big] - a_y^2.$$

When $a_y \neq 0$, the term $a_y^2$ can be removed. Then let $a_y \to 0$, the left-hand side approximates to:

$$\frac{a_{s11}^2 a_{m0}}{a_{s10}} + \frac{a_{s01}^2 (1-a_{m0})}{a_{s00}} + \frac{(1-a_{s11})^2 a_{m0}}{(1-a_{s10})} + \frac{(1-a_{s01})^2 (1-a_{m0})}{(1-a_{s00})};$$

and the right hand side approximates to:

$$2 \cdot \left[\frac{a_{s11}^2 a_{m1}}{a_{s10}} + \frac{a_{s01}^2 (1-a_{m1})}{a_{s00}} + \frac{(1-a_{s11})^2 a_{m1}}{(1-a_{s10})} + \frac{(1-a_{s01})^2 (1-a_{m1})}{(1-a_{s00})}\right] - 1$$

Then the Eq. equation 7 is equivalent to:

$$\frac{a_{s11}^2 (a_{m0}-2a_{m1})}{a_{s10}} + \frac{a_{s01}^2 (2a_{m1}-a_{m0}-1)}{a_{s00}} + \frac{(1-a_{s11})^2 (a_{m0}-2a_{m1})}{(1-a_{s10})} + \frac{(1-a_{s01})^2 (2a_{m1}-a_{m0}-1)}{(1-a_{s00})} > -1.$$

Let $a_{s10} \to 0$, $a_{s11} \to 1$, $a_{s01} = a_{s10} = 0.5$ and $a_{m0} - 2a_{m1} > 0$, the above inequality holds.

### B.4 PROOF FOR THM. 3.3: MIN-MAX PROPERTY

**Theorem 3.3** (Min-max Property). *Denote $h^*(S_-) := \max_J \mathbb{E}_{P_J}[(Y - f_{S_-}(\boldsymbol{x}))^2]$ as the maximal expected loss over J for $S_-$. Then, we have $h^*(S_-) = \max_{P^e \in \mathcal{P}} \mathbb{E}_{P^e}[(Y - f_{S_-}(\boldsymbol{x}))^2]$. In this regard, the optimal subset $S^*$ for $f^* = f_{S^*}$ can be attained via $S^* := argmin_{S_- \subset S} h^*(S_-)$.*

*Proof.* We show the maximum loss is attained when $\mathbf{X}_M$ is a definite function of $\mathrm{PA}(\mathbf{X}_M)$

Let $f_{S_-}(\boldsymbol{x}_{S_-}, \boldsymbol{x}_M) := E[Y|\boldsymbol{x}_{S_-}, do(\boldsymbol{x}_M)]$ be an invariant predictor. Then

$$\mathcal{L}_{P^e}(f_{S_-}) = \int_{\boldsymbol{x}} \int_y (y - f_{S_-}(\boldsymbol{x}_{S_-}, \boldsymbol{x}_M))^2 p(y|pa(y)) \prod_{i \in S} p(x_i|pa(x_i)) \prod_{i \in M} p^e(x_i|pa(x_i)) dy d\boldsymbol{x}.$$

And the maximum loss

$$\mathcal{L}_{S_-}^* = \mathrm{argmax}_{P^e} \mathcal{L}_{P^e}(f_{S_-}) = \mathrm{argmax}_{\{p^e(x_i|pa(x_i))|i \in M\}} \mathcal{L}_{P^e}(f_{S_-}).$$

Let $\mathbf{X}' := \mathbf{X} \backslash (\mathbf{X}_M \cup \mathrm{PA}(\mathbf{X}_M))$ and $h(\boldsymbol{x}_M, pa(\boldsymbol{x}_M)) := \int_{\boldsymbol{x}'} (y - f_{S_-}(\boldsymbol{x}))^2 \prod_{X_i \in \mathbf{X}'} p(x_i|pa(x_i)) d_{\boldsymbol{x}'}$, which does not rely on the mutable distribution $\{p^e(x_i|pa(x_i))|i \in M\}$. Let $\boldsymbol{m}^*(pa(\boldsymbol{x}_M)) := \mathrm{argmax}_{\boldsymbol{x}_M} h(\boldsymbol{x}_M, pa(\boldsymbol{x}_M)))$.

Firstly, consider the case of $\mathbf{X}_M = \{X_M\}$. Then

$$\max_{P^e} \mathcal{L}_{P^e}(f_{S_-}) = \max_{P^e} \int_{pa(\boldsymbol{x}_M)} \left( \int_{\boldsymbol{x}_M} h(\boldsymbol{x}_M, pa(\boldsymbol{x}_M)) p^e(\boldsymbol{x}_M|pa(\boldsymbol{x}_M)) d\boldsymbol{x}_M \right) \prod_{X_i \in \mathrm{PA}(\mathbf{X}_M)} p(x_i|pa(x_i)) dpa(\boldsymbol{x}_M)$$

$$= \int_{pa(\boldsymbol{x}_M)} \max_{P^e} \left( \int_{\boldsymbol{x}_M} h(\boldsymbol{x}_M, pa(\boldsymbol{x}_M)) p^e(\boldsymbol{x}_M|pa(\boldsymbol{x}_M)) d\boldsymbol{x}_M \right) \prod_{X_i \in \mathrm{PA}(\mathbf{X}_M)} p(x_i|pa(x_i)) dpa(\boldsymbol{x}_M)$$

$$= \int_{pa(\boldsymbol{x}_M)} h(\boldsymbol{m}^*(pa(\boldsymbol{x}_M)), pa(\boldsymbol{x}_M)) \prod_{X_i \in \mathrm{PA}(\mathbf{X}_M)} p(x_i|pa(x_i)) dpa(\boldsymbol{x}_M).$$

When $\mathbf{X}_M$ is multivariate, we consider the maximization sequentially by the topological order $\{X_{M,1}, X_{M,2}, \cdots, X_{M,l}\}$, where $X_{M,j}$ is a node that is not a parent of any other nodes in $\{X_{M,i}|i \geq j\}$ in the sub-graph over $\mathbf{X}_M$. That is, we firstly consider $\max_{p^e(x_{M,1}|pa(x_{M,1}))} \int_{x_{M,1}} h(x_{M,1}, pa_{x_{M,1}}) p^e(x_{M,1}|pa_{x_{M,1}}) d_{x_{m,1}}$ and factorize $\max_{P^e}\{\cdots\}$ as

$$\max_{p^e(x_{M,l}|pa(x_{M,l}))} \cdots \max_{p^e(x_{M,2}|pa(x_{M,2}))} \max_{p^e(x_{M,1}|pa(x_{M,1}))} \{\cdots\}.$$

Note that the sub-graph on $\mathbf{X}_M$ is always a DAG, so such a topological order always exists.

$\square$

## B.5 PROOF FOR PROP. 3.4: IDENTIFIABILITY IN THM. 3.3

**Proposition 3.4.** *Denote* $P_J := p(y, \boldsymbol{x}_S|do(\mathbf{X}_M = J(pa(\boldsymbol{x}_M))))$, *and* $f_{S_-} := \mathcal{E}[Y|\boldsymbol{x}_{S_-}, do(\boldsymbol{x}_M)]$. *Under assumptions 2.1 and 2.2, we have* $\mathrm{PA}(\mathbf{X}_M)$, $P_J$, *and* $f_{S_-}$ *are identifiable.*

*Proof.* To generate data distributed as $P_J$, we need to use $J(\mathrm{PA}(\mathbf{X}_M))$ to regenerate $\mathbf{X}_M$, then regenerate $\mathrm{De}(\mathbf{X}_M)$ with structural equations. To estimate $f_{S_-}$, we need to intervene on $\mathbf{X}_M$, then regenerate $\mathrm{De}(\mathbf{X}_M)$ with structural equations. So, it's suffice to show $\mathbf{X}_M, \mathrm{De}(\mathbf{X}_M), \mathrm{PA}(\mathbf{X}_M), \mathrm{PA}(\mathrm{De}(\mathbf{X}_M))$ are identifiable.

Identification of $\mathbf{X}_M$ has been shown in Alg. 2. **We first show the identification of** $\mathrm{De}(\mathbf{X}_M)$.

- Line 5 to 10: this case is the same as Algorithm 3, which is based on (i) the structure $E \rightarrow \cdots \rightarrow X_j - X_i$ and (ii) $X_i$ and $E$ are not adjacent.

- Line 11 to 19: In this case, we identify the direction between $X_i$ and $X_j$ by the "Independent Causal Mechanism (ICM) Principle" following Huang et al. (2020), where $\widehat{\Delta}_{X_j \rightarrow X_i}$ and $\widehat{\Delta}_{X_i \rightarrow X_j}$ are the estimated HSIC (see Eq. 17 in Huang et al. (2020) for the detailed formulation of $\widehat{\Delta}$).

  The ICM principle means that "the conditional distribution of each variable given its causes (i.e., its mechanism) does not inform or influence the other mechanisms.". That is, the changes of $P(X_i|\mathrm{PA}(X_i))$ does not influence the other mechanisms $P(X_j|\mathrm{PA}(X_j))$ for $j \neq i$. The ICM principle is implied in the definition of "structural causal model" in Pearl (2009), where each structural equation represents an autonomous physical mechanism.

---

**Algorithm 4** Detection of $\mathrm{De}(\mathbf{X}_M) \cup \mathbf{X}_M$

---

1: Start with $\mathbf{A} = \mathbf{B} = \mathbf{X}_M$ and $\mathrm{visited}(X_i) = \mathrm{FALSE}$
2: **while** $\mathbf{B} \neq \varnothing$ **do**
3:   **for** $X_j \in \mathbf{B}$ **do**
4:     **for** $X_i \in \mathrm{Adj}(X_j)$ **do**
5:       **if** $X_i \notin \mathbf{X}_M$ and $X_i \perp E | \mathbf{C}_{e,x_i} \cup \{X_j\} \setminus \mathbf{D}_{x_i,e}$ **then**
6:         $\mathbf{A} = \mathbf{A} \cup \{X_i\}$
7:         **if** $\mathrm{visited}(X_i) = \mathrm{FALSE}$ **then**
8:           $\mathbf{B} = \mathbf{B} \cup \{X_i\}$
9:         **end if**
10:       **end if**
11:       **if** $X_i \in \mathbf{X}_M$ and $\widehat{\Delta}_{X_j \to X_i} < \widehat{\Delta}_{X_i \to X_j}$ **then**
12:         $\mathbf{A} = \mathbf{A} \cup \{X_i\}$
13:         **if** $\mathrm{visited}(X_i) = \mathrm{FALSE}$ **then**
14:           $\mathbf{B} = \mathbf{B} \cup \{X_i\}$
15:         **end if**
16:       **end if**
17:     **end for**
18:     Let $\mathbf{B} = \mathbf{B} \setminus \{X_j\}$
19:   **end for**
20: **end while**

---

**We then show the identification of** $\mathrm{PA}(\mathbf{X}_M), \mathrm{PA}(\mathrm{De}(\mathbf{X}_M))$.

---

**Algorithm 5** $\mathrm{PA}(X_i)$ for $X_i \in \mathbf{X}_M \cup \mathrm{De}(\mathbf{X}_M)$.

---

1: **for** $X_j \in \mathbf{X}_M \cup \mathrm{De}(\mathbf{X}_M)$ **do**
2:   **for** $X_i \in \mathrm{Adj}(X_j)$ **do**
3:     **if** $X_i \notin \mathbf{X}_M$ **then**
4:       $X_i \in \mathrm{PA}(X_j)$ if $X_i \not\perp E | \{\mathbf{C}_{e,x_i} \setminus \mathbf{D}_{e,X_i} \cup \{X_j\}\}$
5:     **else if** $X_j \in \mathbf{X}_M$ and $X_i \in \mathbf{X}_m$ **then**
6:       $X_i \in \mathrm{PA}(X_j)$ when $\widehat{\Delta}_{X_j \to X_i} < \widehat{\Delta}_{X_i \to X_j}$.
7:     **else if** $X_j \in \mathbf{X}_M$ and $X_i \notin \mathbf{X}_m$ **then**
8:       $X_i \in \mathrm{PA}(X_j)$ when $E \not\perp X_i | \mathbf{C}_{x_i,e} \cup \{X_j\}$
9:     **end if**
10:   **end for**
11: **end for**

---

- Line 4: this rule is based on the structure $E \to \cdots \to X_j - X_i$ and $\{X_i, E\}$ are not adjacent.

- Line 6: this rule is based on the HSIC criterion in Huang et al. (2020).

- Line 8: this rule is based on the structure $E \to X_i - X_j$ and $\{E, X_j\}$ are not adjacent.

$\square$

## C APPENDIX FOR SEC. 3.2: LEARNING METHOD

### C.1 EQUIVALENT CLASSES AND ITS RECOVERY ALGORITHM

When the graphical condition in Thm. 3.1 fails, Alg. 1 needs to search over subsets of the stable set and identify the optimal predictor. However, we find an exhaustive search of all subsets is redundant, as some subsets are equivalent in the sense of predicting $Y$. Formally speaking,

**Definition C.1** (*p*-equivalence). Two subsets $\mathbf{X}_i$ and $\mathbf{X}_j$ (the subscript $S$ is omitted for simplicity) of the stable set $\mathbf{X}_S$ are probabilistical equivalent, *i.e.*, $\mathbf{X}_i \sim_P \mathbf{X}_j$, if $P(Y|\mathbf{X}_i, do(\mathbf{X}_M)) = P(Y|\mathbf{X}_j, do(\mathbf{X}_M))$.

*Remark* C.2. It's straight forward to see $\sim_P$ satisfies reflexivity ($\mathbf{X}_i \sim_P \mathbf{X}_i$), symmetry ($\mathbf{X}_i \sim_P \mathbf{X}_j \Rightarrow \mathbf{X}_j \sim_P \mathbf{X}_i$), and transitivity ($\mathbf{X}_i \sim_P \mathbf{X}_j, \mathbf{X}_j \sim_P \mathbf{X}_k \Rightarrow \mathbf{X}_i \sim_P \mathbf{X}_k$), thus is a legal equivalent relation.

Under the Markovian assumption, we can infer $p$-equivalence from structure of the causal graph, especially patterns of $d$-separations. In the following, we firstly give the notion of *graphical equivalence*, then show how can we infer $p$-equivalence from it.

**Definition C.3** ($g$-equivalence). Two subsets of vertex $\mathbf{X}_i$ and $\mathbf{X}_j$ are graphically equivalent w.r.t the causal graph $G$, *i.e.*, $\mathbf{X}_i \sim_G \mathbf{X}_j$, if $\exists \mathbf{X}_{ij} \subseteq \mathbf{X}_i \cap \mathbf{X}_j$ such that $Y \perp_{G_{\overline{\mathbf{X}_M}}} \mathbf{X}_i \cup \mathbf{X}_j \backslash \mathbf{X}_{ij} | \mathbf{X}_{ij}, \mathbf{X}_M$.

It's straight forward to see that $\sim_G$ satisfies reflexivity ($\mathbf{X}_i \sim_G \mathbf{X}_i$) and symmetry ($\mathbf{X}_i \sim_G \mathbf{X}_j \Rightarrow \mathbf{X}_j \sim_G \mathbf{X}_i$). To prove it also satisfies transitivity ($\mathbf{X}_i \sim_G \mathbf{X}_j, \mathbf{X}_j \sim_G \mathbf{X}_k \Rightarrow \mathbf{X}_i \sim_G \mathbf{X}_k$), we need to introduce two properties of the $d$-separation.

**Lemma C.4** (Properties of $d$-separation). *(i) If a path $p$ can not be blocked by a vertex set $\mathbf{X}_i$, then any of $p$'s sub-path can not be blocked by $\mathbf{X}_i$ either. (ii) For two vertex sets $\mathbf{X}_i, \mathbf{X}_j$, and a path $p$, if $p$ can not be blocked by $\mathbf{X}_i$ and can be blocked by $\mathbf{X}_i \cup \mathbf{X}_j$, then, $\mathbf{X}_j$ must contain a non-collider in $p$.*

*Proof.* The correctness of property-(i) is straightforward, so we focus on proving property-(ii). Specifically, there are three possibilities about the path $p$:

**1**. all vertices in $p$ are non-colliders. From '$p$ can not be blocked by $\mathbf{X}_i$', we know all vertices in $p$ are not in $\mathbf{X}_i$. From '$p$ can be blocked by $\mathbf{X}_i \cup \mathbf{X}_j$', we know at least a vertex in $p$ is in $\mathbf{X}_j$.

**2**. all vertices in $p$ are colliders. From '$p$ can not be blocked by $\mathbf{X}_i$', we know $\forall X \in p$, $X \in \mathbf{X}_i$ or $X$ has a descendant in $\mathbf{X}_i$. So, $\forall X \in p$, $X \in \mathbf{X}_i \cup \mathbf{X}_j$ or $X$ has a descendant in $\mathbf{X}_i \cup \mathbf{X}_j$, which means $p$ can not be blocked by $\mathbf{X}_i \cup \mathbf{X}_j$ neither.

**3**. vertices in $p$ are colliders and non-colliders. From '$p$ can not be blocked by $\mathbf{X}_i$', we know $\forall X \in p$, if $X$ is a non-collider, $X \notin \mathbf{X}_i$, if $X$ is a collider, $X$ or one vertex in $\mathrm{Dec}(X)$ is in $\mathbf{X}_i$, thus in $\mathbf{X}_i \cup \mathbf{X}_j$. So, $\mathbf{X}_j$ must contain a non-collider in $p$, otherwise, $p$ can not be blocked by $\mathbf{X}_i \cup \mathbf{X}_j$. $\square$

Equipped with the above properties, we now show $\sim_G$ also satisfies transitivity, *i.e.*, $\mathbf{X}_i \sim_G \mathbf{X}_j, \mathbf{X}_j \sim_G \mathbf{X}_k \Rightarrow \mathbf{X}_i \sim_G \mathbf{X}_k$.

*Proof.* Because $\mathbf{X}_i \sim_G \mathbf{X}_j, \mathbf{X}_j \sim_G \mathbf{X}_k$, by definition, $\exists \mathbf{X}_{ij} \subseteq \mathbf{X}_i \cap \mathbf{X}_j$ such that $Y \perp_G \mathbf{X}_i \cup \mathbf{X}_j \backslash \mathbf{X}_{ij} | \mathbf{X}_{ij}$, and $\exists \mathbf{X}_{jk} \subseteq \mathbf{X}_j \cap \mathbf{X}_k$ such that $Y \perp_G \mathbf{X}_j \cup \mathbf{X}_k \backslash \mathbf{X}_{jk} | \mathbf{X}_{jk}$. Different situation of $\mathbf{X}_{ij}$ and $\mathbf{X}_{jk}$ are discussed below:

**1**. $\mathbf{X}_{ij} = \mathbf{X}_{ij} = \mathbf{X}_0$. Then, we have $Y \perp_G \mathbf{X}_i \cup \mathbf{X}_j \cup \mathbf{X}_k \backslash \mathbf{X}_0 | \mathbf{X}_0$. So, $Y \perp_G \mathbf{X}_i \cup \mathbf{X}_k \backslash \mathbf{X}_0 | \mathbf{X}_0$ and $\mathbf{X}_0 \subseteq \mathbf{X}_i \cap \mathbf{X}_k$. So, $\mathbf{X}_i \sim_G \mathbf{X}_k$.

**2**. $\mathbf{X}_{ij} \cap \mathbf{X}_{jk} = \varnothing$. As shown by Fig. 9 (a), we have $\mathbf{X}_{jk} \subseteq \mathbf{X}_j \backslash \mathbf{X}_{ij}$, so, $Y \perp_G \mathbf{X}_{jk} | \mathbf{X}_{ij}$. We also have $\mathbf{X}_{ij} \subseteq \mathbf{X}_j \backslash \mathbf{X}_{jk}$, so, $Y \perp_G \mathbf{X}_{ij} | \mathbf{X}_{jk}$.

In the following, we show any path between $Y$ and $\mathbf{X}_{jk}$ contains at least a collider. We prove this by contradiction, *i.e.*, assume there is a path $p_0 :< Y, X_1, X_2, ..., X_m >$ between $Y$ and $\mathbf{X}_{jk}$ ($X_m \in \mathbf{X}_{jk}$) and every vertex in $p_0$ is a non-collider in $p_0$. Because $Y \perp_G \mathbf{X}_{jk} | \mathbf{X}_{ij}$, so, $\exists X_i, i \leq m-1$ in $p_0$ such that $X_i \in \mathbf{X}_{ij}$. So, there is a path $p_1 :< X_1, X_2, ..., X_i >$ between $Y$ and $\mathbf{X}_{ij}$ where every vertex is a non-collider. Because $Y \perp_G \mathbf{X}_{ij} | \mathbf{X}_{jk}$, so again, $\exists X_l, l \leq i-1$ in $p_1$ such that $X_l \in \mathbf{X}_{jk}$. Iterating like this, we have either $X_1 \in \mathbf{X}_{ij}$ or $X_1 \in \mathbf{X}_{jk}$. Because $X_1 \in \mathrm{Neig}(Y)$, $Y \not\perp_G X_1$ given any subset, which contradicts with $Y \perp \mathbf{X}_j \backslash \mathbf{X}_{jk} | \mathbf{X}_{jk}$ and $Y \perp \mathbf{X}_j \backslash \mathbf{X}_{ij} | \mathbf{X}_{ij}$.

Because any path between $Y$ and $\mathbf{X}_{jk}$ ($\mathbf{X}_{ij}$ can be similarly proved) contains at least a collider, we have $Y \perp_G \mathbf{X}_{ij} | \varnothing$ and $Y \perp_G \mathbf{X}_{jk} | \varnothing$.

In the following, we show any path between $Y$ and $\mathbf{X}_i \backslash \mathbf{X}_{ij}$ contains at least a collider. We prove this by contradiction, *i.e.*, assume there is a path $p_0 :< Y, X_1, X_2, ..., X_m >$ between $Y$ and $\mathbf{X}_i \backslash \mathbf{X}_{ij}$ ($X_m \in \mathbf{X}_i \backslash \mathbf{X}_{ij}$) and every vertex in $p_0$ is a non-collider in $p_0$. Because $Y \perp_G \mathbf{X}_i \backslash \mathbf{X}_{ij} | \mathbf{X}_{ij}$, so, $\exists X_i, i \leq m-1$ in $p_0$ such that $X_i \in \mathbf{X}_{ij}$. So, there is a path $p_1 :< X_1, X_2, ..., X_i >$ between $Y$ and $\mathbf{X}_{ij}$ where every vertex is a non-collider, which contradicts with $Y \perp_G \mathbf{X}_{ij} | \varnothing$.

Because any path path between $Y$ and $\mathbf{X}_i \backslash \mathbf{X}_{ij}$ contains at least a collider, we have $Y \perp_G \mathbf{X}_i \backslash \mathbf{X}_{ij} | \varnothing$ and similarly $Y \perp_G \mathbf{X}_k \backslash \mathbf{X}_{jk} | \varnothing$. Considering $Y \perp_G \mathbf{X}_{ij} | \varnothing$ and $Y \perp_G \mathbf{X}_{jk} | \varnothing$, we now have $Y \perp_G \mathbf{X}_i | \varnothing$ and $Y \perp_G \mathbf{X}_k | \varnothing$. Because $\varnothing \subseteq \mathbf{X}_i \cap \mathbf{X}_k$ and $Y \perp_G \mathbf{X}_i \cup \mathbf{X}_k \backslash \varnothing | \varnothing$, we have $\mathbf{X}_i \sim_G \mathbf{X}_k$.

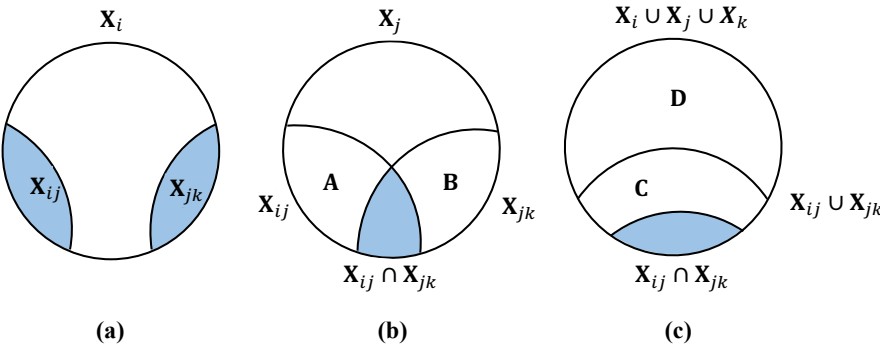

Figure 9

**3**. $\mathbf{X}_{ij} \cap \mathbf{X}_{jk} \neq \varnothing$. We first define $\mathbf{A} := \mathbf{X}_{ij} \backslash \mathbf{X}_{ij} \cap \mathbf{X}_{jk}$ and $\mathbf{B} := \mathbf{X}_{jk} \backslash \mathbf{X}_{ij} \cap \mathbf{X}_{jk}$, as shown by Fig. 9 (b).

In the following, we show any path between $Y$ and $\mathbf{A}$ can be blocked by $\mathbf{X}_{ij} \cap \mathbf{X}_{jk}$. We prove this by contradiction, *i.e.*, assume there is a path $p_0 :< Y, X_1, X_2, ..., X_m >$ between $Y$ and $\mathbf{A}$ such that $p_0$ can not be blocked by $\mathbf{X}_{ij} \cap \mathbf{X}_{jk}$. Because $\mathbf{A} \subseteq \mathbf{X}_j \backslash \mathbf{X}_{jk}$, $p_0$ can be blocked by $\mathbf{X}_{jk}$. By Lemma C.4, $\mathbf{X}_{jk} \backslash \mathbf{X}_{ij} \cap \mathbf{X}_{jk}$ (*i.e.*, the subset $\mathbf{B}$ in Fig. 9 (b)) must contain a non-collider $X_i, i \leq m - 1$ in $p$. So, there is a path $p_1 :< Y, X_1, X_2, ..., X_i >$ between $Y$ and $\mathbf{X}_{jk} \backslash \mathbf{X}_{ij} \cap \mathbf{X}_{jk}$. Be Lemma C.4, as $p_1$ is a sub-path of $p_0$, $p_1$ can not be blocked by $\mathbf{X}_{ij} \cap \mathbf{X}_{jk}$ neither. Because $X_i \in \mathbf{B} \subseteq \mathbf{X}_j \backslash \mathbf{X}_{ij}$, $p_1$ can be blocked by $\mathbf{X}_{ij}$, so again, $\mathbf{X}_{ij} \backslash \mathbf{X}_{ij} \cap \mathbf{X}_{jk}$ (*i.e.*, the subset $\mathbf{A}$ in Fig. 9 (b)) must contain a non-collider in $p_1$. Iterating like this, we have $X_1 \in \mathbf{A}$ or $X_1 \in \mathbf{B}$. Because $X_1$ is adjacent with $Y$, we have either $Y \perp_G \mathbf{X}_j \backslash \mathbf{X}_{jk} | \mathbf{X}_{jk}$ or $Y \perp_G \mathbf{X}_j \backslash \mathbf{X}_{ij} | \mathbf{X}_{ij}$ not hold.

This contradiction means any path between $Y$ and $\mathbf{A}$ (similarly $\mathbf{B}$) can be blocked by $\mathbf{X}_{ij} \cap \mathbf{X}_{jk}$. Formally speaking, $Y \perp_G \mathbf{X}_{ij} \cup \mathbf{X}_{jk} \backslash \mathbf{X}_{ij} \cap \mathbf{X}_{jk} | \mathbf{X}_{ij} \cap \mathbf{X}_{jk}$.

Define $\mathbf{C} := \mathbf{X}_{ij} \cup \mathbf{X}_{jk} \backslash \mathbf{X}_{ij} \cap \mathbf{X}_{jk}$ and $\mathbf{D} := \mathbf{X}_i \cup \mathbf{X}_j \cup \mathbf{X}_k \backslash \mathbf{X}_{ij} \cup \mathbf{X}_{jk}$, as shown by Fig. 9 (c). We have already shown that any path between $Y$ and $\mathbf{C}$ can be blocked by $\mathbf{X}_{ij} \cap \mathbf{X}_{jk}$, we in the following show any path between $Y$ and $\mathbf{D}$ can also be blocked by $\mathbf{X}_{ij} \cap \mathbf{X}_{jk}$.

Again, we prove this by contradiction, *i.e.*, assume there is a path $p_0 :< Y, X_1, X_2, ..., X_m >$ between $Y$ and $\mathbf{D}$ such that $p_0$ can not be blocked by $\mathbf{X}_{ij} \cap \mathbf{X}_{jk}$. Because either $X_m \in \mathbf{X}_i \cup \mathbf{X}_j \backslash \mathbf{X}_{ij}$ or $X_m \in \mathbf{X}_j \cup \mathbf{X}_k \backslash \mathbf{X}_{jk}$, we have $p_0$ can be blocked by $\mathbf{X}_{ij}$ or $\mathbf{X}_{jk}$. Let's assume $p_0$ can be blocked by $\mathbf{X}_{ij}$ (as by $\mathbf{X}_{jk}$ has a similar analysis). By Lemma C.4, $\mathbf{X}_{ij} \backslash \mathbf{X}_{ij} \cap \mathbf{X}_{jk}$ must contain a non-collider $X_i$ in $p_0$. So, we have a path $p_1 :< Y, X_1, X_2, ..X_i >$ between $Y$ and $\mathbf{X}_{ij} \backslash \mathbf{X}_{ij} \cap \mathbf{X}_{jk}$. Because $p_1$ is a sub-path of $p_0$, by Lemma C.4, $p_1$ can not be blocked by $\mathbf{X}_{ij} \cap \mathbf{X}_{jk}$ either, which contradict with $Y \perp_G \mathbf{X}_{ij} \cup \mathbf{X}_{jk} \backslash \mathbf{X}_{ij} \cap \mathbf{X}_{jk} | \mathbf{X}_{ij} \cap \mathbf{X}_{jk}$.

This contradiction means any path between $Y$ and $\mathbf{D}$ can also be blocked by $\mathbf{X}_{ij} \cap \mathbf{X}_{jk}$.

In conclusion, any path between $Y$ and $\mathbf{X}_i \cup \mathbf{X}_j \cup \mathbf{X}_k \backslash \mathbf{X}_{ij} \cap \mathbf{X}_{jk}$ can be blocked by $\mathbf{X}_{ij} \cap \mathbf{X}_{jk}$. Formally speaking, $Y \perp_G \mathbf{X}_i \cup \mathbf{X}_j \cup \mathbf{X}_k \backslash \mathbf{X}_{ij} \cap \mathbf{X}_{jk} | \mathbf{X}_{ij} \cap \mathbf{X}_{jk}$. Because $\mathbf{X}_i \cup \mathbf{X}_k \subseteq \mathbf{X}_i \cup \mathbf{X}_j \cup \mathbf{X}_k$ and $\mathbf{X}_{ij} \cap \mathbf{X}_{jk} \subseteq \mathbf{X}_i \cap \mathbf{X}_k$, we have $\mathbf{X}_i \sim_G \mathbf{X}_k$. □

To conclude, we have shown that $\sim_G$ satisfies reflexivity, symmetry, and transitivity. So, it is also a legal equivalent relation.

Recall the Markovian assumption states for any disjoint sets $\mathbf{X}_i, \mathbf{X}_j, \mathbf{X}_k$, we have $\mathbf{X}_i \perp_G \mathbf{X}_j | \mathbf{X}_k \Rightarrow \mathbf{X}_i \perp \mathbf{X}_j | \mathbf{X}_k$. It builds a bridge from $d$-separation in graph to conditional independence in probability. As a result, under this assumption, we can infer two subsets are equivalent in predicting $Y$ if they are graphical equivalent in the intervened graph $G_{\overline{\mathbf{X}_M}}$[3]. Formally speaking,

**Proposition C.5.** *For two subsets $\mathbf{X}_i$ and $\mathbf{X}_j$ of the stable set, if $\mathbf{X}_i \sim_G \mathbf{X}_j$, then $\mathbf{X}_i \sim_P \mathbf{X}_j$.*

---

[3]The intervened graph means the graph after removing all edges into $\mathbf{X}_M$.

*Remark* C.6. Note that the reverse claim $\mathbf{X}_i \sim_P \mathbf{X}_j \Rightarrow \mathbf{X}_i \sim_G \mathbf{X}_j$ is not true even under the faithfulness assumption. Consider the counter example of $Y, X_1, X_2$, and the structural equations $X_1 \leftarrow Y + N(0,1), X_2 \leftarrow Y + N(0,1)$. We have $\{X_1\} \sim_P \{X_2\}$, but do not have $\{X_1\} \sim_G \{X_2\}$.

Now that we know the definition of two subsets being equivalent and how to infer the equivalence from causal graph, we are ready to introduce the notion of *g*-equivalent class. Denote the power set of the stable variables $\mathbf{X}_S$ as $\mathrm{Pow}(\mathbf{X}_S)$, then elements of the quotient spaces $\mathrm{Pow}(\mathbf{X}_S)/\sim_G$ are called *g*-equivalent classes. Since all predictors in the same *g*-equivalent class have the same power in predicting $Y$, the searching for optimal predictor in Alg. 1 should be conducted among *g*-equivalent classes.

In the following, we introduce an algorithm to recover the $\mathrm{Pow}(\mathbf{X}_S)/\sim_G$ space. The algorithm takes the stable graph[4] $G_S$ as input and recursively explore stable variables in the order of their distance to $Y$. In each step of exploration, it create sub-graphs to represent conditional independence after including/excluding some stable subsets. We use the maximal ancestral graph (MAG) to construct these sub-graphs, thanks to its ability to preserve conditional independence when included (selection) or excluded (latent) variables exist. In the following, we omit the subscript $S$ in $G_S$ and $\mathbf{X}_S$ for brevity.

---

**Algorithm 6** $\mathbf{P}_G = \mathrm{Recover}(G)$

---

**Input:** a causal graph $G$.
**Output:** the set of all *g*-equivalent classes $\mathbf{P}_G$.

 1: Let $\mathbf{X}$ the covariate set of $G$
 2: Find $\mathrm{Neig}_G(Y)$
 3: **if** $\mathrm{Neig}_G(Y) = \varnothing$ **then**
 4:     **return** $\{\mathrm{Pow}(\mathbf{X})\}$
 5: **else**
 6:     $\mathbf{P}_G \leftarrow \{\}$
 7:     **for** $\mathbf{T}$ in $\mathrm{Pow}(\mathrm{Neig}_G(Y))$ **do**
 8:         $\mathbf{S} \leftarrow \mathbf{T}, \mathbf{L} \leftarrow \mathrm{Neig}_G(Y)\backslash\mathbf{T}, \mathbf{O} \leftarrow \mathbf{X}\backslash\mathrm{Neig}_G(Y)$
 9:         $G' \leftarrow \mathrm{MAG}(G, \mathbf{O}, \mathbf{L}, \mathbf{S})$
10:         $\mathbf{P}_{G'} \leftarrow \mathrm{Recover}(G')$
11:         **for** $[\mathbf{X}_i]$ in $\mathbf{P}_{G'}$ **do**
12:             **for** $\mathbf{X}_j$ in $[\mathbf{X}_i]$ **do**
13:                 $\mathbf{X}_j \leftarrow \mathbf{X}_j \cup \mathbf{T}$
14:             **end for**
15:         **end for**
16:         $\mathbf{P}_G \leftarrow \mathbf{P}_G \cup \mathbf{P}_{G'}$
17:     **end for**
18:     **return** $\mathbf{P}_G$
19: **end if**

---

**Algorithm 7** $G' = \mathrm{MAG}(G, \mathbf{O}, \mathbf{L}, \mathbf{S})$

---

**Input:** a causal graph $G$ over $\mathbf{X} = \mathbf{O} \cup \mathbf{L} \cup \mathbf{S}$.
**Output:** a causal graph $G'$ over $\mathbf{O}$.

 1: for each pair of variables $A, B \in \mathbf{O}$, $A$ and $B$ are adjacent in $G'$ if and only if there is an inducing path relative to $< \mathbf{L}, \mathbf{S} >$ between them in $G$.
 2: for each pair of adjacent vertices $A, B$ in $G'$, orient the edge between them as follows:
 3: (a) orient it as $A \rightarrow B$ in $G'$ if $A \in \mathrm{Anc}_G(B \cup \mathbf{S})$ and $B \notin \mathrm{Anc}_G(A \cup \mathbf{S})$;
 4: (b) orient it as $B \rightarrow A$ in $G'$ if $A \notin \mathrm{Anc}_G(B \cup \mathbf{S})$ and $B \in \mathrm{Anc}_G(A \cup \mathbf{S})$;
 5: (c) orient it as $A \leftrightarrow B$ in $G'$ if $A \notin \mathrm{Anc}_G(B \cup \mathbf{S})$ and $B \notin \mathrm{Anc}_G(A \cup \mathbf{S})$;
 6: (d) orient it as $A - B$ in $G'$ if $A \in \mathrm{Anc}_G(B \cup \mathbf{S})$ and $B \in \mathrm{Anc}_G(A \cup \mathbf{S})$;

---

**Proposition C.7.** *Alg. 6 outputs the correct* g-*equivalent classes in causal graph* $G$.

---

[4]The stable graph means the graph after removing all vertex in $\mathbf{X}_M$

*Proof.* We firstly introduce some notions that will be used in the proof. Denote $[\mathbf{X}_i] := \{\mathbf{X}_j | \mathbf{X}_j \sim_G \mathbf{X}_i\}$ the equivalent class with representative element $\mathbf{X}_i$. Denote the set of all equivalent classes as $\mathrm{Pow}(\mathbf{X})/\sim_G$. Define length of a path the number of edges in it. In a causal graph $G$, we say $X_i$ is $Y$'s $w$-order neighbour if the shortest path between $Y$ and $X_i$ has length $w$. As a special case, $X_i$ is called 0-order neighbour of $Y$ if there is no path between $Y$ and $X_i$. Let $\Omega(G) = 0$ if $Y$ does not have any neighbour, let $\Omega(G) = 1, 2, 3, ...$ if $Y$ has $1, 2, 3, ...$-order neighbour, respectively.

Note that if we construct a MAG $G'$ over $\mathbf{O}$ by $G' = \mathrm{MAG}(G, \mathbf{O}, \mathbf{L}, \mathbf{S})$, then, for any vertices sets $\mathbf{V}_i, \mathbf{V}_j, \mathbf{V}_k \subseteq \mathbf{O}$, we have $\mathbf{V}_i \perp_G \mathbf{V}_j | \mathbf{V}_k, \mathbf{S} \Leftrightarrow \mathbf{V}_i \perp_{G'} \mathbf{V}_j | \mathbf{V}_k$. The proof is available at Sect. 2.3 in Zhang (2008).

In the following, we prove the proposition by induction on $\Omega(G)$.

**Base**. For any causal graph $G$ with $\Omega(G) = 0$, we have $\mathrm{Neig}(Y) = \varnothing$. So, for any $\mathbf{X}_i, \mathbf{X}_j \subseteq \mathbf{X}$, we have $\varnothing \subseteq \mathbf{X}_i \cap \mathbf{X}_j$ such that $Y \perp_G \mathbf{X}_i \cap \mathbf{X}_j \backslash \varnothing | \varnothing$, which means $\mathbf{X}_i \sim_G \mathbf{X}_j$. This means $\mathrm{Pow}(\mathbf{X})/\sim_G = \{[\mathbf{X}]\}$, *i.e.*, all subsets of the covariate set are equivalent and there is only one equivalent class.

**Induction Hypotheses**. Assume any causal graph $G_{\leq w}$ with $\Omega(G) \leq w$, $\mathrm{Pow}(\mathbf{X})/\sim_{G_{\leq w}} = \mathrm{Recover}(G_{\leq w})$.

**Step**. In the following, we show any causal graph $G_{w+1}$ with $\Omega(G) = w + 1$, we have $\mathrm{Pow}(\mathbf{X})/\sim_{G_{w+1}} = \mathrm{Recover}(G_{w+1})$.

Denote covariates in $\mathrm{Neig}_{G_{w+1}}(Y)$ as $\{X_1^1, X_2^1, ..., X_{n_1}^1\}$. Denote the power set $\mathrm{Pow}(\{X_1^1, X_2^1, ..., X_{n_1}^1\})$ as $\mathbf{Q} := \{\mathbf{Q}_1, \mathbf{Q}_2, ..., \mathbf{Q}_{2^{n_1}}\}$, where $\mathbf{Q}_1 = \varnothing$, $\mathbf{Q}_2 = \{X_1^1\}$, ..., $\mathbf{Q}_{2^{n_1}} = \{X_1^1, X_2^1, ..., X_{n_1}^1\}$.

Any $\mathbf{X}_i \subseteq \mathbf{X}$ can be written as $\mathbf{X}_i = \mathbf{X}_i^1 \cup \mathbf{X}_i^{\mathrm{other}}$, where $\mathbf{X}_i^1$ contains all 1-order covariates and $\mathbf{X}_i^{\mathrm{other}}$ contains the others. So, $\mathrm{Pow}(\mathbf{X})$ can be partitioned into $2^{n_1}$ sets $\{\mathbf{R}_1, \mathbf{R}_2, ..., \mathbf{R}_{2^{n_1}}\}$, where $\mathbf{R}_i := \{\mathbf{X}_i | \mathbf{X}_i^1 = \mathbf{Q}_i\}$.

Now consider an element $\mathbf{X}_i \in \mathbf{R}_i$ and an element $\mathbf{X}_j \in \mathbf{R}_j$, $i \neq j$. Because $\mathbf{X}_i^1 \neq \mathbf{X}_j^1$ and $\mathbf{X}_i^1, \mathbf{X}_j^1$ are connected with $Y$ in the causal graph, we have $\mathbf{X}_i \not\sim_G \mathbf{X}_j$. This property means $\mathrm{Pow}(\mathbf{X})/\sim_{G_{w+1}} = \cup_{i=1}^{2^{n_1}} \mathbf{R}_i/\sim_{G_{w+1}}$.

By the aforementioned property of MAG Zhang (2008), if we construct $G_i'$ with $\Omega(G_i') \leq w$ by $G_i' := \mathrm{MAG}(G_{w+1}, \mathbf{O} = \mathbf{X} \backslash \mathrm{Neig}_{\mathbf{G_w+1}}(\mathbf{Y}), \mathbf{L} = \mathrm{Neig}_{G_{w+1}}(Y) \backslash \mathbf{Q}_i, \mathbf{S} = \mathbf{Q}_i)$, then we have $\mathbf{R}_i/\sim_{G_{w+1}} = \mathbf{R}_i/\sim_{G_i'}$. So, we further have $\mathrm{Pow}(\mathbf{X})/\sim_{G_{w+1}} = \cup_{i=1}^{2^{n_1}} \mathbf{R}_i/\sim_{G_{w+1}} = \cup_{i=1}^{2^{n_1}} \mathbf{R}_i/\sim_{G_i'}$.

By the induction hypotheses, $\mathbf{R}_i/\sim_{G_i'} = \mathrm{Recover}(G_i')$. So, the above equation can be further written as $\mathrm{Pow}(\mathbf{X})/\sim_{G_{w+1}} = \cup_{i=1}^{2^{n_1}} \mathbf{R}_i/\sim_{G_{w+1}} = \cup_{i=1}^{2^{n_1}} \mathbf{R}_i/\sim_{G_i'} = \cup_{i=1}^{2^{n_1}} \mathrm{Recover}(G_i')$. By design of Alg. 6 (line-18), we have $\cup_{i=1}^{2^{n_1}} \mathrm{Recover}(G_i') = \mathrm{Recover}(G_{w+1})$. So, we eventually have $\mathrm{Pow}(\mathbf{X})/\sim_{G_{w+1}} = \mathrm{Recover}(G_{w+1})$.

$\square$

**Corollary C.8** (*g*-equivalent Classes in Sub-graphs). *Denote the causal graph $G$, the covariate set $\mathbf{X}$. Let $\mathbf{Z} \subseteq \mathbf{X}$ a subset of covariates. Denote covariates in $\mathbf{Z}$ as $\{X_1^z, X_2^z, ..., X_l^z\}$, and the power set $\mathrm{Pow}(\{X_1^z, X_2^z, ..., X_l^z\})$ as $\mathbf{Q} := \{\mathbf{Q}_1, \mathbf{Q}_2, ..., \mathbf{Q}_{2^l}\}$, with $\mathbf{Q}_1 = \varnothing, \mathbf{Q}_2 = \{X_1^z\}, ..., \mathbf{Q}_{2^l} = \{X_1^z, X_2^z, ..., X_l^z\}$. Construct $2^l$ sub-graphs $G_i', i = 1, 2, ..., 2^l$ by $G_i' := \mathrm{MAG}(G, \mathbf{S} = \mathbf{Q}_i, \mathbf{L} = \mathbf{Z} \backslash \mathbf{Q}_i, \mathbf{O} = \mathbf{X} \backslash \mathbf{Z})$. Denote the number of $G$-equivalent classes in the causal graph $G$ and sub-graph $G_i'$ as $N_G$, $N_{G_i'}$, respectively. Then, we have $N_G \leq \sum_{i=1}^{2^l} N_{G_i'}$.*

*Proof.* Because we do not restrict the set $\mathbf{Z}$ to $\mathrm{Neig}(Y)$, an element $\mathbf{X}_i$ from $\mathbf{R}_i$ and an element $\mathbf{X}_j$ from $\mathbf{R}_j$ may be graphical equivalent. So, the equal to mark in Prop. C.7 because a greater than or equal to mark. $\square$

Indeed, the true causal DAG with complete orientation is not identifiable. What we can identify is a partially directed acyclic graph (PDAG), representing all Markovian equivalent graphs of the true

DAG. The following proposition, which states all the Markovian equivalent graphs have the same $g$-equivalent classes, assures that our recovery algorithm can be applied to PDAG.

**Proposition C.9.** *Under assumption 2.2, causal graphs in the same Markovian equivalent class have the same* g-*equivalence.*

*Proof.* By definition, causal graphs in the same Markovian equivalent class have the same probability distribution. Under the Markovian and faithfulness assumptions, this means they have the same set of $d$-separations. As $g$-equivalence is defined on $d$-separation, they also have the same $g$-equivalence. □

## C.2 DETAILS OF CAUSAL DISCOVERY TO DETECT LOCAL COMPONENTS

In this section, we summarize our method to detect $\mathbf{X}_M^0$, $\mathrm{De}(\mathbf{X}_M^0)$, $\mathbf{X}_M$, $\mathrm{De}(\mathbf{X}_M)$, $\mathrm{Blanket}(Y)$, $\{\mathrm{PA}(X_i)\}_{X_i \in \mathbf{X}_M \cup \mathrm{De}(\mathbf{x}_M)}$, $\mathrm{PC}(Y) := \mathrm{PA}(Y) \cup \mathrm{Ch}(Y)$. The $\mathrm{Blanket}(Y)$ denotes the Markovian Blanket of $Y$.

The identification of $\mathbf{X}_M^0 \cup \mathrm{De}(\mathbf{X}_M^0)$ are in Alg. 3. To distinguish $\mathbf{X}_M^0$ and $\mathrm{De}(\mathbf{X}_M^0)$, it suffices to identify the direction of $X_i - X_j$ in the case when both $X_i$ and $X_j$ are in $\mathbf{X}_M^0$, which can be accomplished by comparing $\widehat{\Delta}_{X_i \to X_j}$ and $\widehat{\Delta}_{X_j \to X_i}$ (see Huang et al. (2020) for details). However, it should be noted that distinguishing $\mathbf{X}_M^0$ and $\mathrm{De}(\mathbf{X}_M^0)$ for the estimation of $h(S_-, J)$ and $f_{S_-}$ is unnecessary. The identification of $\mathbf{X}_M$ and $\mathrm{PC}(Y)$ is in Alg. 2, where $\mathrm{PC}(Y)$ can be obtained from the undirected skeleton. $\mathrm{Blanket}(Y)$ can be identified by Aliferis et al. (2003). The identification of $\mathbf{X}_M \cup \mathrm{De}(\mathbf{X}_M)$ is in Alg. 4 and we can distinguish $\mathbf{X}_M$ from $\mathrm{De}(\mathbf{X}_M)$ using the way as in $\{\mathbf{X}_M^0, \mathrm{De}(\mathbf{X}_M^0)\}$. The parents $\{\mathrm{PA}(X_i)|X_i \in \mathbf{X}_M \cup \mathrm{De}(\mathbf{X}_M)\}$ can be identified by Alg. 5.

## C.3 DETAILS OF ESTIMATING $f_{S_-}$

To estimate $f_{S_-}$, we adopt soft-intervention to replace $P^e(\mathbf{X}_M|\mathrm{PA}(\mathbf{X}_M))$ with $P(\mathbf{X}_M)$ and hence define $p'(\boldsymbol{x}, y) = p(y|pa(y)) \prod_{i \in S} p(x_i|pa(x_i))p(\mathbf{X}_M)$. Then we have $f_{S_-} = \mathbb{E}_{P'}[Y|\boldsymbol{x}_{S_-}, \mathbf{X}_M]$. To generate data from $P'$, we first permute $\mathbf{X}_M$ in a sample-wise manner to generate data from $P(\mathbf{X}_M)$. We then regenerate data for $\mathbf{X}_M$'s descendants in the intervened graph via estimating structural equations [5], as summarized in Alg. 8.

---

**Algorithm 8** Estimation of $f_{S_-}$.

---

**INPUT:** training data $\{\boldsymbol{x}_{(k)}, y_{(k)}\}_{k=1}^n$, $S_- \subset S$, $\mathbf{X}_M$, $\mathrm{De}(\mathbf{X}_M)$, and $\{\mathrm{PA}(X_i)\}_{X_i \in \mathrm{De}_{G_{\overline{\mathbf{X}_M}}}(\mathbf{X}_M)}$.

**OUTPUT:** Trained $f_{S_-}$.

  1: Shuffling $\{(\mathbf{x}_M)_{(k)}\}_{k=1}^n$ by randomizing the indices.
  2: For $X_i \in \mathrm{De}_{G_{\overline{\mathbf{X}_M}}}(\mathbf{X}_M)$ do
  3:     Regenerate $\{(x_i)_{(k)}\}_{k=1}^n$ as $\{g_i(pa(x_i)_{(k)})\}_{k=1}^n$.
  4: Train $f_{S_-}$ over the regenerated samples.

---

Indeed, we only need to regenerate $\mathrm{De}_{G_{\overline{\mathbf{X}_M}}}(\mathbf{X}_M) \cap \mathrm{Blanket}(Y)$ since $p'(y|\mathrm{blanket}(y)) = p'(y|\boldsymbol{x})$. To maximally reduce the approximation error in regeneration, we consider intervene on another variable set $X_{do}^* := \mathbf{X}_M^0 \cup (\mathrm{De}(\mathbf{X}_M^0) \setminus \mathrm{Ch}(Y))$ and regenerate variables in $\mathrm{De}_{G_{\overline{\mathbf{x}_{do}^*}}}(\mathbf{X}_{do}^*)$. We prove $\mathrm{De}_{G_{\overline{\mathbf{X}_{do}^*}}}(\mathbf{X}_{do}^*)$ is the minimum regeneration set in the following proposition.

**Proposition C.10.** *Denote* $X_{do}^* := \mathbf{X}_M^0 \cup (\mathrm{De}(\mathbf{X}_M^0) \setminus \mathrm{Ch}(Y))$. *Then:*

  *1. For any admissible set* $\mathbf{X}_{do}$*, we have* $\mathrm{De}_{G_{\overline{\mathbf{X}_{do}}}}(\mathbf{X}_{do}) \cap \mathrm{Blanket}(Y) \supset \mathrm{De}_{G_{\overline{\mathbf{x}_{do}^*}}}(\mathbf{X}_{do}^*)$;

  *2.* $\mathbf{X}_{do}^*$*,* $\mathrm{De}_{G_{\overline{\mathbf{X}_{do}^*}}}(\mathbf{X}_{do}^*)$*, and* $\{\mathrm{PA}(X_i)\}_{X_i \in \mathrm{De}_{G_{\overline{\mathbf{X}_{do}^*}}}(\mathbf{X}_{do}^*)}$ *are identifiable.*

*Proof.* (1) Firstly, we prove that a set of variables $\mathbf{X}_{do}$ is admissible means $p_{do}(y|\boldsymbol{x}) = p(y|\boldsymbol{x}_S, do(\mathbf{X}_M)) \Leftrightarrow \{\mathbf{X}_M \cap \mathrm{Ch}(Y)\} \subset \mathbf{X}_{do}$ and $\{\mathbf{X}_S \cap \mathrm{Ch}(Y)\} \cap \mathbf{X}_{do} = \varnothing$.

---

[5]This can be achieved because $\mathbf{X}_M$, $\mathrm{De}(\mathbf{X}_M)$, and their parents are identifiable, as shown in Alg. 5.

Note that

$$
\begin{aligned}
p(y|\boldsymbol{x}_S, do(\boldsymbol{x}_M)) &= \frac{p(y|pa(y)) \prod_{X_i \in \mathbf{X}_S \cap \mathrm{Ch}(Y)} p(x_i|pa(x_i))}{\int_y p(y|pa(y)) \prod_{X_i \in \mathbf{X}_S \cap \mathrm{Ch}(Y)} p(x_i|pa(x_i)) dy}, \\
p_{do}(y|\boldsymbol{x}) &= \frac{p(y|pa(y)) \prod_{X_i \in \{\mathbf{X} \setminus \mathbf{X}_{do}\} \cap \mathrm{Ch}(Y)} p(x_i|pa(x_i))}{\int_y p(y|pa(y)) \prod_{X_i \in \{\mathbf{X} \setminus \mathbf{X}_{do}\} \cap \mathrm{Ch}(Y)} p(x_i|pa(x_i)) dy}.
\end{aligned}
$$

It can be seen $p_{do}(y|\boldsymbol{x}) = p(y|\boldsymbol{x}_S, do(\mathbf{X}_M)) \Leftrightarrow \mathbf{X} \setminus \mathbf{X}_{do} \cap \mathrm{Ch}(Y) = \mathbf{X}_S \cap \mathrm{Ch}(Y)$, which can be rewritten as

$$
\{\mathbf{X}_M \cap \mathrm{Ch}(Y) \cap \mathbf{X}_{do}^C\} \cup \{\mathbf{X}_S \cap \mathrm{Ch}(Y) \cap \mathbf{X}_{do}^C\} = \mathbf{X}_S \cap \mathrm{Ch}(Y).
$$

The above equation holds if and only if $\{\mathbf{X}_M \cap \mathrm{Ch}(Y)\} \subset \mathbf{X}_{do}$ and $\{\mathbf{X}_S \cap \mathrm{Ch}(Y)\} \cap \mathbf{X}_{do} = \varnothing$.

(2) Secondly, we prove that $\mathbf{X}_{do}^*$ is an admissible set and $\mathrm{De}_{G_{\overline{\mathbf{X}_{do}^*}}}(\mathbf{X}_{do}^*) = \mathrm{De}(\mathbf{X}_M \cap \mathrm{Ch}(Y)) \cap \mathbf{X}_S \cap \mathrm{Ch}(Y)$. To simplify the notations, let $\mathbf{X}_0 := \mathbf{X}_{do}^*$ and $\mathbf{X}_1 := \mathrm{De}_{G_{\overline{\mathbf{X}_{do}^*}}}(\mathbf{X}_{do}^*)$.

The conditions $\{\mathbf{X}_M \cap \mathrm{Ch}(Y)\} \subset \mathbf{X}_0$ and $\{\mathbf{X}_S \cap \mathrm{Ch}(Y)\} \cap \mathbf{X}_0 = \varnothing$ hold by definition.

(2.1) show $\mathrm{De}_{G_{\overline{\mathbf{X}}_0}}(\mathbf{X}_0) \subset \mathbf{X}_1$

Note $\mathbf{X}_0 \subset \{\mathbf{X}_M \cap \mathrm{Ch}(Y)\} \cup \{\mathrm{De}(\mathbf{X}_M \cap \mathrm{Ch}(Y))\}$, we have $\mathrm{De}(\mathbf{X}_0) \subset \mathrm{De}(\mathbf{X}_M \cap \mathrm{Ch}(Y))$. Besides, since $\mathrm{De}_{G_{\overline{\mathbf{X}}_0}}(\mathbf{X}_0) = \mathrm{De}(\mathbf{X}_0) \setminus \mathbf{X}_0$, Then

$$
\begin{aligned}
\mathrm{De}_{G_{\overline{\mathbf{X}}_0}}(\mathbf{X}_0) &= \mathrm{De}(\mathbf{X}_0) \cap \mathbf{X}_0^C = \mathrm{De}(\mathbf{X}_0) \cap \{\mathbf{X}_M \cap \mathrm{Ch}(Y)\}^C \cap \{\mathrm{De}(\mathbf{X}_M \cap \mathrm{Ch}(Y)) \setminus \mathrm{Ch}(Y)\}^C \\
&= \mathrm{De}(\mathbf{X}_0) \cap \{\mathbf{X}_M^C \cup \mathrm{Ch}(Y)\}\} \cap \{\mathrm{De}(\mathbf{X}_M \cap \mathrm{Ch}(Y))^C \cup \mathrm{Ch}(Y)\}\} \\
&\subset \{\mathrm{De}(\mathbf{X}_M \cap \mathrm{Ch}(Y))\} \cap \{\mathbf{X}_M^C \cup \mathrm{Ch}(Y)^C\}\} \cap \{\mathrm{De}(\mathbf{X}_M \cap \mathrm{Ch}(Y))^C \cup \mathrm{Ch}(Y)\}\} \\
&= \mathrm{De}(\mathbf{X}_M \cap \mathrm{Ch}(Y)) \cap \mathbf{X}_M^C \cap \mathrm{Ch}(Y) = \mathrm{De}(\mathbf{X}_M \cap \mathrm{Ch}(Y)) \cap \mathbf{X}_S \cap \mathrm{Ch}(Y) \subset \mathbf{X}_1
\end{aligned}
$$

(2.2) show $\mathbf{X}_1 \subset \mathrm{De}_{G_{\overline{\mathbf{X}}_0}}(\mathbf{X}_0)$

Since $\mathbf{X}_M \cap \mathrm{Ch}(Y) \subset \mathbf{X}_0$, $\mathrm{De}(\mathbf{X}_M \cap \mathrm{Ch}(Y)) \subset \mathrm{De}(\mathbf{X}_0)$, so $\mathbf{X}_1 \subset \mathrm{De}(\mathbf{X}_M \cap \mathrm{Ch}(Y)) \subset \mathrm{De}(\mathbf{X}_0)$ and hence $\mathbf{X}_1 \setminus \mathbf{X}_0 \subset \mathrm{De}(\mathbf{X}_0) \setminus \mathbf{X}_0$. Besides, note that $\mathbf{X}_0 \cap \mathbf{X}_1 = \varnothing$ such that $\mathbf{X}_1 \setminus \mathbf{X}_0 = \mathbf{X}_0$ and $\mathrm{De}_{G_{\overline{\mathbf{X}}_0}}(\mathbf{X}_0) = \mathrm{De}(\mathbf{X}_0) \setminus \mathbf{X}_1$, we have $\mathbf{X}_1 \subset \mathrm{De}_{G_{\overline{\mathbf{X}}_0}}(\mathbf{X}_0)$.

(3) given $\mathbf{X}_{do}$ satisfying the two conditions, we have

$$
\begin{aligned}
\mathbf{X}_M \cap \mathrm{Ch}(Y) \subset \mathbf{X}_{do} &\Rightarrow \mathrm{De}(\mathbf{X}_M \cap \mathrm{Ch}(Y)) \subset \mathrm{De}(\mathbf{X}_{do}); \\
\mathbf{X}_{do} \subset \{\mathbf{X}_S \cap \mathrm{Ch}(Y)\}^C &\Rightarrow \{\mathbf{X}_S \cap \mathrm{Ch}(Y)\} \subset \mathbf{X}_{do}^C
\end{aligned} .
$$

Therefore,

$$
\mathrm{De}(\mathbf{X}_M \cap \mathrm{Ch}(Y)) \cap \{\mathbf{X}_S \cap \mathrm{Ch}(Y)\} \subset \mathrm{De}(\mathbf{X}_{do}) \cap \mathbf{X}_{do}^C,
$$

Thus, $\mathbf{X}_1 \subset \mathrm{De}_{G_{\overline{\mathbf{X}}_{do}}}(\mathbf{X}_{do})$ for any $\mathbf{X}_{do}$ satisfying $\mathbf{X}_M \cap \mathrm{Ch}(Y) \subset \mathbf{X}_{do}$ and $\mathbf{X}_{do} \cap \{\mathbf{X}_S \cap \mathrm{Ch}(Y)\} = \varnothing$.

(4) The identification of $\{\mathrm{PA}(X_i)\}_{X_i \in \mathrm{De}_{G_{\overline{\mathbf{X}_{do}^*}}}(\mathbf{X}_{do}^*)}$, $\mathbf{X}_{do}^*$ and $\mathrm{De}_{G_{\overline{\mathbf{X}_{do}^*}}}(\mathbf{X}_{do}^*)$ can be readily obtained in Sec. C.2. □

## D   APPENDIX FOR SEC. 3.3: COMPLEXITY ANALYSIS

In this section, we given some graphical examples and show the number of $g$-equivalent classes (denoted by $N_G$) over them.

**Lemma D.1** (Adding/Deleting Edges). *In a causal graph G, adding edges does not decrease $N_G$, deleting edges does not increase $N_G$.*

*Proof.* **1**. For any causal graph $G_0$, add an edge in it and call the resulted graph as $G_1$. We show $N_{G_0} \leq N_{G_1}$. We show this by proving for any subsets $\mathbf{X}_i, \mathbf{X}_j$, if $\mathbf{X}_i \not\sim_{G_0} \mathbf{X}_j$, then $\mathbf{X}_i \not\sim_{G_1} \mathbf{X}_j$.

We prove this by contradiction. Suppose there are $\mathbf{X}_i, \mathbf{X}_j$ such that $\mathbf{X}_i \not\sim_{G_0} \mathbf{X}_j$ and $\mathbf{X}_i \sim_{G_1} \mathbf{X}_j$. By $\mathbf{X}_i \sim_{G_1} \mathbf{X}_j$, we have $\exists \mathbf{X}_{ij} \subseteq_{G_1} \mathbf{X}_i \cap \mathbf{X}_j$, $Y \perp_{G_1} \mathbf{X}_i \cup \mathbf{X}_j \backslash \mathbf{X}_{ij} | \mathbf{X}_{ij}$. Because adding an edge does not change the covariate sets, we have $\mathbf{X}_{ij} \subseteq_{G_0} \mathbf{X}_i \cap \mathbf{X}_j$. Because $\mathbf{X}_i \not\sim_{G_0} \mathbf{X}_j$, we have $Y \not\perp_{G_0} \mathbf{X}_i \cup \mathbf{X}_j \backslash \mathbf{X}_{ij} | \mathbf{X}_{ij}$. In other word, there is a path $p$ in $G_0$ between $Y$ and $\mathbf{X}_i \cup \mathbf{X}_j \backslash \mathbf{X}_{ij}$ such that $p$ can not be blocked by $\mathbf{X}_{ij}$.

This means $\mathbf{X}_{ij}$ does not contain any non-collider on $p$, and $\mathbf{X}_{ij}$ contains every collider (or its descendants) on $p$ in $G_0$. Because in $G_1$, $p$ is still a path between $Y$ and $\mathbf{X}_i \cup \mathbf{X}_j \backslash \mathbf{X}_{ij}$. Besides, any collider $X_c$ on $p$ in $G_0$ is still a collider on $p$ in $G_1$. Any variable $X_d \in \mathrm{Dec}(X_c)$, where $X_c$ is a collider on $p$ in $G_0$, is still a descendant of the collider on $p$ in $G_1$. Any non-collider $X_n$ on $p$ in $G_0$ is still a non-collider on $p$ in $G_1$. We have the path $p$ can not be blocked by $\mathbf{X}_{ij}$ in $G_1$, neither, which contradicts with the claim that $Y \perp_{G_1} \mathbf{X}_i \cup \mathbf{X}_j \backslash \mathbf{X}_{ij} | \mathbf{X}_{ij}$.

**2**. For any causal graph $G_1$, delete an edge in it and call the resulted graph $G_0$. It is straight forward to prove $N_{G_1} \geq N_{G_0}$ using the conclusion from **1**. $\qquad\square$

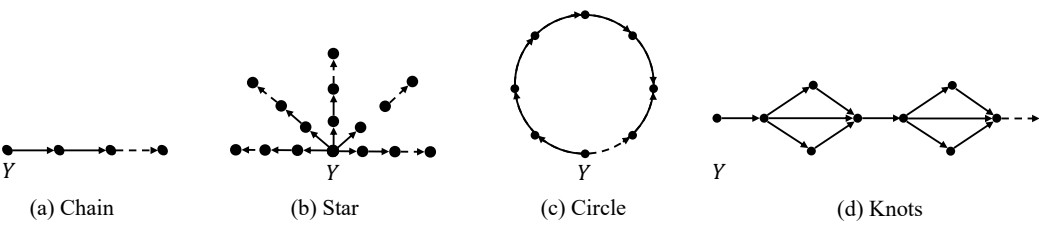

(a) Chain    (b) Star    (c) Circle    (d) Knots

Figure 10

**Example 2** (Chain). A chain graph is a graph whose skeleton is a chain, *i.e.*, $Y - X_n - X_{n-1} - ... - X_1$. For any chain graph with $n$ covariates, $N_{G_n} = n + 1$.

*Proof.* We prove the claim by induction.

**Base**. When $n = 1$, $N_{G_1} = 2 = n + 1$ holds.

**Induction Hypotheses**. Suppose for chain graphs with $n$ covariates, the $N_{G_n} = n + 1$.

**Step**. When there is $n + 1$ covariates in the chain graph, *i.e.*, $G_{n+1}$ has a skeleton $Y - X_{n+1} - X_n - X_{n-1} - ... - X_1$. As $\mathrm{Neig}_{G_{n+1}}(Y) = \{X_{n+1}\}$, we need to discuss the number of $g$-equivalent classes when including and excluding $X_{n+1}$. If $X_{n+1}$ is a collider, then, when including $X_{n+1}$, the induced MAG $G'$ has a skeleton $Y - X_n - X_{n-1} - ... - X_1$; when excluding $X_{n+1}$, skeleton of the induce MAG becomes $Y$ $X_n - X_{n-1} - ... - X_1$. By induction hypotheses, $N_G = n + 1 + 1 = n + 2$ holds. We can have similar conclusion if $X_{n+1}$ is a non-collider, . $\qquad\square$

**Example 3** (Star). A star graph with $k$-branches is a graph whose skeleton is composite of $k$ disjoint chains. For any star graph with $k$-branches and $n$ covariates, $N_{G_n} = O(n^k)$.

*Proof.* A star graph is composite of $k$ disjoint chain graphs, each containing $\frac{n}{k}$ covariates. So, we have $N_G = (\frac{n}{k} + 1)^k = O(n^k)$. $\qquad\square$

**Example 4** (Circle). A circle graph is a graph whose skeleton is a circle, *i.e.*, $Y - X_n - X_{n-1} - ... - X_1$, $Y - X_1$. For any circle graph with $n$ covariates, $N_{G_n} = O(n^2)$.

*Proof.* If $X_n$ is a collider, then, when including $X_n$, the induced MAG will have a skeleton $Y - X_{n-1} - ... - X_1$, $Y - X_1$, which is eventually a circle with $n - 1$ covariates; when excluding $X_{n+1}$, skeleton of the induced MAG becomes a chain with $n - 1$ covariates $Y - X_1 - X_2 - ... - X_{n-1}$. So,

we have $N_{G_n} = n + N_{G'_{n-1}}$, which means $\{N_{G_n}\}_n$ is an arithmetic sequence w.r.t. $n$. According to the summation formula of arithmetic sequence, we have $N_{G_n} = O(n^2)$. $\qquad\square$

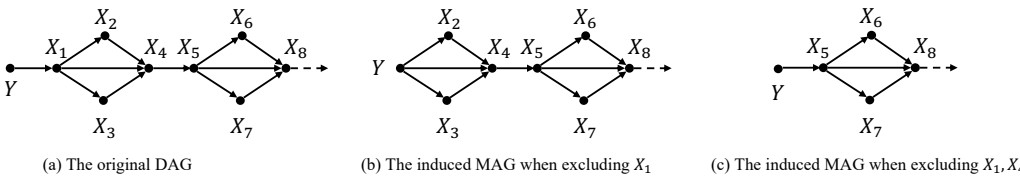

| (a) The original DAG | (b) The induced MAG when excluding $X_1$ | (c) The induced MAG when excluding $X_1, X_4$ |

Figure 11: Recovering $g$-equivalence in knot graph.

**Example 5** (Knots). A knots graph (shown in Fig. 11 (a)) is a generalized directed chain graphs, where each knot contains $4$ covariates. For a knot graph with $n$ covariates, we have $N_{G_n} = O(c^n)$, for some constant $1 < c < 2$.

*Proof.* We prove the claim by showing the recursion formula of $N_G$ w.r.t. the knot number $k$.

For the knot graph shown in Fig. 11 (a), the only neighbour of $Y$ is $X_1$. As $X_1$ is a non-collider, when including $X_1$, $Y$ will not have any neighbour in the induced MAG, so, the number of G-equivalent classes in this sub-graph will be 1. When $X_1$ is excluded, the induced MAG is shown in Fig. 11 (b), where $Y$ is adjacent to three covariates $X_2, X_3, X_4$. As a result, we need to consider $2^3 = 8$ combinations of covariates including/excluding. Out of the $8$ combinations, $4$ of them (including $X_4$) will induce MAGs where $Y$ has no neighbours while the other $4$ combinations (excluding $X_4$) will induce the MAG as shown in Fig. 11 (c), which is eventually a knot graph with $k - 1$ knots.

So, we have the recursion formula of $N_G$ w.r.t. the knot number $k$ to be $N_{G_k} = 1 + (4 + 4 \cdot N_{G_{k-1}})$. This formula means $N_G$ increases exponentially w.r.t. the knot number $k$. Because $k = \frac{n}{4}$, $N_G$ also increases exponentially w.r.t. $n$.

$\qquad\square$

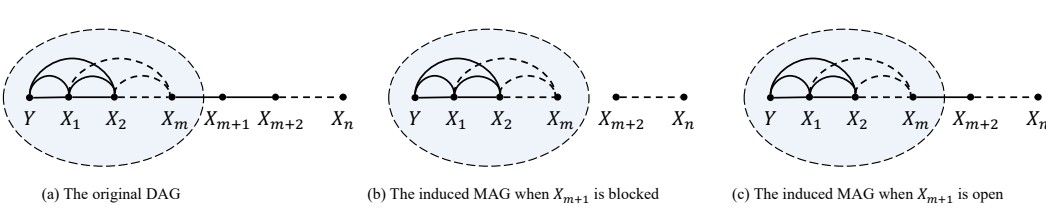

| (a) The original DAG | (b) The induced MAG when $X_{m+1}$ is blocked | (c) The induced MAG when $X_{m+1}$ is open |

Figure 12: Recovering $g$-equivalence in lollipop.

**Example 6** (lollipop). A lollipop graph, as shown in Fig. 12 (a), is constructed by adding edges among $Y$ and the first $m$ covariates in a chain graph. Sugar of the lollipop is made up of $Y, X_1, ..., X_m$, while the stick is $X_{m+1} - X_{m+2} - ... - X_n$. For any lollipop graph with $n$ covariates, $N_{G_n} = O(n)$.

*Proof.* Note that covariates $X_1, X_2, ..., X_m$ in the sugar part may play different roles (non-collider or collider) on different paths through them. So, it can be troublesome to analyze $Y$'s neighbourhood as we did in the chain graph. Fortunately, covariates in the stick only belong to one path, so, we can use them to construct a upper bound of $N_{G_n}$.

Formally speaking, we can construct an upper bound of $N_{G_n}$ with corollary C.8. Specifically, when $X_{m+1}$ is blocked, number of $g$-equivalent classes in the induced MAG will be less than $2^m$; when $X_{m+1}$ is open, the induced MAG $G'$ will be eventually a lollipop with $n - 1$ covariates, as shown in Fig. 12 (c). So, we have the following inequation: $N_{G_n} \le 2^m + N_{G'_{n-1}}$.

Recursively performing the analysis on $G'_{n-1}$, we have $N_{G_n} \le 2^m + 2^m + N_{G''_{n-2}} \le ... \le 2^m + 2^m + ... + 2^m = 2^m(n - m + 1)$. As $m$ is a constant number, we have $N_{G_n} = O(n)$. $\quad\square$

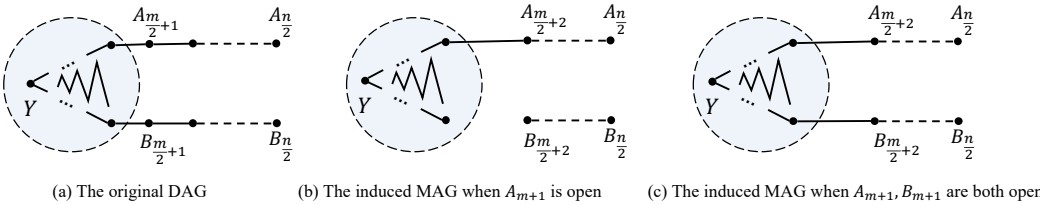

(a) The original DAG     (b) The induced MAG when $A_{m+1}$ is open     (c) The induced MAG when $A_{m+1}, B_{m+1}$ are both open

Figure 13: Recovering $g$-equivalence in $2$-lollipop.

**Example 7** ($k$-lollipop). A lollipop with $k$ sticks is called a $k$-lollipop. For any $k$-lollipop, we have $N_{G_n} = O(n^k)$.

*Proof.* Let's firstly look at the number of $g$-equivalent classes in 2-lollipop, as shown in Fig. 13 (a). Following Example 6, we can construct an upper bound of $N_{G_n}$ by performing Corollary C.8 on $\{A_{m+1}, B_{m+1}\}$. There are $2^2 = 4$ situations, specifically, (i) when $A_{m+1}$ and $B_{m+1}$ are both blocked, number of $g$-equivalent classes in the induced MAG will be less than $2^m$; (ii,iii) when one of covariate in $\{A_{m+1}, B_{m+1}\}$ is blocked, the other is open, the induced MAG will be eventually a $1$-lollipop, as shown in Fig. 13 (b), so the number of equivalent classes in this situation will be bounded by $O(\frac{n}{2})$; (iv) when both $A_{m+1}$ and $B_{m+1}$ are open, the induced MAG $G'$ is a 2-lollipop with $n - 2$ covariates, as shown in Fig. 13 (c). To conclude, we have the following inequation: $N_{G_n} \le 2^m + O(\frac{n}{2}) + N_{G'_{n-2}}$.

Recursively performing the analysis on $G'_{n-2}$, we have $N_{G_n} \le 2^m + O(\frac{n}{2}) + N_{G'_{n-2}} \le 2^m + 2^m + O(\frac{n-2}{2}) + N_{G''_{n-4}} \le ... \le 2^m + ... + 2^m + O(\frac{n}{2}) + O(\frac{n-2}{2}) + ... + O(1) = O(2^m n) + O(n^2)$. As $m$ is a constant number, we have $N_{G_n} = O(n^2)$.

Inspired by this observation, we can analyze $N_{G_n}$ in $k$-lollipop by induction. Formally speaking,

**Base.** For $2$-lollipop, $N_{G_n} = O(n^2)$ holds.

**Induction Hypotheses.** For $k$-lollipop, $N_{G_n} = O(n^k)$ holds

**Step.** For $k+1$-lollipop, we can construct the upper bound of $N_{G_n}$ by performing Corollary C.8 on $\{X_{m+1}^1, ..., X_{m+1}^{k+1}\}$, where $X_{m+1}^i$ is the left-most covariate on the $i$-th stick. There are $2^k$ situations: (i) when at least one of $X_{m+1}^1, ..., X_{m+1}^{k+1}$ is blocked, the induced MAG will be a lollipop with less than or equal to $k$ sticks. According to the induction hypotheses, number of $g$-equivalent classes in these sub-graphs will be at most $O(n^k)$. (ii) When all covariates of $X_{m+1}^1, ..., X_{m+1}^{k+1}$ are open, the induced $G'$ will be eventually a $k+1$-lollipop with $n - k$ covariates. So, we have the following inequation: $N_{G_n} \le O(n^k) + N_{G'_{n-k}}$.

Recursively performing the analysis on $G'_{n-k}$, we have $N_{G_n} \le O(n^k) + N_{G'_{n-k}} \le O(n^k) + O((n-k)^k) + ... + O(1) = O(n^{k+1})$. □

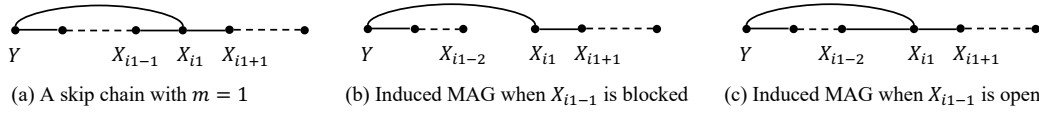

(a) A skip chain with $m = 1$     (b) Induced MAG when $X_{i1-1}$ is blocked     (c) Induced MAG when $X_{i1-1}$ is open

Figure 14: Recovering $g$-equivalence in skip-chain with $m = 1$.

**Example 8** (Skip Chain). A skip chain graph is constructed by adding skip connections among $Y$ and $m$ covariates in a chain. For any $m$-skip chain with $n$ covariates, $N_{G_{m,n}} = O(n^{2m})$.

*Proof.* Let's firstly look at the skip chain graph with $m = 1$, an example of which is shown in Fig. 14 (a). As we can see, this example is constructed by adding skip connections between $Y$ and $X_{i1}$. Following Example 6, we can construct an upper bound of $N_{G_{1,n}}$ by performing Corollary C.8 on $X_{i1-1}$. When $X_{i1-1}$ is blocked, the induced MAG is a two branches star graph, as show in Fig. 14

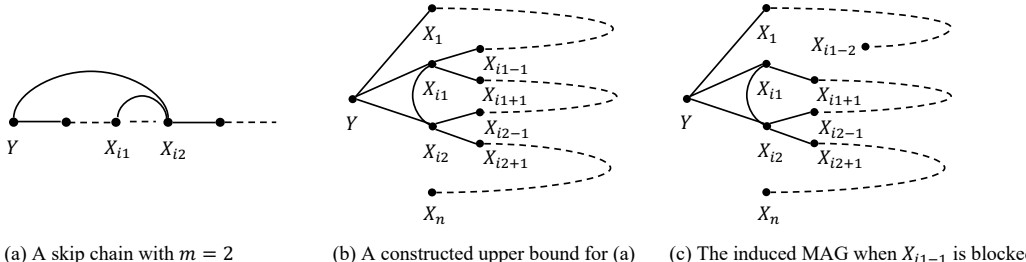

(a) A skip chain with $m = 2$     (b) A constructed upper bound for (a)     (c) The induced MAG when $X_{i1-1}$ is blocked

Figure 15: Recovering $g$-equivalence in skip chain with $m = 2$.

(b). So, the number of $g$-equivalent classes in this sub-graph is $O(n^2)$. When $X_{i1-1}$ is open, the induced MAG $G'$ will be eventually a skip chain with $m = 1$ and $n-1$ covariates, as shown in Fig. 14 (c). So, we have the following inequation: $N_{G_{1,n}} \leq O(n^2) + N_{G'_{1,n-1}}$. Recursively performing this analysis, we have $N_{G_{1,n}} \leq O(n^2) + N_{G'_{1,n-1}} \leq ... \leq O(n^2) + O((n-1)^2) + ... + O(1) = O(n^3)$.

Now that we have analyzed the skip chain graph when $m = 1$, let's look at the situation when $m = 2$, an example of which is shown in Fig. 15 (a). Firstly, we can construct an upper bound for any skip chain graph with $m = 2$ by adding extra connections among $Y, X_{i1}, X_{i2}$, until these three vertices form a complete connection, as shown in Fig. 15 (b). Then, we again perform Corollary C.8 on $X_{i1-1}$. When $X_{i1-1}$ is blocked, the induced MAG contains two disjoint branches, one of which is a chain, the other is eventually a skip chain graph with $m = 1$. So, we have number of $g$-equivalent classes in this sub-graph $O(n^3)$. When $X_{i1-1}$ is open, the induced MAG will be a skip chain with $m = 2$ and $n - 1$ covariates. So, similarly by the inequation when $m = 1$, we have $N_{G_{2,n}} = O(n^4)$.

Following the spirit of the above analysis, we show $N_{G_{m,n}} = O(n^{2m})$ by induction.

**Base.** When $m = 1$, $N_{G_{1,n}} = O(n^2)$ holds.

**Induction Hypotheses.** Suppose for any skip chain graph with $m$, we have $N_{G_{1,n}} = O(n^{2m})$.

**Step.** For skip chain graph with skip connections among $Y$ and $X_{i1}, ..., X_{i_{m+1}}$, firstly construct an upper bound by adding extra connections among $Y$ and $X_{i1}, ..., X_{i_{m+1}}$, until these vertices form a complete connection. Then, in the resulted graph, perform Corollary C.8 on $X_{i1-1}$. When $X_{i1-1}$ is blocked, the induced MAG contains two disjoint branches, one of which is a chain, the other is eventually a skip chain graph with less than or equal to $m$. So, we have number of $g$-equivalent classes in this sub-graph $O(n^{2m+1})$. When $X_{i1-1}$ is open, the induced MAG will be a skip chain with $m + 1$ and $n - 1$ covariates. So, similarly by the inequation when $m = 1$, we have $N_{G_{m,n}} = O(n^{2m+2})$.

$\square$

**Lemma D.2** (Property of Tree). *A tree is an undirected graph in which any two vertices are connected by exactly one path. If there are $d_L$ leaves and $d_{\geq 3}$ vertices of degree at least three in the tree, then $d_L \geq d_{\geq 3} + 2$.*

*Proof.* Denote number of all vertices in the tree as $d_T$, then by the handshaking lemma,

$$d_L + 2(d_T - d_L - d_{\geq 3}) + 3d_{\geq 3} \leq \sum_{i=1}^{d_T} \deg(V_i) = 2(d_T - 1),$$

which indicates $d_L \geq d_{\geq 3} + 2$. $\square$

**Proposition D.3.** *Complexity of Alg. 6 is $\Theta(N_G)$.*

*Proof.* Treat each call of the $\mathrm{MAG}(\cdot)$ function in Alg. 7 as a unit operation.

**1.** In the recursion tree of Alg. 6, number of all vertices $d_T$ is the complexity of Alg. 6, while number of leaves $d_L$ is $N_G$.

**2.** Each interval vertices in the recursion tree has degree at least three (one parent vertex in the tree and at least two children vertices in the tree). By Lemma D.2, $\frac{d_T+1}{2} \leq d_L \leq d_T$, which indicates $d_T = \Theta(d_L)$ and thus complexity of Alg. 6 is $\Theta(N_G)$. $\qquad\square$

## E  APPENDIX FOR SEC. 3.4: SPARSE MIN-MAX OPTIMIZATION

In this section, we introduce theoretical analysis of the following empirical min-max optimization problem and besides, a more efficient algorithm called *Linearized Bregman Iteration* (LBI).

$$\min_{\alpha,\beta} \max_{\theta} \frac{1}{2n} \sum_{i=1}^{n} \left[ \left( y_i - f_\alpha(\boldsymbol{x}_{i,S}\beta, \boldsymbol{x}_{i,M}) \right)^2 \right] + \lambda \|\beta\|_1, \qquad (11)$$

where $\boldsymbol{x}_1, ..., \boldsymbol{x}_n \sim_{i.i.d} \bar{p}(\boldsymbol{x}, y | \boldsymbol{x}_m = J_\theta(pa(\boldsymbol{x}_M)))$. For simplicity, we use $\bar{p}$ to denote $\bar{p}(\boldsymbol{x}, y | \boldsymbol{x}_m = J_\theta(pa(\boldsymbol{x}_M)))$ in the rest of this paper.

### E.1  STATISTICAL CONSISTENCY

Denote $\ell(\beta, \alpha, \theta; \boldsymbol{x}, y) := \left( y - f_\alpha(\boldsymbol{x}_S\beta, \boldsymbol{x}_M) \right)^2$. Suppose we can obtain the worst-case risk with $\theta^* := \arg\max_\theta \mathbb{E}_{\bar{p}}[\ell(\beta, \alpha, \theta; \boldsymbol{x}, y)]$. Then we denote

$$\mathcal{L}(\beta, \alpha) := \frac{1}{2n} \sum_{i=1}^{n} \ell(\beta, \alpha, \theta; \boldsymbol{x}_i, y_i), \ (\alpha^*, \beta^*) := \arg\min_{\alpha,\beta} \mathbb{E}_{\bar{p}}[\mathcal{L}(\beta, \alpha)].$$

Based on this, we denote $\mathcal{A} := \mathrm{supp}(\beta^*)$. In this regard, the optimization with respect to $(\alpha, \beta)$ is Lasso with a general loss. Since our goal is to recover the optimal subset $\mathcal{A}$ and the predictor with $(\alpha^*, \beta^*)$, we are interested in the model selection consistency and $\ell_2$-consistency properties:

- **Model Selection Consistency:** $\lim_{n\to\infty} P(\mathcal{A}_n) = P(\mathcal{A})$, where $\mathcal{A}_n := \mathrm{supp}(\hat{\beta}_n)$.

- **$\ell_2$-Consistency:** $\lim_n \|\hat{\zeta}_n - \zeta^*\|_2^2 = 0$, where $\zeta := (\alpha^\top, \beta^\top)^\top$.

Here, we denote $\hat{\zeta}_n := \arg\min_n \mathcal{L}(\zeta) + \frac{\lambda_n}{n} \|\beta\|_1$. The model selection consistency can ensure us to find the optimal subset and the $\ell_2$-consistency further guarantees the optimality of learned predictor. In the following, we discuss two settings: **i) fixed** when $|S| = d$ is fixed; **ii) high-dimensional** $d$ increases with $n$. We first introduce some assumptions, which are commonly made in Lasso Zhao & Yu (2006); Negahban et al. (2012); Rejchel (2016):

**Assumption E.1** (Restricted Strongly Convexity (RSC))**.** We assume that $\mathcal{L}$ is convex; $\mathcal{L}$ and $Q := \mathbb{E}_{\bar{p}}[\mathcal{L}]$ are twice differentiable and satisfies $H := \nabla^2 \mathcal{L}(\alpha^*, \beta^*) \succeq \gamma * I$ and $\bar{H} := \nabla^2 Q(\alpha^*, \beta^*) \succeq \gamma * I$ for some $\gamma > 0$.

**Assumption E.2** (Square-integrability of the gradient)**.** We assume $\mathbb{E}\left[ |\partial \ell(\alpha, \beta)|^2 \right] < \infty$ for each $(\alpha, \beta)$ in some neighborhood of $(\alpha^*, \beta^*)$.

**Assumption E.3** (Irrepresentable condition)**.** We assume that

$$\left\| H_{\mathcal{A}^c,(\alpha,\mathcal{A})} H_{(\alpha,\mathcal{A}),(\alpha,\mathcal{A})}^\dagger \begin{pmatrix} 0 \\ \mathrm{sign}(\beta^*) \end{pmatrix} \right\|_\infty < 1.$$

*Remark* E.4. The restricted strongly convexity condition has been widely assumed in variable selection Negahban et al. (2012); Zhao & Yu (2006), especially in high-dimensional statistics to ensure the identifiability of the oracle parameter. The irrepresentable condition was almost necessary to recover the true signal set. For regularity, it was needed in general convex loss Niemiro (1992); Rejchel (2016) to ensure asymptotic normality.

Now we are ready to introduce our results. We first introduce the model selection consistency with fixed setting. Before that, we first introduce two lemmas in Rejchel (2016).

**Lemma E.5** (Corollary 2.3 in Rejchel (2016))**.** *Under assumptions E.1, E.2 and set $\lambda_n$ such that* $\lim_n \frac{\lambda_n}{n} = 0$ *and* $\lim_n \frac{\lambda_n}{\sqrt{n}}$, *we have* $\frac{n}{\lambda_n}(\hat{\zeta}_n - \zeta^*) \to_p \zeta^0 := \arg\min_\zeta V(\zeta)$ *with*

$$V(\zeta) = \frac{1}{2} \zeta^\top \bar{H} \zeta + \sum_{j \in \mathcal{A}} \zeta_j \mathrm{sign}(\zeta_j^*) + \sum_{j \notin \mathcal{A}} |\zeta_j|.$$

**Lemma E.6** (Theorem 2.3 in Rejchel (2016)). *Under the same conditions in Lemma. E.5, we have*

$$\sum_{|\zeta - \zeta^*| \le Ma_n} a_n^{-1} \left| \frac{\partial \mathcal{L}(\theta)}{\partial \zeta} - \frac{\partial \mathcal{L}(\zeta^*)}{\partial \zeta} - \bar{H}(\zeta - \zeta^*) \right| \to_p 0.$$

With this lemma, we have the following model selection consistency results:

**Theorem E.7.** *Under the same conditions in Lemma. E.5 and additionally assumption E.3, we have that* $\lim_n P(\mathcal{A}_n) = P(\mathcal{A})$.

The proof is very similar to Corollary 2.4 in Rejchel (2016). We include it here for completeness.

*Proof.* Denote $\mathcal{L}_{\lambda_n}(\zeta) := \mathcal{L}(\zeta) + \lambda_n/n$. Note that if there exists $j \in \mathcal{A}$, then we have $P(j \notin \mathcal{A}) = P(\hat{zeta}_n(j) = 0) \to 0$ according to Lemma. E.5. Thus we have $P(\mathcal{A} \subset \mathcal{A}_n) \to 1$. Next, we show that $P(\mathcal{A}_n \subset \mathcal{A}) \to 1$. Otherwise, $\forall n > 0$, there exists $j \in \mathcal{A}_n$ but not belong to $\mathcal{A}$. Recall that $\hat{\zeta}_n$ minimizes $\mathcal{L}_{\lambda_n}(\zeta)$, we have that

$$\frac{\partial \mathcal{L}(\hat{\zeta}_n)}{\partial \beta_j} + \frac{\lambda_n}{n} \partial |\beta_j| = 0.$$

Since $\beta_j \neq 0$, we have that

$$\frac{n}{\lambda_n} \left| \frac{\partial \mathcal{L}(\hat{\zeta}_n)}{\partial \beta_j} \right| = 1.$$

Besides, we have that

$$\frac{n}{\lambda_n} \frac{\partial \mathcal{L}(\hat{\zeta}_n)}{\partial \beta_j} = \frac{n}{\lambda_n} \left[ \frac{\partial \mathcal{L}(\hat{\zeta}_n)}{\partial \beta_j} - \frac{\partial \mathcal{L}(\zeta^*)}{\partial \beta_j^*} - \bar{H}(\hat{\zeta}_n - \theta^*) \right] + \frac{n}{\lambda_n} \frac{\partial \mathcal{L}(\zeta^*)}{\partial \beta_j^*} + \frac{n}{\lambda_n} \bar{H}(\hat{\zeta}_n - \zeta^*).$$

According to Lemma. E.6, we have that the first term converges to 0 in probability; besides, due to square-integrability and central limit theorem, we have that the second term also converges to 0 in probability. From Lemma. E.5, the third term converges to $\bar{H}\zeta^0$ in probability. Note that $\zeta^0$ satisfies that:

$$H_{(\alpha, \mathcal{A}), (\alpha, \mathcal{A})} (\alpha^{0, \top}, \beta_{\mathcal{A}}^{0, \top})^\top = (0^\top, (\text{sign}(\beta_{\mathcal{A}}^*))^\top)^\top,$$

$$H_{\mathcal{A}^c, (\alpha, \mathcal{A})} (\alpha^{0, \top}, \zeta_{\mathcal{A}}^{0, \top})^\top = \frac{\partial \partial \|\beta_{\mathcal{A}^c}^0\|_1}{\partial \beta_{\mathcal{A}^c}^0}$$

Therefore, we have $|H_{\mathcal{A}^c} \zeta^0| < 1$ since the irrepresentable condition holds. In this regard, we have that

$$\left| \frac{n}{\lambda_n} \frac{\partial \mathcal{L}(\hat{\zeta}_n)}{\partial \beta_j} \right| < 1,$$

which contradicts to the fact that $j \in \mathcal{A}_n$. $\qquad \square$

Next we show that in both *fixed* and *high-dimensional* settings, we have the following $\ell_2$-consistency, which is a natural conclusion applying the results in $M$-estimator Negahban et al. (2012):

**Theorem E.8.** *Under assumptions E.1 and suppose* $\lambda_n \ge 2n\|\nabla \mathcal{L}(\zeta^*)\|_\infty$. *Then we have*

$$\|\hat{\zeta}_n - \zeta^*\|_2^2 = O\left( \frac{\lambda_n^2}{n^2 \gamma^2} (|\mathcal{A}| + \dim(\alpha)) \right)$$

*Proof.* According to theorem 1 in Negahban et al. (2012), we have that

$$\|\hat{\zeta}_n - \zeta^*\|_2^2 = O\left( \frac{\lambda_n^2}{n^2 \gamma^2} \Psi^2 \right),$$

if $\lambda_n \ge 2nR^*(\nabla \mathcal{L}(\zeta^*))$ for some regularization funtion $R$, with $R^*$ denoting the conjugate function of $R$ and $\Psi := \sum_{\beta \neq 0} \frac{R(\beta)}{\|\beta\|}$. In our setting, $R(\beta) := \|\beta\|_1$. Therefore, we have $\Psi \le \sqrt{|\mathcal{A}| + \dim(\alpha)}$. The proof is completed by noting that $R^* = \|\|_\infty$. $\qquad \square$

*Remark* E.9. According to square-integratility and the large law number theorem, we have that $\nabla \mathcal{L}(\zeta^*) \to_{a.s.} 0$ and thus $\|\nabla \mathcal{L}(\zeta^*)\|_\infty \to_{a.s.} 0$. Therefore, as long as $\lambda_n$ satisfies conditions in Thm. E.7, $\lambda_n$ can satisfy $\lambda_n \geq 2n\|\nabla \mathcal{L}(\zeta^*)\|_\infty$. In this regard, both model selection consistency and $\ell_2$-consistency in *fixed* setting can hold; while in high-dimensional setting, we have $\ell_2$-consistency, which is our ultimate goal, *i.e.*, identifying the minimax optimal predictor.

## E.2 Linearized Bregman Iteration

In this section, we introduce an alternative algorithm, namely *Linearized Bregman Iteration* (LBI) to replace the minimization step via Lasso. LBI was firstly proposed in Osher et al. (2005) in image denoising. In Osher et al. (2016); Huang & Yao (2018), the authors established LBI's statistical model selection consistency from the perspective of differential inclusion. Such consistency holds under nearly the same condition in linear model; however additionally requires restricted strongly convexity to hold for each solution in the path, under general convex loss.

More importantly, LBI enjoys more efficiency in implementation, compared to Lasso. Specifically, the condition on $\lambda_n$ for model selection consistency and $\ell_2$-consistency is in an asymptotic form. In practice, to select the optimal $\lambda$, Lasso has to set a sequence of hyperparameters and run an optimization algorithm for each hyperparameter. In contrast, LBI can generate a whole regularization solution path, with each iteration corresponding to a solution in the pat. Motivated by this property, we proposed to replace the minimization step via LBI, which is composed of a gradient descent followed by a soft-thresholding step.

Combined with the gradient ascent step, the algorithm is showed as follows:

**Maximization step:**
$$\theta_{k+1} = \theta_k + \delta \nabla_\theta \ell(\beta_k, \alpha_k, \theta_k), \quad \text{gradient ascent w.r.t. } \theta$$

**Linearized Bregman Iteration:**
$$\alpha_{k+1} = \alpha_k - \kappa\delta \nabla_\alpha \ell(\beta_k, \alpha_k, \theta_k), \quad \text{gradient descent w.r.t. } \alpha$$
$$z_{k+1} = z_k - \delta \nabla_\beta \ell(\beta_k, \alpha_k, \theta_k), \quad \text{gradient descent w.r.t. } \beta$$
$$\beta_{k+1} = \kappa\,\text{sign}(z_{k+1}) \max(0, |z_{k+1}| - 1). \quad \text{soft-thresholding to obtain } \beta$$

Here, the $\delta$ is step size, $z := \|\beta\|_1 + \frac{1}{2\kappa}\|\beta\|_2^2$, and $\kappa > 0$ denotes the damping factor which is trade-off between efficiency and statistical properties. Specifically, Inverse Scale Space (ISS) which can return unbiased solutions, is the limit of LBI as $\kappa \to \infty$; however, large $\kappa$ leads to computation inefficiency by noticing that $\delta$ and $\kappa$ should satisfy $\delta\kappa < 1/\lambda_{\max}(\nabla^2 \ell)$ where $\lambda_{\max}(A)$ denotes the maximal eigenvalue of $A$. Instead of running an optimization algorithm in the minimization step via Lasso, it only spends a gradient descent and a soft-thresholding steps, which is much more efficient for implementation. However, a disadvantage of using LBI to replace Lasso lies in the lack of statistical consistency guarantees, as the LBI alternates with the gradient ascent w.r.t. $\theta$.

## F  APPENDIX FOR SEC. 4: EXPERIMENT

**Implementation of Baselines.** Vanilla uses $E[Y|\boldsymbol{x}]$ to predict $Y$ and is implemented by the same neural network as $f_{S_-}$ (which will be introduced later). Other baselines are implemented by the authors' official codes. Specifically, ICP (`https://github.com/juangamella/icp`); IC (`https://github.com/mrojascarulla/causal_transfer_learning`); Anchor regression (`https://github.com/rothenhaeusler/anchor-regression`); IRM (`https://github.com/facebookresearch/InvariantRiskMinimization`); HRM (`https://github.com/LJSthu/HRM`); IB-IRM (`https://github.com/ahujak/IB-IRM`); As the Surgery Estimator did not provide official codes, we implement it following settings of our method.

### F.1  SIMULATION

**Implementation Details.** In all three settings, SGD is used for optimization. In *Setting*-1,2, the structural equation $x_1 \leftarrow g_1(x_4, y) + u_1$ is estimated by a one-layer fully-connected neural network (FC), with training iterations set to 1000, the learning rate set to 0.01. In *Setting*-3, the equation is estimated by a two-layers FC with a sigmoid activation function in the hidden layer, with training iteration set to 1000, the learning rate set to 0.01. In *Setting-1*: $f_{S_-}$ is parameterized by a one-layer FC, with training iterations set to 2000, the learning rate set to 0.001. $J_\theta$ is parameterized by the same structure, with training iterations set to 2000, the learning rate set to 0.05. In *Setting-2*: $f_{S_-}$ is parameterized by a one-layer FC, with training iterations set to 1000, the learning rate set to 0.001. $J_\theta$ is parameterized by the same structure, with training iterations set to 5000, the learning rate set to 0.05. In *Setting-3*: $f_{S_-}$ is parameterized by a two-layers FC with a sigmoid activation function in the hidden layer, with training iterations set to 5000, the learning rate set to 0.01. $J_\theta$ is parameterized by the same structure, with training iterations set to 2000, the learning rate set to 0.01. The codes are implemented with PyTorch 1.10 and run on a server with an Intel Xeon E5-2699A v4@2.40GHz CPU.

**Additional Results on Causal Discovery.** We randomly generate DAGs according to the Erdos-Renyi model Erdős et al. (1960). We consider three low dimensional settings of nodes number $\{6, 8, 10\}$ and a high dimensional setting with 100 nodes. For the low dimensional settings, we generate 10 domains, where the number of mutable variables is set to $\{2, 3\}$, and the sample size $n_e$ is set to 200 for each domain. For the high dimensional setting, the generated graphs are sparse. We generate 20 domains, where the number of mutable variable is set to 20, and the sample size $n_e$ is set to 500 for each domain. For the low dimensional settings, we implement the PC Spirtes et al. (2000) algorithm to learn the undirected skeletons. For the high dimensional setting, PC-stable Colombo et al. (2014) is used. Our algorithm is then used to determine local components. To remove the effect of randomness, we repeat for 40 times. We report the $F_1$ score, precision, and recall in Tab. 2. As we can see, when the causal graphs are more complicated, our discovery algorithm can still give accurate results, which further validates its effectiveness and stability.

Table 2: Performance of Causal Discovery.

| Nodes / Metrics | 6 | 8 | 10 | 100 |
|---|---|---|---|---|
| $F_1$ | 0.98 | 0.96 | 0.94 | 0.88 |
| precision | 0.97 | 0.96 | 0.94 | 0.86 |
| recall | 0.99 | 0.96 | 0.95 | 0.90 |

**Comparison with Baselines.** We report the maximal mean square error (max MSE) over the test sets for our method and baselines in Tab. 3. Besides, we in Tab. 4 report the standard deviation of mean square error (std. of MSE) over the test sets as a measure of transferring stability. As we can see, the maximum and standard deviation of MSE of our method are both low. For example, max MSE is 0.0075, and std. of MSE is 0.0006 in setting-2. This verifies that our method is both robust and stably transferable to distributional shifts. Besides, our method has a large improvement over baselines in

the highly non-linear setting-3. As for the slight improvements over the baseline, it may be due to the simulation settings being simple enough for the vanilla method to only exploit $X_2$ for prediction.

## F.2 ALZHEIMER'S DISEASE DIAGNOSIS

**Implementation Details.** The imaging data are acquired from structural Magnetic Resonance Imaging (sMRI) scan. After data-preprocessing via Dartel VBM (Ashburner, 2007) and *Statistical Parametric Mapping* (SPM) for segmentation, we partition the whole brain into 9 brain regions according to Tab. 8 and Tab. 7. Data normalization (w.r.t. mean and standard deviation) is used. All structural equations are estimated by a two-layers FC with a sigmoid activation function in the hidden layer. For structural equations generating $X_2, X_3$, the training takes 5000 iterations, with the learning rate set to 0.1. For those generating $X_4, X_5, X_6, X_7$, the training takes 2000 iterations, with the learning rate set to 0.1. $f_{S_-}$ is parameterized by a two-layers FC with a sigmoid activation function in the hidden layer, with training iterations set to 5000, the learning rate set to 0.25 (decrease to 0.1 at iteration 4000). $J_\theta$ is parameterized by the same structure, with training iterations set to 2000, the learning rate set to 0.25. SGD is used for optimization. For the sparsity-based optimization, we set the training iterations to 350, with learning rate set to 0.05, and penalty weight set to 2. Adam is used for optimization.

We pick four domains with more than 40 patients as the training domains and test on the rest three domains. To remove the effect of randomness, we replicate over all the 15 possible train-test splits.

**Additional Results.** We firstly report std. of MSE over the test sets for our method and baselines in Tab. 5. As we can see, our method outperforms other baselines by a significant margin. This result demonstrates the utility of our method in learning stably transferable predictors. Then, we compare the performance of all $g$-equivalent classes with more than one member in Fig. 18. As we can see, most equivalent classes have similar performance (small deviations). As for the several classes with large deviations, it may be due to the approximation error incurred during inferring the causal graph. Next, we show the optimization curve of $h(S_-, J_\theta)$ and max MSE for 100 randomly picked subsets $S_- \subset S$, in Fig. 16 and Fig. 17. As we can see, the optimization over $J_\theta$ is well converged, and the performance of different subsets is consistent with our expectations. This observation again suggests the utility of Thm. 3.5 in finding the optimal predictor. Finally, we show the loss curve of sparsity-based optimization in Fig. 20. As we can see, the optimization over $h^*$ is well converged.

## F.3 GENE FUNCTION PREDICTION

**Implementation Details.** Data normalization (w.r.t. mean and standard deviation) is used. Structural equations generating $X_2, X_4$ are estimated by a two-layers FC with a sigmoid activation function in the hidden layer, with training iterations set to 5000, the learning rate set to 0.01. $f_{S_-}$ is parameterized by a two-layers FC with a sigmoid activation function in the hidden layer, with training iterations set to 20000, the learning rate set to 0.01. $J_\theta$ is parameterized by the same structure, with training iterations set to 120000, the learning rate set to 0.05. SGD is used for optimization. For the sparsity-based optimization, we set the training iterations to 350, with learning rate set to 0.05, and penalty weight set to 0.1. Adam is used for optimization.

We use the wide-type mice and three kinds of gene knockouts in the training domains. To remove the effect of randomness, we generate 45 replications, with each trial appending 2 out of the remaining 10 gene knockouts to the training domains and testing on the rest 8 gene knockouts.

**Additional Results.** Firstly, we report std. of MSE over the test sets for our method and baselines in Tab. 6. Similarly, our method outperforms other baselines by a significant margin, which together with the Alzheimer's disease experiment, shows the utility of our method in learning stably transferable predictors. Then, we show the optimization curve of $h(S_-, J_\theta)$ and the loss curve of sparsity-based optimization in Fig. 19 and Fig. 21, respectively. As we can see, the optimizations both well converge.

Table 3: Maximal MSE comparison on simulation data.

|  | Vanilla | ICP | IC | IRM | AncReg | HRM | IB-IRM | Surg | Ours |
|---|---|---|---|---|---|---|---|---|---|
| setting-1 | $1.90_{\pm.58}$ | $2.17_{\pm1.20}$ | $1.68_{\pm.54}$ | $1.38_{\pm.10}$ | $1.34_{\pm.23}$ | $2.69_{\pm1.74}$ | $1.58_{\pm.91}$ | $1.18_{\pm.06}$ | $\mathbf{1.18_{\pm.06}}$ |
| setting-2 | $.07_{\pm.00}$ | $.17_{\pm.31}$ | $.06_{\pm.02}$ | $.06_{\pm.04}$ | $\mathbf{.0071_{\pm.00}}$ | $.33_{\pm.77}$ | $.29_{\pm.81}$ | $.0075_{\pm.0006}$ | $.0075_{\pm.00}$ |
| setting-3 | $1.72_{\pm.72}$ | $1.61_{\pm.71}$ | $1.54_{\pm.62}$ | $2.98_{\pm1.07}$ | $2.34_{\pm.65}$ | $1.75_{\pm1.42}$ | $1.71_{\pm.41}$ | $1.10_{\pm.05}$ | $\mathbf{1.10_{\pm.05}}$ |

Table 4: Mean (over randomization) of std. (over test domains) of MSE on simulation data.

|  | Vanilla | ICP | IC | IRM | AncReg | HRM | IB-IRM | Surg | Ours |
|---|---|---|---|---|---|---|---|---|---|
| setting-1 | .36 | .22 | .27 | **.01** | .16 | .56 | .14 | .10 | .10 |
| setting-2 | .0057 | .0111 | .0051 | .0034 | **.0005** | .0820 | .0142 | .0006 | .0006 |
| setting-3 | .35 | .14 | .27 | .59 | .40 | .24 | .20 | .09 | **.09** |

Table 5: Mean (over randomization) of std. (over test domains) of MSE on ADNI dataset.

| Vanilla | ICP | IC | IRM | AncReg | HRM | IB-IRM | Ours |
|---|---|---|---|---|---|---|---|
| 0.267 | 0.270 | 0.252 | 0.166 | 0.161 | 0.294 | 0.244 | **0.018** |

Table 6: Mean (over randomization) of std (over test domains) of MSE on IMPC gene dataset.

| Vanilla | ICP | IC | IRM | AncReg | HRM | IB-IRM | Ours |
|---|---|---|---|---|---|---|---|
| 0.257 | 0.274 | 0.302 | 0.319 | 0.275 | 0.278 | 0.259 | **0.017** |

Table 7: Brain regions partition.

| Brain Region | AAL Index |
|---|---|
| Frontal lobe ($X_1$) | 3,4,5,6,7,8,9,10,11,12,13,14,15,16 |
| Medial temporal lobe ($X_2$) | 85,86,87,88 |
| Parietal lobe ($X_3$) | 59,60,61,62 |
| Occipital lobe ($X_4$) | 49,50,51,52,53,54 |
| Cingulum ($X_5$) | 31,32,33,34,35,36 |
| Insula ($X_6$) | 29,30 |
| Amygdala ($X_7$) | 41,42 |
| Hippocampus ($X_8$) | 37,38 |
| Pallidum ($X_9$) | 75,76 |

Table 8: Automatic Anatomical Labeling (AAL) indices for brain regions.

| Brain Region | AAL Index | Brain Region | AAL Index |
|---|---|---|---|
| Precentral_L | 1 | Precentral_R | 2 |
| Frontal_Sup_L | 3 | Frontal_Sup_R | 4 |
| Frontal_Sup_Orb_L | 5 | Frontal_Sup_Orb_R | 6 |
| Frontal_Mid_L | 7 | Frontal_Mid_R | 8 |
| Frontal_Mid_Orb_L | 9 | Frontal_Mid_Orb_R | 10 |
| Frontal_Inf_Oper_L | 11 | Frontal_Inf_Oper_R | 12 |
| Frontal_Inf_Tri_L | 13 | Frontal_Inf_Tri_R | 14 |
| Frontal_Inf_Orb_L | 15 | Frontal_Inf_Orb_R | 16 |
| Rolandic_Oper_L | 17 | Rolandic_Oper_R | 18 |
| Supp_Motor_Area_L | 19 | Supp_Motor_Area_R | 20 |
| Olfactory_L | 21 | Olfactory_R | 22 |
| Frontal_Sup_Medial_L | 23 | Frontal_Sup_Medial_R | 24 |
| Frontal_Mid_Orb_L | 25 | Frontal_Mid_Orb_R | 26 |
| Rectus_L | 27 | Rectus_R | 28 |
| Insula_L | 29 | Insula_R | 30 |
| Cingulum_Ant_L | 31 | Cingulum_Ant_R | 32 |
| Cingulum_Mid_L | 33 | Cingulum_Mid_R | 34 |
| Cingulum_Post_L | 35 | Cingulum_Post_R | 36 |
| Hippocampus_L | 37 | Hippocampus_R | 38 |
| ParaHippocampal_L | 39 | ParaHippocampal_R | 40 |
| Amygdala_L | 41 | Amygdala_R | 42 |
| Calcarine_L | 43 | Calcarine_R | 44 |
| Cuneus_L | 45 | Cuneus_R | 46 |
| Lingual_L | 47 | Lingual_R | 48 |
| Occipital_Sup_L | 49 | Occipital_Sup_R | 50 |
| Occipital_Mid_L | 51 | Occipital_Mid_R | 52 |
| Occipital_Inf_L | 53 | Occipital_Inf_R | 54 |
| Fusiform_L | 55 | Fusiform_R | 56 |
| Postcentral_L | 57 | Postcentral_R | 58 |
| Parietal_Sup_L | 59 | Parietal_Sup_R | 60 |
| Parietal_Inf_L | 61 | Parietal_Inf_R | 62 |
| SupraMarginal_L | 63 | SupraMarginal_R | 64 |
| Angular_L | 65 | Angular_R | 66 |
| Precuneus_L | 67 | Precuneus_R | 68 |
| Paracentral_Lobule_L | 69 | Paracentral_Lobule_R | 70 |
| Caudate_L | 71 | Caudate_R | 72 |
| Putamen_L | 73 | Putamen_R | 74 |
| Pallidum_L | 75 | Pallidum_R | 76 |
| Thalamus_L | 77 | Thalamus_R | 78 |
| Heschl_L | 79 | Heschl_R | 80 |
| Temporal_Sup_L | 81 | Temporal_Sup_R | 82 |
| Temporal_Pole_Sup_L | 83 | Temporal_Pole_Sup_R | 84 |
| Temporal_Mid_L | 85 | Temporal_Mid_R | 86 |
| Temporal_Pole_Mid_L | 87 | Temporal_Pole_Mid_R | 88 |
| Temporal_Inf_L | 89 | Temporal_Inf_R | 90 |
| Cerebelum_Crus1_L | 91 | Cerebelum_Crus1_R | 92 |
| Cerebelum_Crus2_L | 93 | Cerebelum_Crus2_R | 94 |
| Cerebelum_3_L | 95 | Cerebelum_3_R | 96 |
| Cerebelum_4_5_L | 97 | Cerebelum_4_5_R | 98 |
| Cerebelum_6_L | 99 | Cerebelum_6_R | 100 |
| Cerebelum_7b_L | 101 | Cerebelum_7b_R | 102 |
| Cerebelum_8_L | 103 | Cerebelum_8_R | 104 |
| Cerebelum_9_L | 105 | Cerebelum_9_R | 106 |
| Cerebelum_10_L | 107 | Cerebelum_10_R | 108 |
| Vermis_1_2 | 109 | Vermis_3 | 110 |
| Vermis_4_5 | 111 | Vermis_6 | 112 |
| Vermis_7 | 113 | Vermis_8 | 114 |
| Vermis_9 | 115 | Vermis_10 | 116 |

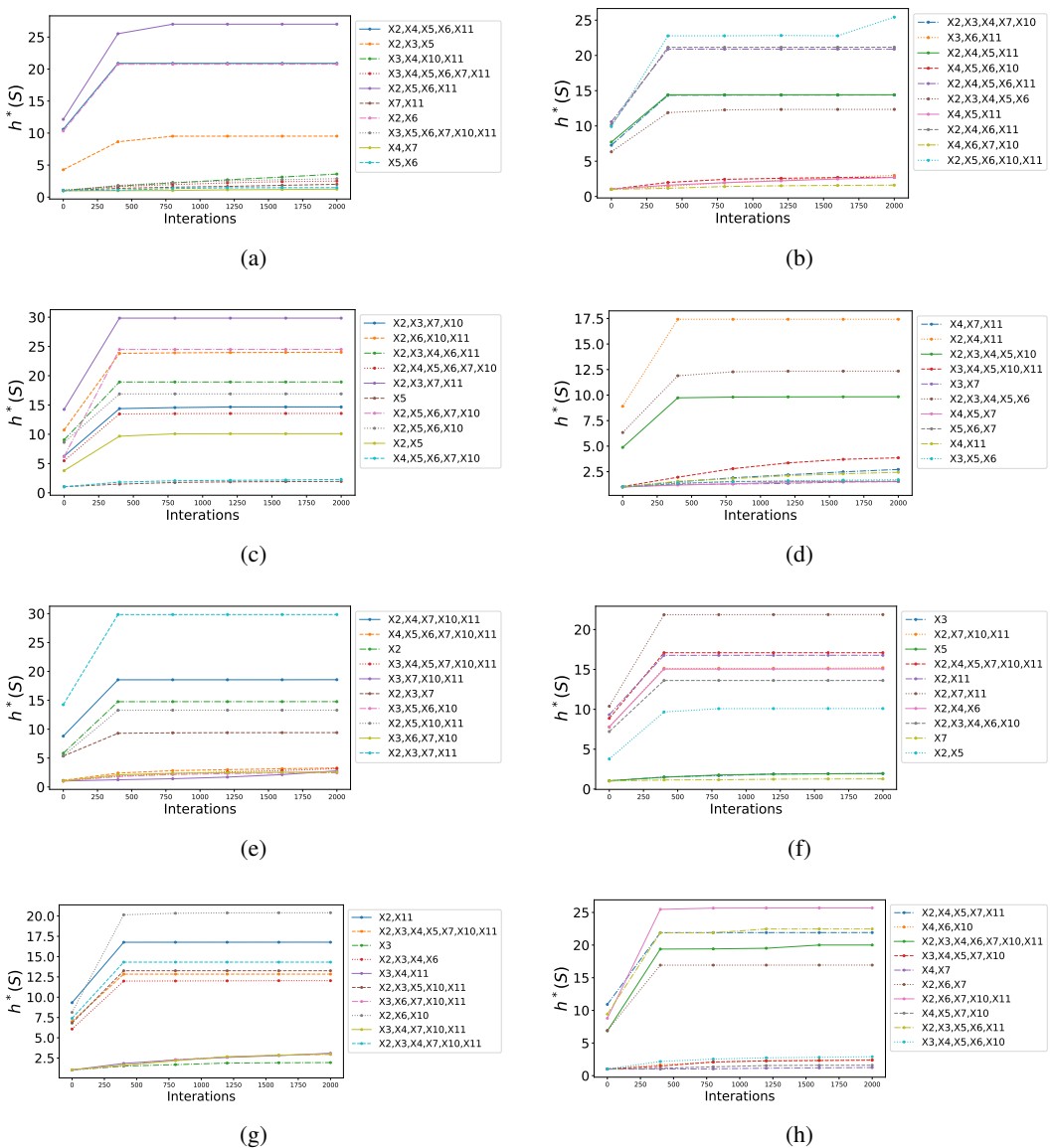

Figure 16: Optimization curves of $h(S_-, J_\theta)$ on ADNI dataset.

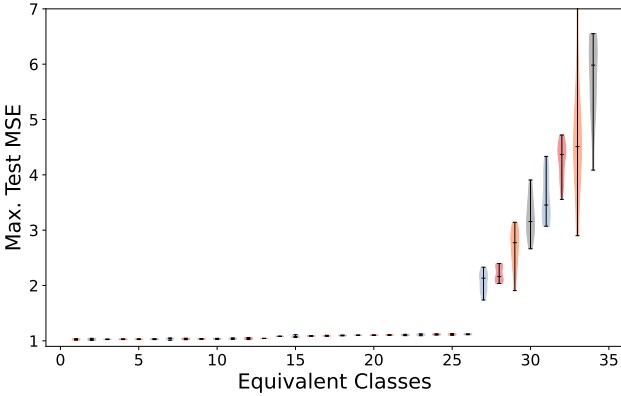

Figure 17: Optimization curves of $h(S_-, J_\theta)$ on ADNI dataset.

Figure 18: Performance of $g$-equivalent classes on ADNI.

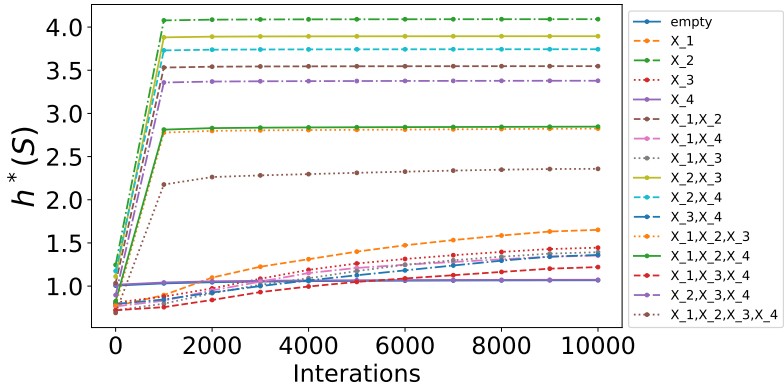

Figure 19: Optimization curves of $h(S_-, J_\theta)$ on IMPC gene dataset.

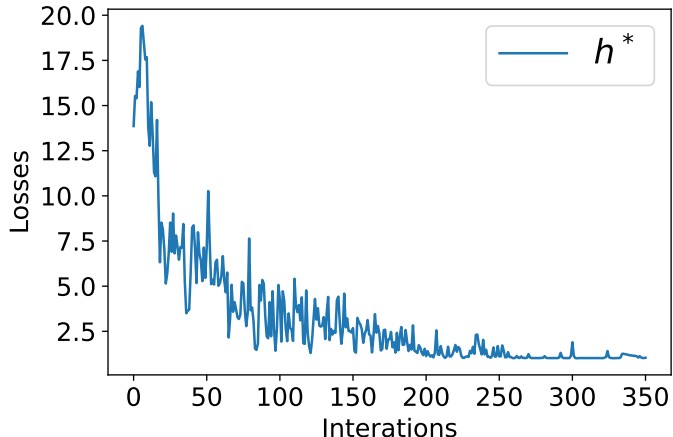

Figure 20: Loss curve of sparsity-based optimization on ADNI dataset

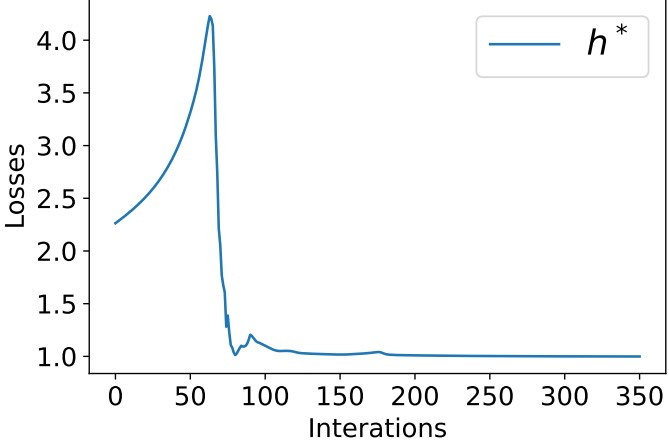

Figure 21: Loss curve of sparsity-based optimization on IMPC gene dataset

