# OpenReview forum: "Which Invariance Should We Transfer? A Causal Minimax Learning Approach"
_ICLR.cc/2023/Conference — Submitted to ICLR 2023_

### Official Review · Reviewer_AmVi · 2022-10-23

**Confidence:** 3
**Correctness:** 2
**Technical Novelty And Significance:** 3
**Empirical Novelty And Significance:** 2
**Recommendation:** 3

**Clarity, Quality, Novelty And Reproducibility:**

The topic of investigating which subset of the whole stable information should the model transfer is very interesting, particularly in combination with causal discovery. Indeed, I spent quite much time on this paper, perhaps triple of reviewing any other paper in my slot.

I have to say that the poor writing makes it hard to evaluate the paper. As a theoretical paper (or at least lots of contributions are theoretical), I feel quite struggling as many places (problem setting, proofs, notations, etc.) are not clearly and rigorously treated. I have the following concerns and questions:

1. I got confused with the definition of stable predictors in this paper. I'm familiar with works related to invariant causal prediction and invariant risk minimization. They assume that the conditional probability $P(Y|pa(Y))$ does not change across all domains of interest. Consider a setting with causal graph $X_1\to Y, Y\to X_2$, and $P(X_1)$, $P(X_2|Y)$ change but $P(Y|X_1)$ keeps constant. According to the definition of stable and unstable set in this set, it seems both $X_1, X_2$ will be treated as unstable variables, and the best estimate should be $\mathbb E(Y)$ . However, in the spirit of IRM, $X_1$ is still treated as invariant variable. Is this correct? Or can you make the setting, the definition of stable and unstable things clear in the paper? Besides, in the invariant learning literature, there has been minimax optimality result regrading the causal parental variables.
2. I feel confused with the do-operator here. Do you set $X_M$​ to be a fixed value or another random variable? I think it needs be elaborated.
3. A drawback of previous work is tghat *''the searching cost is exponentially expensive w.r.t. dS, making it hard to be applied to large-scale scenarios*--but the considered experiments in this paper is not that large-scale. And for large scale case, causal discovery will be very likely to incur errors. Can you consider even larger scale experiment?
4. for showing identification result of Proposition 3.2: why $E$ could be treated as a node in the causal graph? the causal graph part does not say anything about this. (see also a question regarding the training domains)
5. Regarding counter-example: again I feel confused about  $do(X_m)$. Why should the expectation in Eq (8) and (9)  take sum of $x_m$? Why do we have this indicator function  $\mathbb 1(x_s=1, x_m)$  when we calculate the expectation? And in the last three equations in the counter-example, you first cancel $a_y^2$, then take the limit of $a_y\to 0$ in the denominator. I feel that this part gets tricky. Can you give exact numerical values to show the inequality? A quick summary or high-level idea of the counter-example shall be provided in the main text.
6. About Theorem 3.3: I did not see how the proof shows this result. In the proof, the equation $max_{P^e}\mathcal L_{P^e}(f_{S})$=... in the first line, the max is treated w.r.t. the integral of all variables; for the second line, the max is over an integral wrt. $x_M$, and the integral is a function of $pa(X_M)$. Why are they equivalent? Moreover, $P_J$ never appears in the proof, and I cannot see this proof is valid for showing Thm 3.3.
7. About proposition 3.4: in the proof Thm 3.3, $P^e$ and $P_J$ is used as the one that achevies the wort risk. However, in this proof of Prop 3.4, $P_J$ is just distribution that yield the same observation distribution. Why?
8. **More importantly**, I am surprised that there is nothing regarding the environment of training data. Intuitively, if you have one environment, you cannot never know what mechanism is stable and what is not. In the IRM paper, one has to assume the training environments sufficiently many and diverse. Please explain.
9. the experiments are somewhat small scale. And all the causal graphs learnt from observational data are consistent with the true ones. What if the learnt graph has some error edges? Can you try the Sachs protein dataset?



**Strength And Weaknesses:**

Pros:

- the topic of investigating which subset of the whole stable information should the model transfer is very interesting
- interesting theoretical charactizations, which lead to practically efficient algorithms
- good empirical performance in the considered experiments

Cons and limitations:

- the paper relies on causal discovery from observational data, which in practice cannot avoid estimating errors, particularly for large scale settings.
- the proposed method seems not be able to work for image-like data where one has to learn representations.
- writing is poor (see below),
- minor one: causal sufficiency is required

**Summary Of The Paper:**

This paper considers the problem of learning stable information to transfer to unseen environments. In particular, following the work of Subbaswamy et al. (2019), it investigates which subset of the whole stable information should the model transfer, in order to achieve optimal generalization ability. Authors propose to maximize over mutable mechanisms and to search over only equivalent classes in terms of worst-case risk, with certain theoretical characterization based on causal graph. The performance of the proposed method is also empirically validated.

**Summary Of The Review:**

I think that the paper requires a major revision to make it readable and rigorous (e.g., clearly define the setting and notions of stable, unstable things, the do-operator; make the proof precise and give explanations where it is not very very obvious; give summary or high-level idea of your proof; etc.). Thus, I cannot recommend acceptance for the present version. I would like to re-evaluate the paper, and raise my score if authors revise their paper and could address my concerns/questions during the discussion period. I look forward to the upcoming discussions with authors.

---

> ### Author Response · Authors · 2022-11-10
> **Responses to Reviewer AmVi**
>
> We thank the reviewer for your efforts and valuable comments on our paper. We address your concerns below:
>
> **1.** "Can you make the setting, and the definition of stable and unstable things clear in the paper? Besides, in the invariant learning literature, there has been minimax optimality result regarding the causal parental variables."
>
> As we mentioned in Sect. 2, a variable $X$ is called stable if the distribution $P(X|Pa_X)$ does not change across different environments, and is called mutable if otherwise. Denote the set of stable and mutable variables respectively as $S$ and $M$. Then, the set of stable predictors considered in this paper is defined as $\mathcal{F}^S:=$ {$\{ f_{S_{-}}:=E[Y|x_{S_{-}}, do(x_M)]|{S_{-}} \subset {S} \}$}.
>
> **Example**: In your example, both $X_1$ and $X_2$ are mutable variables. The set $\mathcal{F}^S$ only contains one element $f_{\varnothing}:=E[Y|do(x_1),do(x_2)]$. According to the inference rules [1], we have $f_{\varnothing}=E[Y|x_1]$, which is not $E[Y]$.
>
> ICP and IRM explore invariance in the $Y$'s causal feature   $P(Y|\mathrm{Pa}_Y)$. In contrast, our method explores a wider range of invariance beyond the causal feature and achieves minimax optimum. To illustrate, consider the causal graph $X_1 \to Y \to X_2 \to X_3, Y \to X_3$ where $X_1,X_2$ are mutable variables. The ICP and IRM only use $Y$'s causal parent $X_1$ for prediction, while our predictor, defined on the interventional distribution $P(Y|X_3,do(x_1),do(x_2))=P(Y|X_1,X_3,do(x_2))$, can also use features from $X_2$ and $X_3$.
>
> [1] Judea Pearl. Causality. Cambridge University Press, 2009.
>
> **2.** "I feel confused with the do-operator here. Do you set $X_M$ to be a fixed value or another random variable? "
>
> The do-operation in $f_{S_{-}}:=E[Y|x_{S_{-}}, do(x_M)]$ means setting $X_M$ to a fixed value ${x}_M$.
>
> **3.** "Why $E$ could be treated as a node in the causal graph? More importantly, there is nothing regarding the environment of training data."
>
> $E$ is called the domain index variable [2]. In the causal graph over {$Y$} $\cup \mathbf{X} \cup $ {$\{E\}$}, if $E \to X$, then $X$ is a mutable variable.
>
> We supplement the following assumption on environmental heterogeneity:
>
> **Assumption**: We assume the training environments $\varepsilon_{\mathrm{Tr}}$ consistently reflect the mutation in all environments $\varepsilon$, *i.e.*, any mutable variable in $\varepsilon$ is a mutable variable in $\varepsilon_{\mathrm{Tr}}$.
>
> [2] B. Huang, et al. Causal discovery from heterogeneous/nonstationary data. JMLR. 2020.
>
> **4.** About the counter-example.
>
> About Eq. (8): unfold the first expectation by definition, we have $E[E^2[Y|x_s,do(x_m)]]=\sum_{x_s,x_m} p(x_s,x_m) E^2[Y|x_s,do(x_m)]$. Because $p(x_s,x_m)=\sum_y p(x_s,x_m,y)=\sum_y p(x_s|x_m,y)p(x_m|y)p(y)$, Eq. (8) holds. Derivation of Eq. (9) is similar.
>
> About the indicator functions: we assume $X_s,X_m$ are both binary-valued variables, as a result, the summation over $x_s, x_m$ traverses over four possibilities $1(x_s=0,x_m=0)$, $1(x_s=0,x_m=1)$, $1(x_s=1,x_m=0)$, and $1(x_s=1,x_m=1)$.
>
> About the numerical example: set $a_{s10}=0.001, a_{s11}=0.999, a_{s00}=a_{s01}=a_{s10}=0.5, a_{m0}-2a_{m1}=1, a_y=0.001$, Eq. (7) becomes $994>-1$.
>
> **5.** About the proof of Thm. 3.3.
>
> About the equivalence of the first and second line: in the equation $\max_{P^e} L_{P^e}(f_{S_{-}})$, we consider cases where there is only one mutable variable $\mathbf{X}_M=$ {$X_M$}, which means for any variable $X_i \neq X_M$,  the $p(x_i|pa(x_i))$ keeps constant when $P^e$ changes across $e$.
>
> As a result, the term $\prod_{X_i\in Pa(\mathbf{X}_M)} p(x_i|pa(x_i))$ is constant in different $P^e$ and the $\max$ can be moved inside.
>
> About the missing of $J$: the third line of the equation $\max_{P^e} L_{P^e}(f_{S_{-}})$ shows that the maximum loss is attained when $x_M=m^*(pa(x_M))$, *i.e.*,  when $x_M$ is a deterministic function $m^*(\cdot)$ of $pa_M$. Here, $m^*$ is exactly the optimized $J$ function.
>
> **6.** About the proof of Prop. 3.4.
>
> $P^e(\mathbf{X},Y)$ is the joint distribution of $\mathbf{X}, Y$ over all environments $e \in \varepsilon$. $P_J$ is an instrumental distribution defined as $P_J:=P(Y,\mathbf{X}_S|do(\mathbf{X}_M=J(\mathrm{Pa}(\mathbf{X}_M)))$. The worst-case risk is attained via maximization of $P^e$ over $e$ and can be estimated by maximization of $P_J$ over $J$. In the proof of Prop 3.4, we show $P_J$ is identifiable for any $J$, which ensures the maximization of $P_J$ over $J$ is tractable.
>
> **7.** About causal discovery.
>
> We show in Sect. F1 in the appendix that the causal discovery algorithm can consistently provide reliable results in large-scale settings.
>
> **8.** About image-like data.
>
> The proposed method focuses on choosing the optimal subset of variables to transfer. Theoretically, it also works when these variables are latent variables extracted from images.

---

> > ### Comment · Reviewer_AmVi · 2022-11-17
> > **thanks for response and clarifications**
> >
> > Thanks for clarifications. I believe the overall technical contribution with a crystal clear presentation is worth a top conference. I keep my score for the current version.

---

### Official Review · Reviewer_s36p · 2022-10-24

**Confidence:** 3
**Correctness:** 3
**Technical Novelty And Significance:** 3
**Empirical Novelty And Significance:** 2
**Recommendation:** 5

**Clarity, Quality, Novelty And Reproducibility:**

The paper seems to have some new ideas, though the discussion for each is thin. It could have presented fewer topics with more discussion on each topic.
The paper is overall well written with confusing notations and unclarity at a couple of places.

**Strength And Weaknesses:**

+1: the paper is technically sound
+2: the paper has both numerical implementations and theoretical justifications
+3: the paper studies an important yet under-developed area, and is very timely.

-1: the paper presents too many concepts but without digging up each one with sufficient depth. Theorem 3.1 is equivalent to the degeneration condition in Subbaswamy et al. (2019). Theorem 3.3 seems to be new. The equivalent class is interesting, but I wonder how much saving it will really lead to (considering that there are costs associated with identifying these equivalent classes, and there may also be errors in this identification.) The lasso part is somewhat disentangled from the rest of the topics and lasso is also not new. If the paper were to present fewer concepts but with enough discussion on each one to provide a better justification, it may work out better.
-2: The main point of Theorem 3.3 is that the main source of variability of the distributions in $\mathcal{P}$ is from the variety in $J$, which defines a mutable variable in terms of its parents using a definite function. I wonder how in practice the richness of $\{P_J\}$ aligns with the richness of $\mathcal{P}$. While $\mathcal{P}$ should contain the data distribution from all environments, how does one consider the maximization of all the possible $J$ functions? Note that all measurable functions $J$ are uncountable. If in practice one only uses a small parametric class of $J$ functions, then they may not be able to cover the base for all $\mathcal{P}$. I do not really find how $\{P_J\}$ is defined at the implementation level from the paper.
-3: As mentioned above, while the idea of the equivalent class could potentially save some training time, it also takes time to identify these equivalent classes. If the time for training each model is smaller than the time spent on the identification of these equivalent classes, then it may not be necessary to bother with these equivalent classes.
-4: The section on lasso looks a bit disconnected from the rest of the paper. Here is why. The paper went through all the efforts with Theorem 3.3, dismissing the use of the validation error in Subbaswamy et al. (2019), and defining the equivalent classes. Then all of sudden, the paper says basically if this is too much then we can just use lasso, which is a regularized version of the empirical loss function. If lasso works then what's the point of all the previous discussions?
-5: There is ambiguity as to what $f_\alpha(x_S \beta, x_M)$ is in the lasso formulation (3). What are the parameters $\alpha$ and $\beta$? Should $x_S \beta$ be thought of as the inner product? Is $\alpha$ then a parameter that governs how the single index $x_S \beta$ and the mutable variable $x_M$ are used to form the prediction function $f$?
-6: The variable selection consistency of lasso was only proved for the linear model by Zhao and Yu 2016. I am not sure if some results exist for an arbitrary non-linear model. I may be wrong about this.
-7: I think the idea is that $x_M$ should not be used as part of the predictive model. That is what I saw in Algorithm 1. However, why x_M is included in the lasso model?
-8: Some notations are confusing to me. I think D ⊥G Y | Z is defined as D and Y are d-separated by Z in the paper. But what is Y ⊥ G_{\overline{X^0_M} K| X^0_M (this appears in Example 1). There is also Y ⊥ G_{\overline{X} \underline{Z}} Z | X (see top of page 16). What do these notations mean? Still d-separation?
-9: please clarify what the two columns h*(S-) and max MSE mean respectively. As far as I can tell, h*(S-) is also a maximal MSE (over ...?) as it is defined. So it is unclear to me what is the difference. Is max MSE the test data maximal MSE over different observed environments?

**Summary Of The Paper:**

This paper concerns how to select/learn a predictive model from causal relations between variables in the distributional robustness setting. Specifically, the paper argues that given a DAG, a predictive model should be learned from a subset of the variables which are stable across a group of learning environments, while not using the information of those mechanisms which can vary between different environments. The idea is similar to that of Subbaswamy et al. (2019) except that the latter proposed to directly search for the optimal subset from the set of stable variables by comparing their validation errors. This paper provides a more complete analysis and argues that instead of using the validation error one can directly compare and minimize the worst-case risk over a class of distributions with varying mechanisms of how those variables in the mutable set are generated. In addition, in order to save the computational cost, the paper advocates to search equivalent classes of subsets instead of searching all subsets. In the case where the computation is still heavy, the paper suggests to adopt the lasso regression for subset selection.

**Summary Of The Review:**

This paper provides some analysis of the methods in Subbaswamy et al. (2019). It then proposed to compare the max (worst-case) risk directly, defining the equivalent class to save computation, and introduce lasso for consistent variable selection. Each of these could potentially be important but due to the amount of work that is presented in the paper, there are still questions on each of these. Some notations can be clarified at various places.

---

> ### Author Response · Authors · 2022-11-10
> **Responses to Reviewer s36p**
>
> Thanks for your efforts and valuable comments. We address your concerns below:
>
> **1.** "Theorem 3.1 is equivalent to the degeneration condition in Subbaswamy et al. (2019)."
>
> Our contribution is to establish the equivalence between the degenerating condition in [1] and the graphical condition in Thm. 3.1. The latter one's advantage is that it can be easily tested via causal discovery. This saves end-users from exhaustively running the ID algorithm for each DAG and checking whether the interventional distribution can degenerate.
>
> [1] A. Subbaswamy, et al. Preventing failures due to dataset shift: Learning predictive models that transport. AISTATS, 2019.
>
> **2.** "I wonder how in practice the richness of $P_J$ aligns with the richness of $P$. While $P$ should contain the data distribution from all environments, how does one consider the maximization of all the possible $J$ functions?"
>
> In the proof of Thm. 3.3, we show the maximum loss is attained when $\mathbf{X}_M$ is a deterministic function of $\mathrm{Pa}(\mathbf{X}_M)$. So we parameterize $J: \mathrm{Pa}(\mathbf{X}_M) \to \mathbf{X}_M$ as the neural network, which has the ability to approximate any deterministic function [2].
>
> To obtain $h^\star(S_{-})$, we optimize over parameters of the neural network, as mentioned in Sect. F in the appendix.
>
> [2] K. Hornik, et al. Multilayer feedforward networks are universal approximators. Neural networks, 1989.
>
> **3.** "While the idea of the equivalent class could potentially save some training time, it also takes time to identify these equivalent classes. If the time for training each model is smaller than the time spent on the identification of these equivalent classes, then it may not be necessary to bother with these equivalent classes."
>
> Denote the number of equivalent classes in $G$ as $N_G$. The following proposition, which states the complexity of identifying equivalent classes is at most $O(N_G)$, ensures the overall cost of searching over equivalent classes is still $O(N_G)$ and smaller than the cost of searching over all subsets.
>
> **Proposition D.3**: The complexity of the equivalent class identification algorithm is $O(N_G)$.
>
> Besides, the unit operation in identifying equivalent classes is to modify the adjacency matrix, which is much faster than training neural networks to estimate the worst-case risk of a subset.
>
> **4.** "I think the idea is that $X_M$ should not be used as part of the predictive model. That is what I saw in Algorithm 1. However, why $X_M$ is included in the lasso model?"
>
> As mentioned in Sect. 2, the stable predictor with subset $S_{-}$ is $f_S-:=E_{P}[Y|x_{S_{-}},do(x_M)]$,  which means $X_M$ is used as part of the predictive model.
>
> **5.** "Some notations are confusing to me. I think $D \perp_{G} Y | Z$ is defined as D and Y are d-separated by Z in the paper. But what is $Y \perp_{G_{\overline{X^0_M}}} K| X^0_M$ (this appears in Example 1). There is also $Y \perp_{ G_{\overline{X} \underline{Z}}} Z | X$ (see top of page 16). What do these notations mean? Still d-separation?"
>
> For a subset $\mathbf{X}$, $G_{\overline{\mathbf{X}}}$ denotes the graph obtained by deleting arrows pointing to any member of $\mathbf{X}$, $G_{\underline{\mathbf{X}}}$ denotes the graph obtained by deleting arrows emerging from nodes in $\mathbf{X}$. To represent the deletion of both incoming and outgoing arrows, we use the notation $G_{\overline{\mathbf{X}}\underline{\mathbf{Z}}}$.  The notions $\perp_{G_{\overline{\mathbf{X}}}}$ and $\perp_{G_{\overline{\mathbf{X}}\underline{Z}}}$ respectively denote d-separation in the graph $G_{\overline{\mathbf{X}}}$ and $G_{\overline{\mathbf{X}}\underline{\mathbf{Z}}}$.
>
> The same notations are also adopted in [1].
>
> [1] Judea Pearl. Causality. Cambridge University Press, 2009.
>
> **6.** "Please clarify what the two columns h*(S-) and max MSE mean respectively."
>
> The $h^\star(S_{-})$ is the estimated worst-case risk for subset $S_{-}$ from the training environments. The max MSE is the maximal mean square error of the subset $S_{-}$ in the test environments.

---

> > ### Comment · Reviewer_s36p · 2022-11-21
> > **Thanks for the response**
> >
> > I thank the authors for the responses. In light of the questions I had and those of the other reviewers, I would keep my original rating as I believe that a rewrite would substantially improve the clarity of the paper.

---

### Official Review · Reviewer_K5Kv · 2022-10-27

**Confidence:** 4
**Correctness:** 2
**Technical Novelty And Significance:** 4
**Empirical Novelty And Significance:** 4
**Recommendation:** 3

**Clarity, Quality, Novelty And Reproducibility:**

The paper's main weakness is clarity. There are just too many ideas, many of which are good, presented at the same time. Many key details are hidden deep in appendices. While there are many useful claims that are proved well, there are many other claims that are unsubstantiated, at least in the text as it stands. The authors need to prioritize the particular claims that they wish to make in the paper. Also, many proofs seem more convoluted than they need to be.

Overall, the quality of the work is high. As I mentioned, many of the ideas are good, but difficult to dig out.

The contributions wrt prior work are clear.

There appear to be enough details for reproducibility in the appendix, but there are too few details about the actual experiments in the main text.

**Strength And Weaknesses:**

**Strengths:**
 + The authors identify several practical gaps in the proposed implementation of the Subbaswamy et al 2019 surgery estimator. The graphical criterion for conditioning on all stable variables is particularly nice, as it saves end-uses from having to run the ID algorithm for each DAG to understand whether the intervened distribution can be written as a conditional distribution.
 + The observation that the best subset of variables based on validation risk in the training environments does not correspond to the best subset for minimax risk is important. I wish the disconnect between these two were highlighted more prominently in a very simple toy example.
 + I like the claim that the minimax risk can be estimated by exploring the set of potential downstream distributions over $X_M$, although this was also not treated in enough detail.
 + The proposed application of LASSO style selection as an alternative to subset search is interesting, although this was also not treated in enough detail.
 + Experimental results seem promising and Figures 1, 6, and 7 are particularly compelling, although absolute numbers for MSE and comparisons of performance across different target distributions would be appreciated.

**Weaknesses:**

**Missing Technical Details**
 - The authors try to do too much in the paper, so many ideas are not treated thoroughly enough. While some results are shown with some rigor (e.g., the graphical criterion in Theorem 3.1, crucial details of other parts of the algorithm are missing. Several examples of missing details follow. I would suggest cutting down the claimed contributions of the paper, or expanding to a longer format venue.
- How does one solve the maximization problem over $P_J$ that defines $h^*$ in practice? $P_J$ is a non-parametric class of functions that allows arbitrary dependence for the structural equation $Pa(X_M) \rightarrow X_M$. This seems like a critical detail for the proposed algorithm to be usable. Does one need to make assumptions about $J$?
 - There appear to be missing assumptions for the structure identifiability claim in Proposition 3.4, and the proof for this proposition is not actually a proof, but the statement of an algorithm. At the very least, it seems important to have some assumption akin to positivity about the set of environments that is observed in training; as of now, there are no conditions stated about how the environments should be heterogeneous, or whether all environments need to change all mutable variables or something similar. I imagine there are many stronger conditions that are given in the Huang et al 2020 paper that is quoted extensively.
 - The g-equivalence algorithm seems quite critical given the claims in the paper, but is omitted from the main text. In particular, it seems important to understand when composing the g-equivalence algorithm with the reduced variable selection problem is actually more efficient than the original variable selection problem. I suspect it is, but it seems necessary to substantiate this claim. Meanwhile, the complexity analysis section goes into examples that don't feel particularly relevant.
 - It is not clear to me whether the Estimate $f_{S_-}$ section is correct, or why the algorithm needs to be so complicated; if $f_{S_-}$ is identifiable, shouldn't we be able to to estimate it by obtaining a functional from the ID algorithm? This seems more straightforward than estimating and generating from structural equations, which also requires additional assumptions for correctness. At any rate, it needs to be proved that the proposed algorithm actually estimates the appropriate interventional distribution.
 - The sparse min-max optimization section has far too little detail, and it is not clear what claims the authors want to make here. None of the cited work has results about doing variable selection with the inner maximization (rather they assume risk is minimized across the same distribution regardless of the $\alpha, \beta$ parameters), nor does it extend beyond linear models. If this is meant to be a main contribution, more details of the exact approach should be given in the main text. It is not necessary to prove anything about this practical approach, but the citations to selection consistency work suggest that the authors want to make stronger claims than they can substantiate.
 - For the actual implementations, no details are given about how the predictors are actually estimated. Is everything a linear model? How are hyperparameters set (for example, the very important robustness parameter in Anchor Regression)?
 - How are bi-directed edges handled by the algorithm? There is no mention of these, and some of the proofs assume a topological ordering exists, but the causal structure discovery algorithm returns bidirected edges in some cases.

**Other issues**
 - In the set-up, there is no formal definition of a "stable predictor". Currently there is a sentence stating that such a predictor can be "transferred to a broader family of environments without any adjustment", but it is not clear what "transferred" means in this sentence. What is the actual invariance criterion, i.e., the property of f(X) that is the same across environments?
 - I think there is an error in Example 1. When the dashed arrow is absent, by the do calculus rules 1 then 3, $p(y \mid k, do(x^0_m)) = p(y \mid do(x^0_m)) = p(y)$. This is consistent, I believe, with Theorem 3.1, which reduces to the conditional $p(y \mid k_2)$, but $k_2$ is the empty set in this example.
 - Notation in the Proof of Theorem 3.3 is different from the notation in the statement.

**Suggestions:**
 - Can the proof of Theorem 1 be written using the rules of do calculus? It seems like this would be more straightforward than going back to conditional probability statements.
 - It would be nice to have a simple toy example showing how changes to the $X_M$ conditional distributions can induce changes in the risk of $f_{S_-}$, specifically showing that $f_{S_-}$ with the lowest in-distribution validation risk may not have the lowest worst-case validation risk.
 - The causal structure discovery portion of the algorithm seems like a distraction. Most of the result is quoting identification results from Huang et al and not a core contribution of this work. In addition, the assumptions for the structure discovery algorithms to work are stronger than the assumptions provided here. I would suggest dropping the structure discovery portion from the main development, and discussing structure discovery as a practical tool for implementation in the experiments section, as the contribution of this paper is mostly about doing the min-max optimization after causal structure has been ascertained, and there is no analysis of, say, the propagation of error from causal structure discovery to the final predictor. This would provide room for other important details, such as the algorithm for identifying g-equivalence classes or the sparse selection algorithm.
 - Comparisons of MSE across environments (not just worst-case MSE) would be useful here, since there is often a tradeoff between the two. It would also be useful to see how the variables selected by Surgery and the proposed algorithm differ. Finally, it would be useful to understand whether the MSE of these estimators is actually in a range that would be useful for prediction. Comparing to, e.g., an oracle estimator that is trained on data from the target environment, or a baseline that just takes the mean outcome would be useful.

**Summary Of The Paper:**

The paper proposes a model selection procedure for choosing stable models that have minimax optimal risk across a family of distributions  induced by intervening on mutable variables. This work extends the shift-stable prediction principles established in Subbaswamy et al 2019, who established that one can construct a shift-stable predictor by estimating the intervened conditional expectation of Y given an intervention on mutable variables, and some subset of stable variables. This work addresses some practical gaps left in that paper: in particular, it is often ambiguous which set of stable variables one should condition on (in addition to intervening on mutable variables) from the perspective of minimax MSE, and cases where there exists no ambiguity are hard to identify. To address these gaps, the authors propose (1) a simple-to-confirm graphical criterion that is equivalent to the case where one should condition on all the stable variables to obtain a minimax predictor; (2) a selection criterion based on estimated minimax risk for selecting a minimax optimal subset of variables; and (3) algorithms for reducing the complexity of this subset selection. In addition, the authors propose using causal structure discovery from multiple environments (Huang et al 2020) to learn causal structure and mutable/immutable variables. The authors demonstrate that their algorithm reduces worst-case MSE in several cross-environment generalization experiments.

**Summary Of The Review:**

The overall method here seems promising theoretically and empirically, and many of the claims are interesting. However, there are too many additional extraneous claims that are unsubstantiated or distracting.

---

> ### Author Response · Authors · 2022-11-10
> **Responses to Reviewer K5Kv**
>
> We thank the reviewer for your efforts and valuable comments on our paper. We address your concerns below:
>
> **1.** "How does one solve the maximization problem over $J$ that defines $h^\star$ in practice? Does one need to make assumptions about $J$?"
>
> In the proof of Thm. 3.3, we show the maximum loss is attained when $J$ is a deterministic function. So we parameterize $J$ as the neural network, which has the ability to approximate any deterministic function [1]. To obtain $h^\star(S_{-})$, we optimize over parameters of the neural network, as mentioned in Sect. F in the appendix.
>
> [1] K. Hornik, et al. Multilayer feedforward networks are universal approximators. Neural networks, 1989.
>
> **2.** "There appear to be missing assumptions for the structure identifiability claim in Proposition 3.4, and the proof for this proposition is not actually a proof, but the statement of an algorithm."
>
> We supplement the following assumption on environmental heterogeneity:
>
> **Assumption**: We assume the training environments $\varepsilon_{\mathrm{Tr}}$ consistently reflect the mutation in all environments $\varepsilon$, *i.e.*, any mutable variable in $\varepsilon$ is a mutable variable in $\varepsilon_{\mathrm{Tr}}$.
>
> The proof of Prop. 3.4 shows the identifiability of $\mathrm{Pa}(\mathbf{X}_M)$, $P_J$, and $f_S-$ by offering an algorithm to identify them.
>
> **3.** "In particular, it seems important to understand when composing the g-equivalence algorithm with the reduced variable selection problem is actually more efficient than the original variable selection problem."
>
> It is found that the g-equivalence algorithm can largely reduce the searching cost from exponential to polynomial when the causal graph is sparse. Examples are available in Sect. D in the appendix.
>
> **4.** "It is not clear to me whether the Estimate $f_{S_{-}}$ section is correct, or why the algorithm needs to be so complicated."
>
> Indeed, the $f_{S_{-}}$ is identifiable, which means we can directly estimate it from the joint distribution. However, the estimation can be cumbersome because the joint distribution learned from data may not be accurate. To tackle this problem, we in Sect. 3.2 provide a permutation-regeneration scheme to remove the ``do'' and convert the joint distribution estimation into a regression problem. The validity of this method is proved in Sect. C1 in the appendix
>
> **5.** "For the actual implementations, no details are given about how the predictors are actually estimated. Is everything a linear model? How are hyperparameters set?"
>
> We use two-layers feedforward neural networks with sigmoid activation in the hidden layer to implement the predictors, which together with the hyper-parameters are mentioned in Sect. F in the appendix.
>
> **6.** "How are bi-directed edges handled by the algorithm? There is no mention of these, and some of the proofs assume a topological ordering exists, but the causal structure discovery algorithm returns bi-directed edges in some cases."
>
> The structure identifiability results (Prop. 3.3 and Prop. 3.4) ensure the edges and topological orderings involved in our algorithm can be identified.
>
> **7.** "In the set-up, there is no formal definition of a 'stable predictor'.'' "I think there is an error in Example 1."
>
> A predictor is a stable predictor if it is independent of the environment $e$.
>
> There is indeed a typo in the Example. 1. We have fixed it in the updated manuscript.
>
> **8.** About the extra experimental results.
>
> The exact numbers of max MSE are provided under the axis of Fig. 1 and Fig. 6. A comparison of the average (over random seeds) std. (over test environments) of MSE is provided in Tab. 4,5,6 in the appendix.
>
> On ADNI, the variables selected by the Surgery estimator include cingulum, gender, and ApoE, while the variables selected by our methods are occipital and insular. On the IMPC dataset, the Surgery estimator selects the cell counts of neutrophil and eosinophil as the optimal subset, while our method only uses the mutable variable (cell counts of the large unstained cells) for prediction.
>
> On ADNI, the oracle predictor trained in the target environment (environments with less than $100$ samples are omitted) achieves a max MSE of $0.925 \pm 0.154$, compared with the $1.00 \pm 0.04$ max MSE achieved by our method. On the IMPC dataset, neither of the test environments has more than $20$ samples, so it is hard to train the oracle predictor.

---

### Official Review · Reviewer_ra9S · 2022-11-02

**Confidence:** 3
**Correctness:** 3
**Technical Novelty And Significance:** 3
**Empirical Novelty And Significance:** 3
**Recommendation:** 5

**Clarity, Quality, Novelty And Reproducibility:**

The paper can be organized better and written more clearly. To reiterate a point made earlier, only one paragraph devoted to a method that's competitive w.r.t. the main algorithm seems less. Some notations are not clear. Example J: Pa(X_m) -> X_m was unclear - the authors probably meant alphabets? The prose can also be made more explanatory. For example, a central contribution of the paper is the concept of g-equivalence but apart from the definition there is no intuitive explanation of g-equivalence.

**Strength And Weaknesses:**

The paper’s strongest contribution is the empirical validation on the real world datasets considered albeit still being relatively low-dimensional. Although I didn’t check the proofs in detail, both Both the methods proposed seem theoretically sound and believable. The concept of equivalences is also novel and interesting in its own right.

Apart from the possibly exponential search complexity that the authors acknowledge, I have concerns about the complexity of searching over the parametrization of the shifts on mutable variables, I.e. estimation of h*(S_). How is this parametrized and why is the cost considered constant in the complexity analysis? If I am understanding the parametrization correctly, this should be exponential in the size of the mutable variables?

Given that the sparse optimization method is competitive relative to the min-max identification algorithm, it deserves more explanation than just a paragraph.



**Summary Of The Paper:**

For a supervised learning task where data comes from multiple environments, invariant prediction relies on identifying a stable set of features that don't influence the shift in probability distributions over different training environments. This paper proposes a method to find the optimal sufficient set of stable features. The method is split into two cases depending on whether the optimal set of stable features is the entire set of stable features. The paper proposes a sufficient graphical condition for the latter that can be tested by causal discovery. If the condition fails, the paper parametrizes the dataset shift and does a brute-force search to get the min-max optimal subset. To reduce the search space, an equivalence class of the set of subsets is identified. While acknowledging that this brute force search might be inefficient, the authors frame an alternative sparse min-max optimization to solve the subset selection problem. The brute force search method is validated on synthetic data and both methods are validated on real world data.

**Summary Of The Review:**

Overall, the authors propose two methods for an important problem - one of which potentially has an exponential search complexity and it's not clear to me how they parametrize the mutable shifts. The other method while having decent empirical performance is not addressed much in the theory portion of the paper. If the authors can address either of the two concerns, it would make the paper much stronger.

---

> ### Author Response · Authors · 2022-11-10
> **Responses to Reviewer ra9S**
>
> Thanks for your efforts and valuable comments on our paper. We address your concerns below:
>
> **1.** ''I have concerns about the complexity of searching over the parametrization of the shifts on mutable variables. How is this parameterized and why is the cost considered constant in the complexity analysis?''
>
> We parameterize $J$ as the neural network, due to its ability to approximate any deterministic function [1]. Specifically, we use a two-layers feedforward network with sigmoid activation in the hidden layer, as mentioned in Sect. F in the appendix. We optimize over parameters of the neural network to obtain $h^\star(S_{-})$, which means a searching over the parametrization can be omitted and the cost can be considered as constant.
>
> [1] K. Hornik, et al. “Multilayer feedforward networks are universal approximators,” Neural networks, 1989.
>
> **2.** "There is no intuitive explanation of g-equivalence.'' "The notation $J: \mathrm{Pa}(\mathbf{X}_M) \to \mathbf{X}_M$ is not clear."
>
> Intuitively, if two subsets of the stable set $S_i, S_j$ are g-equivalent, then they have the same power in predicting $Y$, *i.e.*, $P(Y|s_i,do(m))=P(Y|s_j,do(m))$
>
> The notation $J: \mathrm{Pa}(\mathbf{X}_M) \to \mathbf{X}_M$ means $J$ is a determinstic function from $\mathrm{Pa}(\mathbf{X}_M)$ to $\mathbf{X}_M$.

---

> > ### Comment · Reviewer_ra9S · 2022-11-18
> > **Reviewer Response**
> >
> > Thanks for the response. I will keep my score unchanged as I think the paper rewritten with more clarity and suggestions that i have mentioned above would make it much stronger.

---

### Decision · Program_Chairs · 2023-01-20

**Decision:**

Reject

**Justification For Why Not Higher Score:**

This is a difficult paper to follow, with a lot of clarity issues raised in the reviews across the board. The authors will benefit from a rewrite, with reviewers and myself supportive that a future version of this paper can generate bigger impact after some more time spent in reorganizing it.

**Justification For Why Not Lower Score:**

N/A

**Metareview: Summary, Strengths And Weaknesses:**

When making predictions, it can be unclear how well we will do at test time if we are exposed to possible distribution shifts. This paper taps into causal learning methods to effectively draw which subset of information is reliable to such shifts.

Strengths: a characterization of information transfer that is robust in a minimax sense, tapping into reasonably well-understood ideas of causal discovery, among others.

Weaknesses: this is a difficult paper to follow, with a lot of clarity issues raised in the reviews across the board. The authors will benefit from a rewrite, with reviewers and myself supportive that a future version of this paper can generate bigger impact after some more time spent in reorganizing it.